# DoCoM: Compressed Decentralized Optimization with Near-Optimal Sample Complexity

**Chung-Yiu Yau**                                             *cyyau@se.cuhk.edu.hk*
*The Chinese University of Hong Kong*

**Hoi-To Wai**                                                *htwai@se.cuhk.edu.hk*
*The Chinese University of Hong Kong*

**Reviewed on OpenReview:** *https://openreview.net/forum?id=W0ehjkl9x7*

## Abstract

This paper proposes the Doubly Compressed Momentum-assisted stochastic gradient tracking algorithm (DoCoM) for communication-efficient decentralized optimization. The algorithm features two main ingredients to achieve a near-optimal sample complexity while allowing for communication compression. First, the algorithm tracks both the averaged iterate and stochastic gradient using compressed gossiping consensus. Second, a momentum step is incorporated for adaptive variance reduction with the local gradient estimates. We show that DoCoM finds a near-stationary solution at all participating agents satisfying $\mathbb{E}[\|\nabla f(\theta)\|^2] = \mathcal{O}(1/T^{2/3})$ in $T$ iterations, where $f(\theta)$ is a smooth (possibly non-convex) objective function. Notice that the proof is achieved via analytically designing a new potential function that tightly tracks the one-iteration progress of DoCoM. As a corollary, our analysis also established the linear convergence of DoCoM to a global optimal solution for objective functions with the Polyak-Łojasiewicz condition. Numerical experiments demonstrate that our algorithm outperforms several state-of-the-art algorithms in practice.

## 1 Introduction

Decentralized algorithms tackle an optimization problem with inter-connected agents/workers possessing local data without a central server. For many scenarios in large-scale learning, these algorithms improve computational scalability and preserve data privacy. Owing to these reasons, decentralized algorithms have become the critical enabler for sensor networks (Schizas et al., 2007), federated learning (Konečnỳ et al., 2016; Wang et al., 2021), etc.

This paper concentrates on the *communication and sampling efficiency* issues with decentralized algorithms, which is a key bottleneck as decentralized algorithms rely heavily on the bandwidth limited inter-agent communication links (Wang et al., 2021). Inefficiently designed algorithms may lead to significant overhead and slow down to downstream applications. Several approaches have been studied to tame with this issue. The first approach is to consider algorithms that are optimal in terms of the number of communication rounds. Scaman et al. (2019); Gorbunov et al. (2019); Uribe et al. (2021) studied algorithms with an optimal iteration complexity, Sun & Hong (2019); Sun et al. (2020); Lu & De Sa (2021) focused on non-convex problems and studied lower bounds on the number of communication rounds needed. We remark that a common paradigm to achieve better communication or sampling efficiency requires multiple gradient steps (Nadiradze et al., 2021) or multiple consensus steps (Lu & De Sa, 2021).

Perhaps a more direct approach to improve communication efficiency is to apply *compression* to control bandwidth usages in every communication step of the algorithm. This idea was first studied in the context of distributed optimization where workers communicate with a central server. Many algorithms have been studied with compression strategies such as sparsification (Stich et al., 2018; Alistarh et al., 2018; Wangni et al., 2018), quantization (Wen et al., 2017; Alistarh et al., 2017; Bernstein et al., 2018; Reisizadeh et al.,

Table 1: **Comparison of decentralized stochastic optimization algorithms for smooth *non-convex* problems.** Sample complexity is the no. of samples required per agent to obtain an $\epsilon$-stationary solution, $\bar{\theta}$, such that $\mathbb{E}[\|\nabla f(\bar{\theta})\|^2] \leq \epsilon^2$. Constants $\delta, \sigma^2, \overline{G}_0, \rho$ are defined in Assumption 2.2, 2.3, 2.4, Theorem 3.1. Highlighted in red are dominant terms when $\epsilon \to 0$.

| Algorithms | Sample Complexity | Compress | Remarks |
|---|---|---|---|
| DSGD (Lian et al., 2017) | $\mathcal{O}\left(\max\left\{\frac{\sigma^2}{n}\epsilon^{-4}, \frac{n(\sigma^2+\varsigma^2)}{\rho^2\epsilon^2}\right\}\right)$ | ✗ | $\varsigma^2 = \sup\limits_{i,\theta} \|\nabla f_i(\theta) - \nabla f(\theta)\|^2$ |
| GNSD (Lu et al., 2019) | $\mathcal{O}\left(\frac{1}{C_0^2 C_1^2}\epsilon^{-4}\right)$ | ✗ | $C_0, C_1$ are not explicitly defined, see (Lu et al., 2019). |
| DeTAG (Lu & De Sa, 2021) | $\mathcal{O}\left(\max\left\{\frac{\sigma^2}{n}\epsilon^{-4}, \frac{B\log\left(n+\varsigma_0 n\epsilon^{-1}\right)}{\rho^{0.5}\epsilon^2}\right\}\right)$ | ✗ | $\varsigma_0$ is variance of init. stoc. gradient, $B$ is batch size. |
| GT-HSGD (Xin et al., 2021) | $\mathcal{O}\left(\max\left\{\frac{\sigma^3}{n}\epsilon^{-3}, \frac{\overline{G}_0}{\rho^3\epsilon^2}, \frac{n^{0.5}\sigma^{1.5}}{\rho^{2.25}\epsilon^{1.5}}\right\}\right)$ | ✗ | |
| CHOCO-SGD (Koloskova et al., 2020) | $\mathcal{O}\left(\max\left\{\frac{\sigma^2}{n}\epsilon^{-4}, \frac{G}{\delta\rho^2\epsilon^3}\right\}\right)$ | ✓ | $G = \sup\limits_{i,\theta} \mathbb{E}_{\zeta\sim\mu_i}[\|\nabla f_i(\theta;\zeta)\|^2]$ |
| BEER (Zhao et al., 2022) | $\mathcal{O}\left(\max\left\{\frac{\sigma^2}{\delta^2\rho^3}\epsilon^{-4}, \frac{1}{\delta\rho^3\epsilon^2}\right\}\right)$ | ✓ | Requires batch size of $\mathcal{O}(\sigma^2/(\delta\epsilon^2))$. |
| CEDAS (Huang & Pu, 2023) | $\mathcal{O}\left(\max\left\{\frac{\sigma^2}{n}\epsilon^{-4}, \frac{n\sigma^2}{\rho\epsilon^2}, \frac{nG_0}{\rho\epsilon}\right\}\right)$ | ✓ | $G_0 = n^{-1}\sum_{i=1}^n \|\nabla f_i(\theta_i^0)\|^2$ |
| DoCoM | $\mathcal{O}\left(\max\left\{\frac{\sigma^3}{n}\epsilon^{-3}, \frac{n\overline{G}_0}{\delta^2\rho^4\epsilon^2}, \frac{n^{1.25}\sigma^{1.5}}{\delta^{2.25}\rho^{4.5}\epsilon^{1.5}}\right\}\right)$ | ✓ | See Theorem 3.1 |

2020), low-rank approximation (Vogels et al., 2019), etc., often used in combination with error compensation (Mishchenko et al., 2019; Tang et al., 2019); also see the recent work (Richtárik et al., 2021) for more details on compression strategies.

For decentralized optimization which operates in the absence of a central server, the design of compression-enabled algorithms is more challenging. Tang et al. (2018a) proposed an extrapolation compression method, Koloskova et al. (2019; 2020) proposed the CHOCO-SGD algorithm which combines decentralized SGD (Lian et al., 2017) with error compensation, Vogels et al. (2020) studied compression with low-rank matrices, Zhao et al. (2022) considered algorithms deploying large batch size. Despite the simplicity and appealing practical performance, algorithms such as CHOCO-SGD suffer from sub-optimal iteration/sample complexity. Their analysis also shows that the performance hinges on a measure of data similarity across agents which is not ideal in light of applications such as federated learning with non-i.i.d. data. We inquire

*Can we design a compression-enabled decentralized algorithm for non-i.i.d. data with near-optimal sample complexity? Does such algorithm work well in practice?*

This paper addresses the above questions by incorporating two ingredients in decentralized optimization algorithm: (A) compression with gradient tracking, (B) momentum-based variance reduction. In summary,

- We design the *Doubly Compressed Momentum-assisted Stochastic Gradient Tracking* (DoCoM) algorithm which utilizes two levels of compressions for tackling decentralized stochastic optimization. The design of DoCoM involves a judicious combination of compression with gradient tracking to maximize convergence speed. Our algorithm finds a stationary solution without relying on restrictive conditions such as bounded similarity between data distributions found in prior compression-enabled algorithms.

- We provide a unified convergence analysis for DoCoM. Let $f(\bar{\theta})$ be the overall objective function across the network to be defined in (1), we show that DoCoM finds the averaged iterate $\bar{\theta}^T$ in $T$ iterations and communications rounds with $\mathbb{E}[\|\nabla f(\bar{\theta}^T)\|^2] = \mathcal{O}(1/T^{2/3})$ for mean square smooth objective functions, and with $\mathbb{E}[f(\bar{\theta}^T) - f^\star] = \mathcal{O}(\log T/T)$ for objective functions satisfying the Polyak-Łojasiewicz condition. Note the algorithm takes $\mathcal{O}(1)$ sample per iteration. For the latter case, we show that if deterministic gradients are available, DoCoM converges *linearly* to optimal solution.

- We note that the analysis of DoCoM comprises a number of inter-dependent error quantities, whose convergences are not straightforward to observe due to the nonlinear coupling between them. To this end, our

analysis technique, which can be of independent interest, relies on the construction of a *tight* potential function to yield the desirable (tight) bound; see Lemma 3.10.

- We empirically evaluate the performance of DoCoM on training linear models and deep learning models using synthetic and real data, on non-convex losses.

Note that recently, Zhao et al. (2022) proposed the BEER algorithm for tackling decentralized non-convex optimization with compression and optimal communication complexity. The latter has been achieved using a large batch size per iteration. Huang & Pu (2023) proposed the CEDAS algorithm which achieves an improved transient time for approaching the asymptotic convergence rate as centralized SGD. Yan et al. (2023) proposed the CDProxSGT algorithm to handle composite objective functions (possibly non-smooth) using proximal update. However, under the mean square smoothness assumption considered in this paper, the best known analysis for these algorithms only show a suboptimal sample complexity of $\mathcal{O}(\epsilon^{-4})$ as they do not incorporate a momentum-based variance reduction step as in DoCoM. We summarize the sample complexities for state-of-the-art decentralized algorithms in Table 1. As seen, DoCoM is the only algorithm with compression and an $\mathcal{O}(\epsilon^{-3})$ sample complexity. Such sample complexity matches the complexity lower bound for stochastic first order algorithms (Arjevani et al., 2022), making DoCoM the first compression-enabled algorithm to achieve near-optimal sample complexity under the mean square smoothness assumption.

**Related Works** Algorithms for decentralized optimization have been first studied in (Nedic & Ozdaglar, 2009). It has been extended to the stochastic setting (a.k.a. DSGD) in (Ram et al., 2010), and to directed graphs (Tsianos et al., 2012; Assran et al., 2019). Notably, multiple works (Qu & Li, 2017; Di Lorenzo & Scutari, 2016; Shi et al., 2015) proposed a gradient tracking technique where agents communicate local gradients to accelerate convergence.

In stochastic non-convex optimization, Lian et al. (2017) provided a performance analysis of DSGD; Lu et al. (2019) proposed GNSD which combines gradient tracking with stochastic gradient (also see (Tang et al., 2018b)); Lu & De Sa (2021) proposed DeTAG with optimal computation-communication tradeoff; Xin et al. (2021) proposed GT-HSGD which extended GNSD with momentum-based variance reduction, and similar algorithms are in (Xin et al., 2020; Zhang et al., 2021b). Note that the momentum-based variance reduction idea was proposed in (Tran-Dinh et al., 2021; Cutkosky & Orabona, 2019) to achieve optimal sampling complexity for centralized SGD. For a general overview, see (Chang et al., 2020).

On the other hand, methods for reducing communication burden in decentralized algorithms have been developed. Aysal et al. (2008); Kashyap et al. (2007); Reisizadeh et al. (2019); Saha et al. (2021) studied quantization for average consensus which is a main building block for decentralized algorithms. Notably, recent works (Liu et al., 2020; Liao et al., 2021; Song et al., 2021; Zhang et al., 2021a; Song et al., 2021) showed that combining compression with gradient tracking lead to algorithms that achieve linear convergence to an optimal solution. These algorithms bear similar structure to DoCoM, yet are limited to strongly convex objective functions and full batch gradients; see (Kovalev et al., 2021) for the extension to stochastic settings.

**Notations** $\|\cdot\|, \|\cdot\|_F$ denote Euclidean norm, Frobenius norm, respectively. The subscript-less operator $\mathbb{E}[\cdot]$ is total expectation taken over all randomnesses in operand.

## 2 Problem Setup & Background

Consider a connected, weighted and undirected graph $G = (\mathcal{N}, \mathcal{E}, \mathbf{W})$ with $\mathcal{N} = \{1, \ldots, n\}$ representing a set of $n$ agents, $\mathcal{E} \subseteq \mathcal{N} \times \mathcal{N}$ representing the communication links between agents, and $\mathbf{W} \in \mathbb{R}^{n \times n}$ is a symmetric, weighted adjacency matrix. Note that self-loops are included such that $\{i, i\} \in \mathcal{E}$ for all $i$. Our goal is to tackle the following optimization problem:

$$\min_{\theta \in \mathbb{R}^d} f(\theta) := \frac{1}{n} \sum_{i=1}^{n} f_i(\theta), \tag{1}$$

where $f_i : \mathbb{R}^d \to \mathbb{R}$ is a continuously differentiable (possibly non-convex) objective function known to the $i$th agent. The objective function can be expressed as $f_i(\theta) = \mathbb{E}_{\zeta \sim \mu_i}[f_i(\theta; \zeta)]$ such that $\mu_i$ denotes the data distribution available at agent $i$. We assume that $f(\theta) > -\infty$ for any $\theta \in \mathbb{R}^d$.

Throughout this paper, we assume the following conditions on the objective function and the adjacency matrix $\mathbf{W}$:

**Assumption 2.1.** There exists $L \in \mathbb{R}_+$ such that for any $i \in \mathcal{N}$, $\theta, \theta' \in \mathbb{R}^d$,

$$\mathbb{E}_\zeta \left[ \|\nabla f_i(\theta; \zeta) - \nabla f_i(\theta'; \zeta)\|^2 \right] \leq L^2 \|\theta - \theta'\|^2. \tag{2}$$

**Assumption 2.2.** The adjacency matrix $\mathbf{W} \in \mathbb{R}_+^{n \times n}$ satisfies: (i) $W_{ij} = 0$ if $\{i, j\} \notin \mathcal{E}$; (ii) $\mathbf{W}\mathbf{1}_n = \mathbf{W}^\top \mathbf{1}_n = \mathbf{1}_n$; (iii) let $\mathbf{U} \in \mathbb{R}^{n \times (n-1)}$ be a matrix with orthogonal columns satisfying $\mathbf{I}_n - (1/n)\mathbf{1}\mathbf{1}^\top = \mathbf{U}\mathbf{U}^\top$, there exists $\rho \in (0, 1]$ such that $\|\mathbf{U}^\top \mathbf{W}\mathbf{U}\| \leq 1 - \rho$; (iv) there exists $\bar{\omega} \in (0, 2]$ such that $\|\mathbf{W} - \mathbf{I}_n\| \leq \bar{\omega}$.

The above conditions are standard. Assumption 2.1 requires the objective function to be mean square smooth. It is also known that there exists $\mathbf{W}$ such that Assumption 2.2 is satisfied when $G$ is a connected graph, e.g., by using the Metropolis-Hastings weight; see (Boyd et al., 2004). For any $\mathbf{W}$ which is a weighted adjacency matrix on a connected graph satisfying conditions (i)-(ii), $\mathbf{1}$ is the unique eigenvector of $\mathbf{W}$. It follows that $\|\mathbf{U}^\top \mathbf{W}\mathbf{U}\| = \max\{\lambda_2, |\lambda_n|\}$ and conditions (i)-(iii) are equivalent to (Koloskova et al., 2019, Definition 1), see Appendix D.

We assume that the gradient of $f_i$ can be estimated as $\nabla f_i(\theta; \zeta)$ satisfying:

**Assumption 2.3.** There exists $\sigma \geq 0$ such that for any $\theta \in \mathbb{R}^d$, $i = 1, ..., n$, the gradient estimate $\nabla f_i(\theta; \zeta)$ with $\zeta \sim \mu_i$ is unbiased with bounded second order moment, i.e.,

$$\mathbb{E}_{\zeta \sim \mu_i}[\nabla f_i(\theta; \zeta)] = \nabla f_i(\theta), \quad \mathbb{E}_{\zeta \sim \mu_i}[\|\nabla f_i(\theta; \zeta) - \nabla f_i(\theta)\|^2] \leq \sigma^2. \tag{3}$$

Again, Assumption 2.3 is a standard setting for stochastic optimization.

**DSGD and CHOCO-SGD** Equipped with Assumption 2.2, 2.3, a common practice for tackling (1) in a decentralized manner is to utilize $\mathbf{W}$ as a mixing matrix and combine mixing with stochastic gradient descent. To illustrate the basic idea, we observe the decentralized stochastic gradient (DSGD) algorithm (Ram et al., 2010): at iteration $t \geq 0$ and all $i = 1, \ldots, n$,

$$\theta_i^{t+1} = \sum_{j=1}^n W_{ij}\theta_j^t - \eta \nabla \widehat{f}_i^t \quad \text{where} \quad \nabla \widehat{f}_i^t \equiv \nabla f_i(\theta_i^t; \zeta_i^t), \tag{4}$$

such that $\eta > 0$ is the step size and $\nabla \widehat{f}_i^t$ is a shorthand notation for the unbiased stochastic gradient with the data $\zeta_i^t \sim \mu_i$ drawn independently upon fixing $\theta_i^t$ and satisfying Assumption 2.3. For agent $i$, the *consensus step* $\sum_{j=1}^n W_{ij}\theta_j^t$ can be computed with a local average among the neighbors of $i$.

Notice that for (4), agents are required to transmit $d$ real numbers on the graph $G$ to their neighbors at every iteration. In practice, the communication links between agents are bandwidth limited. To this end, a remedy is to apply *compression* to messages transmitted on $G$. Formally, we consider a stochastic compression operator $\mathcal{Q} : \mathbb{R}^d \to \mathbb{R}^d$ satisfying the condition:

**Assumption 2.4.** For any $x \in \mathbb{R}^d$, the compressor output $\mathcal{Q}(x)$ is the random vector $\tilde{\mathcal{Q}}(x; \xi)$ with $\xi \sim \pi_x$ such that there exists $\delta \in (0, 1]$ satisfying

$$\mathbb{E}\left[\|x - \mathcal{Q}(x)\|^2\right] = \mathbb{E}\left[\|x - \tilde{\mathcal{Q}}(x; \xi)\|^2\right] \leq (1 - \delta)\|x\|^2. \tag{5}$$

The above is a general condition on compressors as discussed in (Koloskova et al., 2019). It is satisfied by a number of designs. For instance, with $k \leq d$, the top-$k$ (resp. random-$k$) *sparsifier*:

$$[\mathcal{Q}(x)]_i = x_i \quad \text{if} \quad i \in \mathcal{I}_x, \quad [\mathcal{Q}(x)]_i = 0 \quad \text{otherwise}. \tag{6}$$

where $\mathcal{I}_x \subseteq \{1, \ldots, d\}$ with $|\mathcal{I}_x| = k$ is the set of the coordinates of $x$ with the largest $k$ magnitudes (resp. uniformly selected at random), satisfies Assumption 2.4 with $\delta = k/d$. Other compressors such as random quantization (Wen et al., 2017; Alistarh et al., 2017; Stich et al., 2018; Alistarh et al., 2018) can also satisfy (5) with re-scaling; see Appendix C.4 and (Koloskova et al., 2019). Note that sending $\mathcal{Q}(x)$ in (6) over a communication channel requires only $k$ real number transmission, achieving a $k/d$ compression ratio.

However, applying $\mathcal{Q}(\cdot)$ to the consensus step in (4) directly does not lead to a convergent algorithm as (i) the compressor is not unbiased, and (ii) the compression error will accumulate with $t \to \infty$. The CHOCO-SGD algorithm (Koloskova et al., 2019) resolves the issue by incorporating an error feedback step: at iteration $t$,

$$\widehat{\theta}_i^{t+1} = \widehat{\theta}_i^t + \mathcal{Q}(\theta_i^t - \eta \nabla \widehat{f}_i^t - \widehat{\theta}_i^t), \tag{7}$$

$$\theta_i^{t+1} = \theta_i^t - \eta \nabla \widehat{f}_i^t + \gamma \sum_{j=1}^n W_{ij}(\widehat{\theta}_j^{t+1} - \widehat{\theta}_i^{t+1}), \tag{8}$$

for all $i$, where $\gamma > 0$ is the consensus step size, and $\eta, \nabla \widehat{f}_i^t$ were defined in (4). Instead of directly transmitting a compressed version of $\theta_i^t - \eta \nabla \widehat{f}_i^t$, a key feature of CHOCO-SGD is that the latter maintains an auxiliary variable $\widehat{\theta}_i^t$ that accumulates the compressed *difference* $\mathcal{Q}(\theta_i^t - \eta \nabla \widehat{f}_i^t - \widehat{\theta}_i^t)$. Koloskova et al. (2020) proved that in $T$ iterations, CHOCO-SGD finds a near-stationary solution of (1), $\{\theta_i^\mathsf{T}\}_{i=1}^n$ with $\mathsf{T} \in \{0, \ldots, T-1\}$, satisfying $\mathbb{E}[\|\nabla f(n^{-1} \sum_{i=1}^n \theta_i^\mathsf{T}))\|^2] = \mathcal{O}(1/\sqrt{T})$.

However, a drawback of CHOCO-SGD is that its convergence requires the stochastic gradient $\mathbb{E}[\|\nabla \widehat{f}_i^t\|^2]$ to be bounded for any $i, t$, see (Koloskova et al., 2019; 2020); or it can be shown that it requires the *data similarity* $\sup_{\theta \in \mathbb{R}^d} \|\nabla f_i(\theta) - \nabla f(\theta)\|$ is bounded. These conditions may not be valid when the local data are non-i.i.d. such as in the federated learning setting (Konečnỳ et al., 2016).

## 3   Proposed DoCoM Algorithm

Taking a closer look at CHOCO-SGD (8) reveals that the algorithm is only able to utilize information from the local gradient estimates $\nabla \widehat{f}_i^t \approx \nabla f_i(\theta_i^t)$ in the local update step. The local update dynamics may thus remain non-stationary even when the solution $\theta_i^t$ is close to a stationary point of (1). The issue is particularly severe when the local objective functions are not similar in the sense that $\nabla f_i(\theta) \neq \nabla f_j(\theta)$. This motivates us to design an algorithm that will make $n^{-1} \sum_{i=1}^n \nabla \widehat{f}_i^t$ available locally.

We propose the *Doubly Compressed Momentum-assisted Stochastic Gradient Tracking* (DoCoM) algorithm. Our algorithm involves two main ingredients: (A) a *gradient tracking* step with *compression* where each agent maintains an estimate of $n^{-1} \sum_{i=1}^n \nabla \widehat{f}_i^t$ at low communication cost; (B) adaptive momentum-based variance reduction that improves the variance of estimate of $\nabla \widehat{f}_i^t$ using $\mathcal{O}(1)$ sample per iteration.

Let $\eta > 0$ be step size, $\gamma, \beta \in (0, 1]$, the DoCoM algorithm at iteration $t \in \mathbb{N}$ reads: for $i = 1, \ldots, n$,

$$\theta_i^{t+1} = \theta_i^t - \eta g_i^t + \gamma \sum_{j=1}^n W_{ij} \left( \widehat{\theta}_j^{t+1} - \widehat{\theta}_i^{t+1} \right) \tag{9a}$$

$$\widehat{\theta}_i^{t+1} = \widehat{\theta}_i^t + \mathcal{Q} \left( \theta_i^t - \eta g_i^t - \widehat{\theta}_i^t \right) \tag{9b}$$

$$v_i^{t+1} = \beta \nabla \widehat{f}_i^{t+1} + (1 - \beta) \left( v_i^t + \nabla \widehat{f}_i^{t+1} - \nabla \widetilde{f}_i^t \right) \tag{9c}$$

$$g_i^{t+1} = g_i^t + v_i^{t+1} - v_i^t + \gamma \sum_{j=1}^n W_{ij} \left( \widehat{g}_j^{t+1} - \widehat{g}_i^{t+1} \right) \tag{9d}$$

$$\widehat{g}_i^{t+1} = \widehat{g}_i^t + \mathcal{Q} \left( g_i^t + v_i^{t+1} - v_i^t - \widehat{g}_i^t \right) \tag{9e}$$

In the above, we draw $\zeta_i^{t+1} \sim \mu_i$ at agent $i$ (or a minibatch of samples) as $\nabla \widehat{f}_i^{t+1} \equiv \nabla f_i(\theta_i^{t+1}; \zeta_i^{t+1})$, $\nabla \widetilde{f}_i^t \equiv \nabla f_i(\theta_i^t; \zeta_i^{t+1})$ such that the stochastic gradients in (9c) are formed using the same data batch. To implement (9c) with $\beta \neq 1$, agent $i$ needs direct access to the data batch $\zeta_i^{t+1}$ and the oracle $\nabla f_i(\cdot; \zeta_i^{t+1})$. Readers are referred to Algorithm 1 for details on the initialization and decentralized implementation.

Unlike CHOCO-SGD (8), the local update steps in (9a), (9b) are computed along the direction given by $g_i^t$. The latter is then updated according to (9d), (9e), which aims at *tracking the dynamically updated average gradient estimator* $g_i^t \approx n^{-1} \sum_{j=1}^n v_j^t$ with compressed communications given by $\mathcal{Q}(\cdot)$. In this way, we say that the DoCoM algorithm is *doubly compressed*. Furthermore, as for $v_j^t$, (9c) uses a recursive variance reduced estimate to estimate the (exact) gradient $v_j^t \approx \nabla f_j(\theta_j^t)$ (Cutkosky & Orabona, 2019; Tran-Dinh et al., 2021). Together, DoCoM yields a consensus based algorithm where the variance reduced and averaged gradient $g_i^t \approx n^{-1} \sum_{j=1}^n v_j^t$ is simultaneously available at each agent.

From an implementation perspective, DoCoM shares the communication and computation costs per iteration of the same order as CHOCO-SGD at $\mathcal{O}(d)$. In fact, only an extra communication step (with compression) is

---

**Algorithm 1** DoCoM Algorithm

---

1: **Input:** mixing matrix $\mathbf{W}$; step sizes $\eta$, $\gamma$, $\beta$; initial batch number $b_0$; initial iterate $\bar{\theta}^0 \in \mathbb{R}^d$.

2: Initialize $\quad \theta_i^0 = \bar{\theta}^0 \; \forall i \in [n]; \quad \widehat{\theta}_{i,j}^0 = \bar{\theta}^0 \; \forall \{i,j\} \in \mathcal{E}$.

3: Initialize stochastic gradient estimate

$$v_i^0 = \tfrac{1}{b_0} \sum_{r=1}^{b_0} \nabla f_i(\theta_i^0; \zeta_i^{0,r}), \{\zeta_i^{0,r}\}_{r=1}^{b_0} \sim \mu_i; \quad g_i^0 = v_i^0 \; \forall i \in [n]; \quad \widehat{g}_{i,j}^0 = \mathbf{0}_d \; \forall \{i,j\} \in \mathcal{E}.$$

4: **for** $t$ **in** $0, \dots, T-1$ **do**

5: $\quad$ (UPDATE) $\forall i \in [n]$ : Agent $i$ updates $\theta_i^{t+\frac{1}{2}} = \theta_i^t - \eta g_i^t$

6: $\quad$ **for** $\{i,j\} \in \mathcal{E}$ (notice $\{i,i\} \in \mathcal{E}$) **do**

7: $\qquad$ (PRM. GOSSIP) Agent $j$ receive $\mathcal{Q}(\theta_i^{t+\frac{1}{2}} - \widehat{\theta}_{i,i}^t)$ from agent $i$ and update $\widehat{\theta}_{j,i}^{t+1} = \widehat{\theta}_{j,i}^t + \mathcal{Q}(\theta_i^{t+\frac{1}{2}} - \widehat{\theta}_{i,i}^t)$

8: $\quad$ **end for**

9: $\quad$ (PRM. AGGREGATE) $\forall i \in [n]$ : Agent $i$ updates $\theta_i^{t+1} = \theta_i^{t+\frac{1}{2}} + \gamma \sum_{j:\{i,j\}\in\mathcal{E}} W_{ij}(\widehat{\theta}_{i,j}^{t+1} - \widehat{\theta}_{i,i}^{t+1})$

10: $\quad$ Draw data batch $\zeta_i^{t+1} \sim \mu_i$ and compute $\nabla \widehat{f}_i^{t+1} = \nabla f_i(\theta_i^{t+1}; \zeta_i^{t+1})$, $\nabla \widetilde{f}_i^t = \nabla f_i(\theta_i^t; \zeta_i^{t+1})$

11: $\quad$ (MOMENTUM) $\forall i \in [n]$ : Agent $i$ updates $v_i^{t+1} = \beta \nabla \widehat{f}_i^{t+1} + (1-\beta)(v_i^t + \nabla \widehat{f}_i^{t+1} - \nabla \widetilde{f}_i^t)$

12: $\quad$ (GRAD. TRACKER) $\forall i \in [n]$ : Agent $i$ updates $g_i^{t+\frac{1}{2}} = g_i^t + v_i^{t+1} - v_i^t$

13: $\quad$ **for** $\{i,j\} \in \mathcal{E}$ (notice $\{i,i\} \in \mathcal{E}$) **do**

14: $\qquad$ (G.T. GOSSIP) Agent $j$ receive $\mathcal{Q}(g_i^{t+\frac{1}{2}} - \widehat{g}_{i,i}^t)$ from agent $i$ and update $\widehat{g}_{j,i}^{t+1} = \widehat{g}_{j,i}^t + \mathcal{Q}(g_i^{t+\frac{1}{2}} - \widehat{g}_{i,i}^t)$

15: $\quad$ **end for**

16: $\quad$ (G.T. AGGREGATE) $\forall i \in [n]$ : Agent $i$ updates $g_i^{t+1} = g_i^{t+\frac{1}{2}} + \gamma \sum_{j:\{i,j\}\in\mathcal{E}} W_{ij}(\widehat{g}_{i,j}^{t+1} - \widehat{g}_{i,i}^{t+1})$

17: **end for**

18: **Output:** pick the $\mathsf{T}$th iterate $\theta_i^{\mathsf{T}}$, where $\mathsf{T}$ is uniformly selected from $\{0, \dots, T-1\}$ or $\mathsf{T} = T$.

---

needed for the tracking of $n^{-1} \sum_{i=1}^n v_i^t$ and an extra computation step is needed for computing $\nabla \widetilde{f}_i^t$, in (9d), (9e). A detailed comparison on computational costs is shown in Table 5. As we will show later, the above shortcomings can be compensated by the improved convergence rate of DoCoM.

## 3.1 Main Results

We show that DoCoM achieves state-of-the-art convergence rate for smooth problems. Let $\bar{\theta}^t := n^{-1} \sum_{i=1}^n \theta_i^t$ be the averaged iterate, $\overline{G}_0 := n^{-1} \mathbb{E}[\sum_{i=1}^n \|g_i^0\|^2]$ be the initial expected gradient norm, $f^\star := \min_{\theta'} f(\theta')$ be the optimal objective value. We first summarize the convergence results under the mentioned assumptions where (1) is smooth but possibly non-convex:

**Theorem 3.1.** *Under Assumption 2.1, 2.2, 2.3, 2.4. Suppose that the step sizes satisfies*

$$\eta \leq \min\left\{\eta_\infty, \sqrt{\overline{\beta} n/(8\mathbb{C}_{\bar{g}})}\right\}, \quad \gamma \leq \gamma_\infty, \tag{10}$$

*where $\gamma_\infty, \eta_\infty$ are defined in (24). Set $\beta \in (0,1)$, $\overline{\beta} = \min\{\frac{\rho\gamma}{8}, \frac{\delta\gamma}{8}, \beta\}$. For any $T \geq 1$, it holds*

$$\sum_{t=0}^{T-1} \mathbb{E}\left[\frac{\|\nabla f(\bar{\theta}^t)\|^2}{2T} + \frac{L^2}{nT}\sum_{i=1}^n \|\theta_i^t - \bar{\theta}^t\|^2\right] \leq \frac{f(\bar{\theta}^0) - f^\star}{\eta T} + \mathbb{C}_\sigma \frac{2\beta^2\sigma^2}{\overline{\beta} n} + \frac{4\sigma^2}{b_0 \overline{\beta} T n} + \frac{\eta^2}{\overline{\beta} T}\frac{236 L^2 \overline{G}_0}{\rho^2 \gamma^2 (1-\gamma)},$$

*where*

$$\mathbb{C}_\sigma = 4 + \frac{\eta^2}{\gamma^3}\frac{672 L^2 n}{\rho^3} + \frac{\eta^2}{\gamma}\frac{6 L^2 n \rho^4 \delta}{25 \bar{\omega}^2} + \frac{\eta^2}{\gamma^2}\frac{4 L^2 n}{\bar{\omega}^2}, \tag{11}$$

$$\mathbb{C}_{\bar{g}} = 8(1-\beta)^2 L^2 (1-\rho\gamma)^2 + \frac{L^2 n}{\rho\gamma}\left(96 + \frac{141}{400}\frac{\rho^2}{\bar{\omega}^2}\right).$$

We provide the proof of Theorem 3.1 in Section 3.2. Below we discuss its main consequences.

**Near-optimal Iteration/Sample Complexity** Setting the step sizes and parameters as $\eta = \frac{n^{2/3}}{LT^{1/3}}, \gamma = \gamma_\infty, \beta = \frac{n^{1/3}}{T^{2/3}}, b_0 = \frac{T^{1/3}}{n^{2/3}}$. Further, we select the Tth iterate as the output of DoCoM such that T is independently and uniformly selected from $\{0, \ldots, T-1\}$ [cf. the output of Algorithm 1], similar to (Ghadimi & Lan, 2013). For a sufficiently large $T$, it can be shown that

$$\frac{1}{n}\sum_{i=1}^n \mathbb{E}\left[\left\|\nabla f(\theta_i^{\mathsf{T}})\right\|^2\right] = \mathcal{O}\left(\frac{L(f(\bar\theta^0) - f^\star)}{(nT)^{2/3}} + \frac{\sigma^2}{(nT)^{2/3}} + \frac{n\overline{G}_0}{\delta^2\rho^4 T} + \frac{\sigma^2 n^{5/3}}{\delta^3\rho^6 T^{4/3}}\right), \tag{12}$$

where we have used the Lipschitz continuity of $\nabla f_i(\cdot)$ [cf. Assumption 2.1] to derive a bound on the gradient of individual iterate $\theta_i^{\mathsf{T}}$.

For any agent $i = 1, \ldots, n$, the iterate $\theta_i^{\mathsf{T}}$ is guaranteed to be $\mathcal{O}(1/T^{2/3})$-stationary to (1). Notice that this is the state-of-the-art convergence rate for first order stochastic optimization even in the centralized setting; see (Cutkosky & Orabona, 2019; Tran-Dinh et al., 2021); and it also matches the lower bound in (Arjevani et al., 2022). Our rate is comparable to or faster than a number of decentralized algorithms with or without compression; see Table 1. Further, we remark that Theorem 3.1 does not impose condition on $f_i$'s similarity $\sup_\theta \|\nabla f_i(\theta) - \nabla f(\theta)\|$ in which our convergence rate is independent of the data heterogeneity.

Note that the step size configuration in (12) requires $n \leq \mathcal{O}(L^{3/4}T^{1/2})$ in order to satisfy $\eta \leq \eta_\infty$, thereby necessitating $T = \Omega(n^2)$. As an alternative, it is possible to select $\eta = \frac{1}{LT^{1/3}}, \gamma = \gamma_\infty, \beta = \frac{n^{1/3}}{T^{2/3}}, b_0 = \frac{T^{1/3}}{n^{2/3}}$, which yields

$$\frac{1}{n}\sum_{i=1}^n \mathbb{E}\left[\left\|\nabla f(\theta_i^{\mathsf{T}})\right\|^2\right] = \mathcal{O}\left(\frac{L(f(\bar\theta^0) - f^\star)}{T^{2/3}} + \frac{\sigma^2}{(nT)^{2/3}} + \frac{\overline{G}_0}{n^{1/3}\delta^2\rho^4 T} + \frac{\sigma^2 n^{5/3}}{\delta^3\rho^6 T^{4/3}}\right). \tag{13}$$

In this case satisfying $\eta \leq \eta_\infty$ requires $n \leq \mathcal{O}(L^{3/2}T)$ and thus $T = \Omega(n)$ similar to Koloskova et al. (2020). However, as a trade-off, it gives a worse dependence on $f(\bar\theta^0) - f^\star$.

**Impacts of Network Topology and Compressor** Eq. (12) indicates the impacts of network topology (due to $\rho$) and compressor (due to $\delta$) vanish as $T \to \infty$. This can be observed by recognizing that the last two terms in (12) are $\mathcal{O}(1/T), \mathcal{O}(1/T^{4/3})$. In Appendix A.9, we demonstrate with a similar set of step sizes, for any $T \geq T_{\mathsf{trans}} = \Omega(n^3\overline{G}_0^3/(\sigma^6\delta^6\rho^{12}))$, DoCoM enjoys a matching convergence behavior as a centralized SGD algorithm employing a momentum-based variance reduced gradient estimator with a batch size of $n$, e.g., (Tran-Dinh et al., 2021). In the latter case, we have $n^{-1}\sum_{i=1}^n \mathbb{E}\left[\left\|\nabla f(\theta_i^{\mathsf{T}})\right\|^2\right] = \mathcal{O}(\sigma^2/(nT^{2/3}))$. The constant $T_{\mathsf{trans}}$ is also known as the transient time of decentralized algorithm (Pu et al., 2020).

Our result does not require any assumption on the data heterogeneity level nor the boundedness of gradient as in CHOCO-SGD (Koloskova et al., 2020) or DSGD (Lian et al., 2017). As hinted before, this is a consequence of gradient tracking. In Appendix B, we provide a separate analysis for the case of $\beta = 1$ when no momentum is applied in (9c). Interestingly, in the latter case, the convergence rate is only $\mathcal{O}(1/\sqrt{T})$ [cf. (60)], indicating that the momentum term is crucial in accelerating DoCoM.

**PL Condition** Finally, we show that the convergence rate can be improved when the objective function satisfies the Polyak-Łojasiewicz (PL) condition:

**Assumption 3.2.** For any $\theta \in \mathbb{R}^d$, it holds that $\|\nabla f(\theta)\|^2 \geq 2\mu[f(\theta) - f^\star]$ for some $\mu > 0$.

Notice that the PL condition is satisfied by strongly convex functions as well as a number of non-convex functions; see Karimi et al. (2016). We obtain:

**Corollary 3.3.** *Under Assumption 2.1, 2.2, 2.3, 2.4, 3.2. Suppose that the step size condition (10) holds and $\beta \in (0,1)$. Then, for any $t \geq 1$, it holds*

$$\Delta^t + \frac{2L^2\eta}{\overline{\beta}n}\sum_{i=1}^n \mathbb{E}[\|\theta_i^t - \bar{\theta}^t\|^2] \leq \left(1 - \widetilde{\beta}\right)^t \left(\Delta^0 + \frac{2\eta}{\overline{\beta}n}\mathtt{V}^0\right) + \frac{\eta\beta^2}{\overline{\beta}\widetilde{\beta}}\frac{2\mathbb{C}_\sigma\sigma^2}{n}, \tag{14}$$

*where $\widetilde{\beta} := \min\left\{\eta\mu, \overline{\beta}/2\right\}$, $\Delta^t := \mathbb{E}[f(\bar{\theta}^t)] - f^\star$ is the expected optimality gap and the constant $\mathbb{C}_\sigma$ is defined in (11). Notice that $\mathtt{V}^0$ can be upper bounded with (29).*

Setting the step sizes and parameters as $\eta = \log T/T, \gamma = \gamma_\infty, \beta = \log T/T, b_0 = \Omega(1)$. For sufficiently large $T$, it can be shown that

$$\mathbb{E}[f(\bar{\theta}^T)] - f^\star = \mathcal{O}(\log T/T), \tag{15}$$

$$\frac{1}{n}\sum_{i=1}^n \mathbb{E}[\|\theta_i^T - \bar{\theta}^T\|^2] = \mathcal{O}(\log T/T), \tag{16}$$

see Appendix A.10. Moreover, in the *deterministic gradient case with $\sigma^2 = 0$*, we can select a constant $\beta, \eta$. Then, (14) shows that DoCoM converges *linearly* to an optimal solution such that $\mathbb{E}[f(\bar{\theta}^T)] - f^\star = \mathcal{O}((1-\widetilde{\beta})^T)$. We remark that the latter rates match the recent algorithms with compression (Liu et al., 2020; Liao et al., 2021; Song et al., 2021; Kovalev et al., 2021) for strongly convex problems.

### 3.2 Proof of Theorem 3.1

We preface the proof by defining the following notations for the variables in DoCoM. For any $t \geq 0$:

$$\Theta^t = \begin{pmatrix} (\theta_1^t)^\top \\ \vdots \\ (\theta_n^t)^\top \end{pmatrix}, V^t = \begin{pmatrix} (v_1^t)^\top \\ \vdots \\ (v_n^t)^\top \end{pmatrix}, G^t = \begin{pmatrix} (g_1^t)^\top \\ \vdots \\ (g_n^t)^\top \end{pmatrix}$$

which are $n \times d$ matrices. Similarly, we define the matrices $\widehat{\Theta}^t, \widehat{G}^t$ based on $\{\hat{\theta}_i^t\}_{i=1}^n, \{\hat{g}_i^t\}_{i=1}^n$, and the matrices $\nabla\widehat{F}^t, \nabla\widetilde{F}^t, \nabla F$ based on $\{\nabla\hat{f}_i^t\}_{i=1}^n, \{\nabla\tilde{f}_i^t\}_{i=1}^n, \{\nabla f_i(\theta_i^t)\}_{i=1}^n$.

The norm of the matrix $\Theta_o^t = \mathbf{U}^\top\Theta^t$, i.e., $\|\mathbf{U}^\top\Theta^t\|_F^2 = \|\mathbf{U}\mathbf{U}^\top\Theta^t\|_F^2 = \|(\mathbf{I} - (1/n)\mathbf{1}\mathbf{1}^\top)\Theta^t\|_F^2$, measures *consensus error* of the iterate $\Theta^t$. We denote $G_o^t = \mathbf{U}^\top G^t$ such that $\|G_o^t\|_F^2$ measures *consensus error* of $G^t$.

Denote the average variables $\bar{\theta}^t = n^{-1}\mathbf{1}^\top\Theta^t$, $\bar{v}^t = n^{-1}\mathbf{1}^\top V^t$, $\bar{g}^t = n^{-1}\mathbf{1}^\top G^t$, $\overline{\nabla F}^t = n^{-1}\mathbf{1}^\top\nabla F^t$. We first make the following observation regarding the $\bar{\theta}^t$-update:

**Lemma 3.4.** *Under Assumption 2.1 and the step size condition $\eta \leq \frac{1}{2L}$. Then, for any $t \geq 0$, it holds*

$$f(\bar{\theta}^{t+1}) \leq f(\bar{\theta}^t) - \frac{\eta}{2}\left\|\nabla f(\bar{\theta}^t)\right\|^2 + \frac{L^2\eta}{n}\left\|\Theta_o^t\right\|_F^2 + \eta\left\|\bar{v}^t - \overline{\nabla F}^t\right\|^2 - \frac{\eta}{4}\left\|\bar{g}^t\right\|^2. \tag{17}$$

The proof is relegated to Appendix A.1. The above lemma utilizes just Assumption 2.1 and results in a deterministic bound of $f(\bar{\theta}^{t+1})$. We highlight that the bound contains a negative term of the stochastic gradient $-\frac{\eta}{4}\|\bar{g}^t\|^2$, which is slightly different from the standard bound implied by the descent lemma. Such negative term is crucial for deriving the near-optimal sampling complexity of DoCoM in Theorem 3.1, also see (Cutkosky & Orabona, 2019).

Lemma 3.4 shows that controlling $\left\|\nabla f(\bar{\theta}^t)\right\|^2$ requires bounding $\|\Theta_o^t\|_F^2$ and $\|\bar{v}^t - \overline{\nabla F}^t\|^2$. While the latter are anticipated to converge to zero, we see that characterizing their convergence results in a set of *coupled recursions* as follows:

**Lemma 3.5.** *Under Assumption 2.2, 2.4. Then, for any $t \geq 0$, it holds*

$$\mathbb{E}[\|\Theta_o^{t+1}\|_F^2] \leq (1 - \frac{\rho\gamma}{2})\mathbb{E}[\|\Theta_o^t\|_F^2] + \frac{2}{\rho}\frac{\eta^2}{\gamma}\mathbb{E}[\|G_o^t\|_F^2] + \frac{\bar{\omega}^2}{\rho}\gamma\,\mathbb{E}\left[\left\|\Theta^t - \eta G^t - \widehat{\Theta}^t\right\|_F^2\right]. \tag{18}$$

**Lemma 3.6.** *Under Assumption 2.1, 2.2, 2.3, 2.4 and let $\beta \in [0, 1)$. Then, for any $t \geq 0$, it holds*

$$\mathbb{E}\left[\left\|\bar{v}^{t+1} - \overline{\nabla F}^{t+1}\right\|^2\right] \tag{19}$$

$$\leq (1-\beta)^2 \mathbb{E}\left[\left\|\bar{v}^t - \overline{\nabla F}^t\right\|^2\right] + 2\beta^2 \frac{\sigma^2}{n} + (1-\beta)^2 \frac{8L^2}{n^2} \eta^2 (1-\rho\gamma)^2 \mathbb{E}\left[\left\|G_o^t\right\|_F^2 + \frac{n}{2}\left\|\bar{g}^t\right\|^2\right]$$

$$+ (1-\beta)^2 \frac{8L^2}{n^2} \bar{\omega}^2 \gamma^2 \mathbb{E}\left[\frac{1-\delta}{2}\left\|\Theta^t - \eta G^t - \widehat{\Theta}^t\right\|_F^2 + \left\|\Theta_o^t\right\|_F^2\right].$$

The proofs are in Appendix A.2, A.3. Notice that (18), (19) further depend on the quantities $\mathbb{E}[\|G_o^t\|_F^2]$, $\mathbb{E}[\|\Theta^t - \eta G^t - \widehat{\Theta}^t\|_F^2]$, $\mathbb{E}[\|G^t - \widehat{G}^t\|_F^2]$, which are handled by the following lemmas:

**Lemma 3.7.** *Under Assumption 2.1, 2.2, 2.3, 2.4 and the step size conditions $\eta \leq \frac{\rho\gamma}{10L(1-\rho\gamma)\sqrt{1+\gamma^2\bar{\omega}^2}}$, $\gamma \leq \frac{1}{8\bar{\omega}}$. For any $t \geq 0$, it holds*

$$\mathbb{E}\left[\left\|G_o^{t+1}\right\|_F^2\right] \leq \left(1 - \frac{\rho\gamma}{4}\right)\mathbb{E}\left[\left\|G_o^t\right\|_F^2\right] + \gamma\frac{2\bar{\omega}^2}{\rho}\mathbb{E}\left[\left\|G^t - \widehat{G}^t\right\|_F^2\right] + \gamma\frac{25L^2\bar{\omega}^2}{\rho}\mathbb{E}\left[\left\|\Theta_o^t\right\|_F^2\right] \tag{20}$$

$$+ \gamma\frac{13L^2}{\rho}\bar{\omega}^2(1-\delta)\mathbb{E}\left[\left\|\Theta^t - \eta G^t - \widehat{\Theta}^t\right\|_F^2\right]$$

$$+ \gamma\frac{\rho n}{5}\mathbb{E}\left[\left\|\bar{g}^t\right\|^2\right] + \frac{7}{\rho\gamma}\beta^2\mathbb{E}\left[\left\|V^t - \nabla F^t\right\|_F^2\right] + \frac{7n}{\rho\gamma}\beta^2\sigma^2.$$

**Lemma 3.8.** *Under Assumption 2.1, 2.2, 2.3, 2.4 and the step size conditions $\eta \leq \min\{\frac{\rho\gamma}{10L(1-\rho\gamma)\sqrt{1+\gamma^2\bar{\omega}^2}}, \frac{1}{4L}\}$, $\gamma^2 \leq \min\{\frac{\delta}{16\bar{\omega}^2(1-\delta)(1+3\eta^2L^2)(1+2/\delta)}, \frac{1}{\rho^2}, \frac{\delta^2}{64\bar{\omega}^2}\}$, $\eta^2\gamma \leq \frac{\delta^2\rho}{1248\bar{\omega}^2L^2}$. For any $t \geq 0$, it holds*

$$\mathbb{E}\left[\left\|\Theta^{t+1} - \eta G^{t+1} - \widehat{\Theta}^{t+1}\right\|_F^2\right] \leq \left(1 - \frac{\delta}{8}\right)\mathbb{E}\left[\left\|\Theta^t - \eta G^t - \widehat{\Theta}^t\right\|_F^2\right] \tag{21}$$

$$+ \eta^2\frac{50}{\delta}\mathbb{E}\left[\left\|G_o^t\right\|_F^2\right] + \eta^2\frac{3\bar{\omega}}{\rho}\mathbb{E}\left[\left\|G^t - \widehat{G}^t\right\|_F^2\right]$$

$$+ \left[\frac{29}{\delta}\bar{\omega}^2\gamma^2 + \frac{38L^2\bar{\omega}\eta^2}{\rho}\right]\mathbb{E}\left[\left\|\Theta_o^t\right\|_F^2\right] + \frac{18\eta^2}{\delta}n\mathbb{E}\left[\left\|\bar{g}^t\right\|^2\right]$$

$$+ \left(18 + \frac{84}{\rho\gamma}\right)\frac{\beta^2\eta^2}{\delta}\mathbb{E}\left[\left\|V^t - \nabla F^t\right\|_F^2\right] + \left(18 + \frac{84}{\rho\gamma}\right)\frac{\beta^2\eta^2n\sigma^2}{\delta}.$$

**Lemma 3.9.** *Under Assumption 2.1, 2.2, 2.3, 2.4 and the step size conditions $\gamma \leq \frac{\delta}{8\bar{\omega}}$, $\eta \leq \frac{\rho\gamma}{10L(1-\rho\gamma)\sqrt{1+\gamma^2\bar{\omega}^2}}$. For any $t \geq 0$, it holds*

$$\mathbb{E}\left[\left\|G^{t+1} - \widehat{G}^{t+1}\right\|_F^2\right] \leq \left(1 - \frac{\delta}{8}\right)\mathbb{E}\left[\left\|G^t - \widehat{G}^t\right\|_F^2\right] + \frac{10}{\delta}\gamma^2\left(\bar{\omega}^2 + \frac{\rho^2}{8}\right)\mathbb{E}\left[\left\|G_o^t\right\|_F^2\right] \tag{22}$$

$$+ \gamma^2\frac{122L^2\bar{\omega}^2}{\delta}\mathbb{E}\left[\left\|\Theta_o^t\right\|_F^2\right] + \gamma^2\frac{60L^2\bar{\omega}^2}{\delta}\mathbb{E}\left[\left\|\Theta^t - \eta G^t - \widehat{\Theta}^t\right\|_F^2\right]$$

$$+ \gamma^2\frac{3\rho^2}{5\delta}n\mathbb{E}\left[\left\|\bar{g}^t\right\|^2\right] + \frac{31}{\delta}\beta^2\mathbb{E}\left[\left\|V^t - \nabla F^t\right\|_F^2\right] + \frac{31}{\delta}\beta^2n\sigma^2.$$

The proofs are in Appendix A.4, A.5, A.6.

**Potential Function** As we are equipped with the above lemmas, showing the convergence of DoCoM requires tracking the error quantities in a unified fashion. This may not be obvious at the first glance due to the coupling between error quantities illustrated in the lemmas. Naturally, one can proceed by defining the sequence of potential function values:

$$\mathsf{V}^t = \mathbb{E}\left[L^2\left\|\Theta_o^t\right\|_F^2 + n\left\|\bar{v}^t - \overline{\nabla F}^t\right\|^2 + \frac{1}{n}\left\|V^t - \nabla F^t\right\|_F^2\right]$$

$$+ \mathbb{E}\left[ a \left\| G_o^t \right\|_F^2 + b \left\| G^t - \widehat{G}^t \right\|_F^2 + c \left\| \Theta^t - \eta G^t - \widehat{\Theta}^t \right\|_F^2 \right] \tag{23}$$

that comprises of the coupled error quantities, where $a, b, c > 0$ are constants to be determined. Our plan is then to study the convergence of $\mathsf{V}^t$. However, fully specifying the potential function and to ensure the *near-optimal sample complexity* for DoCoM require finding the tight conditions on $a, b, c > 0$, together with the step size conditions, which is not trivial as it requires approximately solving a $5 \times 5$ system of (nonlinear) inequalities; see (46) in the appendix.

In Appendix A.7, we provide a systematic construction for finding the parameters of the tight potential function, which results in the following lemma:

**Lemma 3.10.** *Under Assumption 2.1, 2.2, 2.3, 2.4 and let $\beta \in (0,1)$. Suppose that the step sizes satisfy:*

$$\gamma \leq \min\left\{ \frac{1}{4\rho}, \frac{\rho n}{64\bar{\omega}^2}, \frac{\delta}{10\bar{\omega}}, \frac{\delta\rho\sqrt{1 - \delta/(8\bar{\omega}^2)}}{259\bar{\omega}^2} \right\} =: \gamma_\infty,$$

$$\eta \leq \frac{\gamma}{L} \min\left\{ \sqrt{\frac{1-\beta}{\beta n}} \frac{\sqrt{\gamma\rho^3}}{45}, \frac{\rho^2}{240\bar{\omega}} \right\} =: \eta_\infty. \tag{24}$$

*Set the parameters in the potential function $\mathsf{V}^t$ such that*

$$a = \frac{96L^2}{\rho^2\gamma^2}\eta^2, \quad b = \frac{\eta^2}{\gamma(1-\gamma)} \frac{3072\bar{\omega}^2 L^2}{\delta\rho^3}, \quad c = \frac{\gamma}{1-\gamma} \frac{48L^2\bar{\omega}^2}{\delta\rho}. \tag{25}$$

*Then, for any $t \geq 0$, it holds*

$$\mathsf{V}^{t+1} \leq (1 - \overline{\beta})\mathsf{V}^t + \beta^2 \mathbb{C}_\sigma \sigma^2 + \eta^2 \mathbb{C}_{\bar{g}} \mathbb{E}\left[ \left\| \bar{g}^t \right\|^2 \right], \tag{26}$$

*where $\mathbb{C}_\sigma, \mathbb{C}_{\bar{g}}, \overline{\beta}$ were defined in Theorem 3.1.*

The above lemma shows that the potential function $\mathsf{V}^t$ is connected to the noise variance $\sigma^2$ and the gradient norm $\left\| \bar{g}^t \right\|^2$. The convergence of the latter term is of interest to our theorem. We observe the following consequence.

Equipped with (26) and define $\Delta^t := \mathbb{E}[f(\bar{\theta}^t)] - f^\star$. From Lemma 3.4, we can deduce that

$$\Delta^{t+1} + \frac{2\eta}{n\overline{\beta}}\mathsf{V}^{t+1} \leq \Delta^t + \frac{2\eta}{n\overline{\beta}}\mathsf{V}^t + \frac{2\eta}{n\overline{\beta}}\beta^2\mathbb{C}_\sigma\sigma^2 - \eta\,\mathbb{E}\left[ \frac{1}{2}\left\| \nabla f(\bar{\theta}^t) \right\|^2 + \frac{L^2}{n}\left\| \Theta_o^t \right\|_F^2 \right] \tag{27}$$

$$+ \left( (n\overline{\beta})^{-1}2\eta^3\mathbb{C}_{\bar{g}} - 4^{-1}\eta \right) \mathbb{E}\left[ \left\| \bar{g}^t \right\|^2 \right].$$

Setting $\eta \leq \sqrt{\frac{\overline{\beta}n}{8\mathbb{C}_{\bar{g}}}}$ as in (10) shows that the last term in the r.h.s. of the above can be upper bounded by zero. Summing up both sides of (27) from $t = 0$ to $t = T - 1$ yields

$$\eta \sum_{t=0}^{T-1} \mathbb{E}\left[ \frac{1}{2}\left\| \nabla f(\bar{\theta}^t) \right\|^2 + \frac{L^2}{n}\left\| \Theta_o^t \right\|_F^2 \right] \leq \Delta^0 + \frac{2\eta}{n\overline{\beta}}\mathsf{V}^0 + \frac{2\eta T}{n\overline{\beta}}\beta^2\mathbb{C}_\sigma\sigma^2. \tag{28}$$

Furthermore, with the initialization, choice of $a, b, c$ and the step size $\gamma \leq \gamma_\infty$, it can be shown that

$$\mathsf{V}^0 \leq \frac{2\sigma^2}{b_0} + \frac{118L^2 n}{\rho^2\gamma^2(1-\gamma)}\overline{G}_0\eta^2. \tag{29}$$

Dividing (28) by $\eta T$ and observing $\left\| \Theta_o^t \right\|_F^2 = \left\| (\mathbf{I} - (1/n)\mathbf{1}\mathbf{1}^\top)\Theta^t \right\|_F^2$ concludes the proof.

**Proof of Corollary 3.3** Applying the PL condition of Assumption 3.2 to the inequality (17) shows

$$\Delta^{t+1} \leq (1 - \eta\mu)\Delta^t + \eta\mathbb{E}\left[ \frac{L^2}{n}\left\| \Theta_o^t \right\|_F^2 + \left\| \bar{v}^t - \overline{\nabla F}^t \right\|^2 \right] - \frac{\eta}{4}\mathbb{E}\left[ \left\| \bar{g}^t \right\|^2 \right]$$

$$\leq (1 - \eta\mu)\Delta^t + \frac{\eta}{n}\mathsf{V}^t - \frac{\eta}{4}\mathbb{E}\left[\left\|\bar{g}^t\right\|^2\right] \tag{30}$$

Combining with Lemma 3.10 shows that

$$\Delta^{t+1} + \frac{2\eta}{\overline{\beta}n}\mathsf{V}^{t+1} \leq \left(1 - \widetilde{\beta}\right)\left[\Delta^t + \frac{2\eta}{\overline{\beta}n}\mathsf{V}^t\right] + \frac{\eta\beta^2}{\overline{\beta}n}2\mathbb{C}_\sigma\sigma^2 + \left(\frac{\eta^3}{\overline{\beta}n}2\mathbb{C}_{\bar{g}} - \frac{\eta}{4}\right)\mathbb{E}\left[\left\|\bar{g}^t\right\|^2\right], \tag{31}$$

where we used $1 - \overline{\beta} + \frac{\overline{\beta}n}{2\eta}\frac{\eta}{n} \leq 1 - \min\{\eta\mu, \overline{\beta}/2\}$. Set $\eta^2 \leq \frac{\overline{\beta}n}{4\mathbb{C}_{\bar{g}}}$ and telescope the relation concludes the proof.

## 4 Numerical Experiments

**Setup**  We run the decentralized optimization algorithms on a 40 threads Intel(R) Xeon(R) Gold 6148 CPU @ 2.40GHz server with MPI-enabled PyTorch and evaluate the performance of trained models on a Tesla K80 GPU server. To simulate heterogeneous data distribution, each agent has a disjoint set of training samples, while we evaluate each trained model on all training/testing data.

**Hyperparameter Tuning**  For all algorithms we choose the learning rate $\eta$ from $\{0.1, 0.01, 0.001\}$, and fix the regularization parameter as $\lambda = 10^{-4}$ [cf. (32)]. For compressed algorithms, we implement the top-$k$ compressor and random quantizer, and we tune the consensus step size $\gamma$ starting from the theoretical value of $\delta$. For DeTAG, we adopt the parameters from (Lu & De Sa, 2021). For DoCoM and GT-HSGD, we choose the best momentum parameter $\beta$ in $\{0.0001, 0.001, 0.01, 0.1, 0.5, 0.9\}$ and fix the initial batch number as $b_{0,i} = m_i$. We choose the batch sizes such that all algorithms spend the same amount of computation on stochastic gradient per iteration, except for BEER which requires large batch size according to (Zhao et al., 2022). The tuned parameters and additional numerical results can be found in Appendix C.

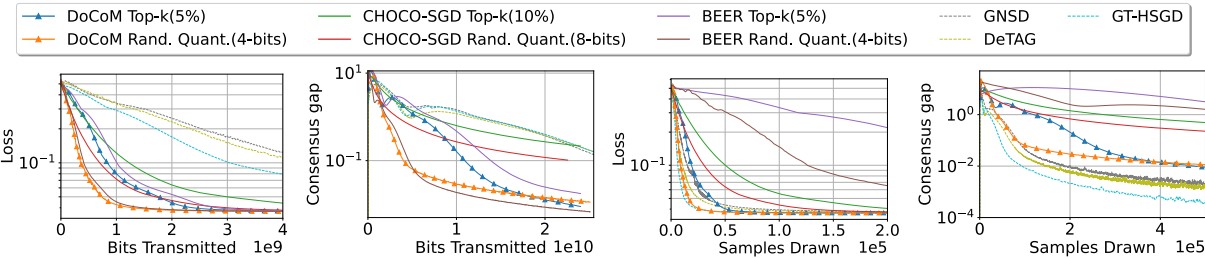

Figure 1: **Experiments on Synthetic Data with Linear Model.** Worst-agent's train loss value and consensus gap against the number of bits transmitted (left) and total number of samples drawn for gradient approximation (right).

**Synthetic Data with Linear Model**  Consider a set of synthetic data generated with the leaf benchmarking framework (Caldas et al., 2019) which provides features from agent-dependent distributions. The task is to train a linear classifier for a set of $d = 1000$-dimensional features with $m = 1443$ samples partitioned into $n = 25$ non-i.i.d. portions, each held by an agent that is connected to the others on a ring graph with uniform edge weights. Each feature vector is labeled into one of 5 classes. Altogether, the local dataset for the $i$th agent is given by $\{x_j^i, \{\ell_{j,k}^i\}_{k=1}^5\}_{j=1}^{m_i}$, where $m = \sum_{i=1}^{25} m_i$, $x_j^i \in \mathbb{R}^{1000}$ denotes the $j$th feature, and $\{\ell_{j,k}^i\}_{k=1}^5 \in \{0,1\}^5$ is the label such that $\ell_{j,k}^i = 1$ if the $j$th feature has label $k \in \{1, ..., 5\}$.

To train a linear classifier $\theta = (\theta_1, \ldots, \theta_5) \in \mathbb{R}^{5000}$, we consider (1) with the following objective function that models a modified logistic regression problem with sigmoid loss and $\ell_2$ regularization:

$$f_i(\theta) = \frac{1}{m_i}\sum_{j=1}^{m_i}\sum_{k=1}^5 \phi\left(\ell_{j,k}^i \left\langle x_j^i \mid \theta_k\right\rangle\right) + \frac{\lambda}{2}\left\|\theta\right\|_2^2, \tag{32}$$

where $\phi(z) = (1 + e^{-z})^{-1}$ and $\lambda = 10^{-4}$ is the regularization parameter. The function $f_i(\cdot)$ is not convex, and we estimate its gradient by sampling a mini-batch of data.

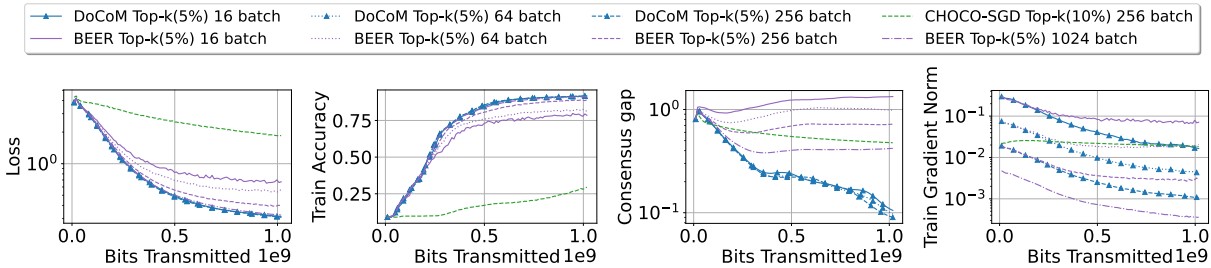

Figure 2: **Experiments on MNIST Data with Feed-forward Network.** Worst-agent's train loss values, train accuracy, consensus gap and train gradient norm against the number of bits transmitted.

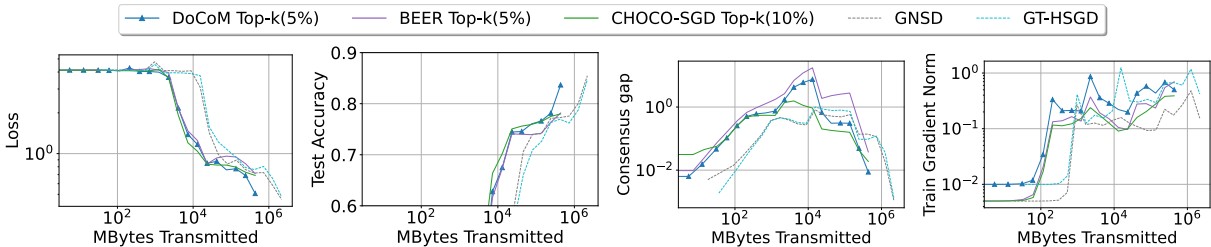

Figure 3: **Experiments on FEMNIST Data with LeNet-5.** Worst-agent's train loss values, test accuracy, consensus gap and train gradient norm against the number of MBytes transmitted.

Figure 1 compares the worst agent's loss values $\max_i f(\theta_i^t)$ and consensus gap $\sum_{i=1}^n \|\theta_i^t - \bar{\theta}^t\|$ against the communication and gradient computation costs. For compressed algorithms that require communication of two compressed variables (DoCoM and BEER), we use half the amount of bits/retained non-zeros after sparsification with (6) to make a fair comparison with CHOCO-SGD. DoCoM achieves the fastest convergence in terms of the communication cost (number of bits transmitted) and shows fast convergence on par with uncompressed algorithms in terms of gradient computation cost, when used with a 4-bit random quantizer. Comparing among compressed algorithms, DoCoM and BEER find solutions with the lower consensus gap (10 times lower than CHOCO-SGD) and DoCoM stands out to be more sample efficient than all existing compressed approaches. Observe that DoCoM outperforms CHOCO-SGD and BEER due to the use of gradient tracking and variance reduction.

**MNIST Data with Feed-forward Network**   We consider training a 1 hidden layer (with 100 neurons) feed-forward neural network with sigmoid activation function on the MNIST dataset. The samples are partitioned into $n = 10$ agents where each agent only gets 1 class of samples. These agents are arranged according to a ring topology with uniform edge weights. We tackle (1) with $f_i(\theta)$ taken as the cross entropy loss function of the local dataset and an $\ell_2$ regularization is applied with the parameter of $\lambda = 10^{-4}$.

Figure 2 compares the worst-agent's loss function, $\max_i f(\theta_i^t)$, and other metrics in the same manner against the communication cost (i.e., bits transmitted). We observe that DoCoM already achieved nearly the best performance in loss and accuracy using just a small batch size of 16. On the other hand, CHOCO-SGD suffered from slower convergence due to the heterogeneity nature of data under the unshuffled MNIST setup, and the performance of BEER is sensitive to the choice of batch sizes. Notice that in this experiment, we selected a compression ratio $k/d$ of 0.05 for DoCoM and BEER, and 0.1 for CHOCO-SGD for a fair comparison.

**FEMNIST Data with LeNet-5**   Lastly, we consider training the LeNet-5 (with $d = 60850$ parameters) neural network on the FEMNIST dataset. The dataset contains $m = 805263$ samples of $28 \times 28$ hand-written character images, each belonging to one of the 62 classes. The samples are partitioned into $n = 36$ agents according to the groups specified in (Caldas et al., 2019). These agents are arranged according to a ring topology with uniform edge weights. We scale the learning rate $\eta$ by 0.1 at the $\{4, 40, 80\}$-th thousand

iteration and momentum $\beta$ by 0.1 at the $\{10, 40, 80\}$-th thousand iteration for `DoCoM`; see Table 4 in the appendix for the hyperparameters values.

Denote the LeNet-5 classifier $\mathbf{g}(x; \theta) : \mathbb{R}^{28 \times 28} \to \mathbb{R}^{62}$ which is parameterized by the weights vector $\theta \in \mathbb{R}^{66126}$. We optimize (1) with cross-entropy loss and $\ell_2$ regularization such that $f_i$ is defined over the local dataset $\{(x_j^i, y_j^i)\}_{j=1}^{m_i}$ of (image, class) pairs by

$$f_i(\theta) = -\frac{1}{m_i} \sum_{j=1}^{m_i} \log \left( \frac{\exp([\mathbf{g}(x_j^i; \theta)]_{y_j^i})}{\sum_{k=1}^{62} \exp([\mathbf{g}(x_j^i; \theta)]_k)} \right) + \frac{\lambda}{2} \|\theta\|_2^2, \tag{33}$$

where $x_j^i \in \mathbb{R}^{28 \times 28}$, $y_j^i \in \{1, \dots, 62\}$.

Figure 3 compares the performance of benchmarked algorithms against the communication cost. We observe that `DoCoM` achieves similar performance as `CHOCO-SGD` and `BEER`, while demonstrating a slightly better performance in terms of the consensus gap. We speculate that the performance gap has narrowed due to the highly non-smooth nature of LeNet-5.

## 5   Conclusions

We have proposed the `DoCoM` algorithm for communication efficient decentralized learning and shown that the algorithm achieves a state-of-the-art $\mathcal{O}(\epsilon^{-3})$ sampling complexity. Future works include investigating the effect of reducing the frequency of (compressed) communication. For example, through considering asynchronous updates with possibly time varying or random graph.

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

## A    Missing Proofs from Section 3.2

Using the matrix notations defined in the preface of Section 3.2, we observe that DoCoM (9) can be expressed conveniently as

$$
\begin{array}{ll}
\Theta^{t+1} & = \Theta^t - \eta G^t + \gamma(\mathbf{W} - \mathbf{I})\widehat{\Theta}^{t+1} \\
\widehat{\Theta}^{t+1} & = \widehat{\Theta}^t + \mathcal{Q}(\Theta^t - \eta G^t - \widehat{\Theta}^t) \\
V^{t+1} & = \beta\nabla\widehat{F}^{t+1} + (1-\beta)(V^t + \nabla\widehat{F}^{t+1} - \nabla\widetilde{F}^t) \\
G^{t+1} & = G^t + V^{t+1} - V^t + \gamma(\mathbf{W} - \mathbf{I})\widehat{G}^{t+1} \\
\widehat{G}^{t+1} & = \widehat{G}^t + \mathcal{Q}(G^t + V^{t+1} - V^t - \widehat{G}^t)
\end{array}
\quad , \mathcal{Q}(X) = \begin{pmatrix} \mathcal{Q}(x_1)^\top \\ \vdots \\ \mathcal{Q}(x_n)^\top \end{pmatrix}, \ \forall \ X \in \mathbb{R}^{n \times d}.
$$

The above simplified expression of the algorithm will be useful for our subsequent analysis.

## A.1   Proof of Lemma 3.4

Using the $L$-smoothness of $f$ [cf. Assumption 2.1], we obtain:

$$
\begin{aligned}
f(\bar{\theta}^{t+1}) &\leq f(\bar{\theta}^t) + \left\langle \nabla f(\bar{\theta}^t) \mid \bar{\theta}^{t+1} - \bar{\theta}^t \right\rangle + \frac{L}{2} \left\| \bar{\theta}^{t+1} - \bar{\theta}^t \right\|^2 \\
&= f(\bar{\theta}^t) - \eta \left\langle \nabla f(\bar{\theta}^t) \mid \bar{g}^t \right\rangle + \frac{L\eta^2}{2} \left\| \bar{g}^t \right\|^2 \\
&= f(\bar{\theta}^t) - \frac{\eta}{2} \left( \left\| \bar{g}^t \right\|^2 + \left\| \nabla f(\bar{\theta}^t) \right\|^2 - \left\| \bar{g}^t - \nabla f(\bar{\theta}^t) \right\|^2 \right) + \frac{L\eta^2}{2} \left\| \bar{g}^t \right\|^2 \\
&\overset{(a)}{\leq} f(\bar{\theta}^t) - \frac{\eta}{4} \left\| \bar{g}^t \right\|^2 - \frac{\eta}{2} \left\| \nabla f(\bar{\theta}^t) \right\|^2 + \frac{\eta}{2} \left\| \bar{g}^t - \nabla f(\bar{\theta}^t) \right\|^2 \\
&\leq f(\bar{\theta}^t) - \frac{\eta}{4} \left\| \bar{g}^t \right\|^2 - \frac{\eta}{2} \left\| \nabla f(\bar{\theta}^t) \right\|^2 + \eta \left( \left\| \bar{g}^t - \overline{\nabla F}^t \right\|^2 + \left\| \overline{\nabla F}^t - \nabla f(\bar{\theta}^t) \right\|^2 \right) \\
&\leq f(\bar{\theta}^t) - \frac{\eta}{4} \left\| \bar{g}^t \right\|^2 - \frac{\eta}{2} \left\| \nabla f(\bar{\theta}^t) \right\|^2 + \eta \left\| \bar{g}^t - \overline{\nabla F}^t \right\|^2 + \frac{L^2 \eta}{n} \left\| \Theta^t - \bar{\Theta}^t \right\|_F^2 \quad (34)
\end{aligned}
$$

where (a) is due to $\eta \leq \frac{1}{2L}$. We remark that $\langle x \mid y \rangle = x^\top y$ denotes the inner product between the vectors $x, y$.

Note that by construction and the initialization $v_i^0 = g_i^0$, we have $\bar{g}^t = \bar{v}^t$ for any $t \geq 0$; see (9d). Applying the upper bound

$$
\left\| \Theta^t - \bar{\Theta}^t \right\|_F^2 = \left\| (\mathbf{I} - (1/n)\mathbf{1}\mathbf{1}^\top)\Theta^t \right\|_F^2 = \left\| \mathbf{U}\mathbf{U}^\top \Theta^t \right\|_F^2 \leq \left\| \Theta_o^t \right\|_F^2 \quad (35)
$$

leads to Lemma 3.4.

## A.2   Proof of Lemma 3.5

Observe that

$$
\begin{aligned}
\Theta_o^{t+1} &= \mathbf{U}^\top (\Theta^t - \eta G^t + \gamma(\mathbf{W} - \mathbf{I})\hat{\Theta}^{t+1}) \\
&= \mathbf{U}^\top \left\{ \Theta^t - \eta G^t + \gamma(\mathbf{W} - \mathbf{I}) \left[ \hat{\Theta}^t + \mathcal{Q}(\Theta^t - \eta G^t - \hat{\Theta}^t) - \Theta^t + \eta G^t + \Theta^t - \eta G^t \right] \right\} \\
&= \mathbf{U}^\top \left\{ [I + \gamma(\mathbf{W} - \mathbf{I})](\Theta^t - \eta G^t) + \gamma(\mathbf{W} - \mathbf{I}) \left[ \mathcal{Q}(\Theta^t - \eta G^t - \hat{\Theta}^t) - (\Theta^t - \eta G^t - \hat{\Theta}^t) \right] \right\} \quad (36)
\end{aligned}
$$

Notice that it holds $\mathbf{U}^\top (I + \gamma(\mathbf{W} - \mathbf{I})) = \mathbf{U}^\top (I + \gamma(\mathbf{W} - \mathbf{I}))\mathbf{U}\mathbf{U}^\top$ and $\left\| \mathbf{U}^\top (I + \gamma(\mathbf{W} - \mathbf{I}))\mathbf{U} \right\| \leq 1 - \rho\gamma$. Taking the Frobenius norm on (36) and the conditional expectation $\mathbb{E}_t[\cdot]$ on the randomness in DoCoM up to the $t$th iteration:

$$
\mathbb{E}_t[\left\| \Theta_o^{t+1} \right\|_F^2] \quad (37)
$$

$$
= \mathbb{E}_t \left[ \left\| \mathbf{U}^\top (I + \gamma(\mathbf{W} - \mathbf{I}))\mathbf{U}\mathbf{U}^\top (\Theta^t - \eta G^t) + \gamma \mathbf{U}^\top (\mathbf{W} - \mathbf{I}) \left[ \mathcal{Q}(\Theta^t - \eta G^t - \hat{\Theta}^t) - (\Theta^t - \eta G^t - \hat{\Theta}^t) \right] \right\|_F^2 \right]
$$

$$
\leq (1 + \alpha) \left\| \mathbf{U}^\top (I + \gamma(\mathbf{W} - \mathbf{I}))\mathbf{U}(\Theta_o^t - \eta G_o^t) \right\|_F^2
$$

$$
+ (1 + \alpha^{-1})\mathbb{E}_t \left[ \left\| \gamma \mathbf{U}^\top (\mathbf{W} - \mathbf{I}) \left[ \mathcal{Q}(\Theta^t - \eta G^t - \hat{\Theta}^t) - (\Theta^t - \eta G^t - \hat{\Theta}^t) \right] \right\|_F^2 \right]
$$

$$
\leq (1 + \alpha) \left\| \mathbf{U}^\top (I + \gamma(\mathbf{W} - \mathbf{I}))\mathbf{U} \right\|^2 \left\| \Theta_o^t - \eta G_o^t \right\|_F^2
$$

$$
+ (1 + \alpha^{-1})\gamma^2 \left\| \mathbf{U}^\top (\mathbf{W} - \mathbf{I}) \right\|^2 \mathbb{E}_t \left[ \left\| \mathcal{Q}(\Theta^t - \eta G^t - \hat{\Theta}^t) - (\Theta^t - \eta G^t - \hat{\Theta}^t) \right\|_F^2 \right]
$$

$$
\overset{(a)}{\leq} (1 + \alpha)(1 - \rho\gamma)^2 \left\| \Theta_o^t - \eta G_o^t \right\|_F^2 + (1 + \alpha^{-1})\bar{\omega}^2 \gamma^2 (1 - \delta) \left\| \Theta^t - \eta G^t - \hat{\Theta}^t \right\|_F^2
$$

$$
\leq (1 + \alpha)(1 - \rho\gamma)^2 (1 + \beta) \left\| \Theta_o^t \right\|_F^2 + (1 + \alpha)(1 - \rho\gamma)^2 (1 + \beta^{-1})\eta^2 \left\| G_o^t \right\|_F^2
$$

$$+ \bar{\omega}^2 \gamma^2 (1 + \alpha^{-1})(1 - \delta) \left\| \Theta^t - \eta G^t - \hat{\Theta}^t \right\|_F^2$$

$$\overset{(b)}{\leq} (1 - \frac{\rho\gamma}{2}) \left\| \Theta_o^t \right\|_F^2 + \frac{2}{\rho\gamma} \eta^2 \left\| G_o^t \right\|_F^2 + \frac{\bar{\omega}^2 \gamma}{\rho} \left\| \Theta^t - \eta G^t - \hat{\Theta}^t \right\|_F^2$$

where (a) is due to $\|\mathbf{W} - \mathbf{I}\| \leq \bar{\omega}$ and (b) is due to the choices $\alpha = \frac{\rho\gamma}{1-\rho\gamma}, \beta = \frac{\rho\gamma}{2}$. The proof is completed.

### A.3 Proof of Lemma 3.6

Defining $\overline{\nabla} \widehat{F}^t = n^{-1} \mathbf{1}^\top \nabla \widehat{F}^t, \overline{\nabla} \widetilde{F}^t = n^{-1} \mathbf{1}^\top \nabla \widetilde{F}^t$, we get

$$\bar{v}^{t+1} - \overline{\nabla F}^{t+1} = \overline{\nabla} \widehat{F}^{t+1} + (1 - \beta)(\bar{v}^t - \overline{\nabla} \widetilde{F}^t) - \overline{\nabla F}^{t+1} \tag{38}$$
$$= (1 - \beta)(\bar{v}^t - \overline{\nabla F}^t) - \beta(\overline{\nabla F}^{t+1} - \overline{\nabla} \widehat{F}^{t+1})$$
$$+ (1 - \beta)(\overline{\nabla F}^t - \overline{\nabla} \widetilde{F}^t - (\overline{\nabla F}^{t+1} - \overline{\nabla} \widehat{F}^{t+1}))$$

It follows that

$$\mathbb{E}\left[ \left\| \bar{v}^{t+1} - \overline{\nabla F}^{t+1} \right\|^2 \right] \leq (1 - \beta)^2 \mathbb{E}\left[ \left\| \bar{v}^t - \overline{\nabla F}^t \right\|^2 \right] + 2\beta^2 \mathbb{E}\left[ \left\| \overline{\nabla F}^{t+1} - \overline{\nabla} \widehat{F}^{t+1} \right\|^2 \right]$$
$$+ 2(1 - \beta)^2 \mathbb{E}\left[ \left\| \overline{\nabla F}^t - \overline{\nabla} \widetilde{F}^t - (\overline{\nabla F}^{t+1} - \overline{\nabla} \widehat{F}^{t+1}) \right\|^2 \right]$$
$$\leq (1 - \beta)^2 \mathbb{E}\left[ \left\| \bar{v}^t - \overline{\nabla F}^t \right\|^2 \right] + 2\beta^2 \frac{\sigma^2}{n} + 2(1 - \beta)^2 \frac{L^2}{n^2} \mathbb{E}\left[ \left\| \Theta^{t+1} - \Theta^t \right\|^2 \right]$$

Furthermore, applying Lemma A.1 leads to

$$\mathbb{E}\left[ \left\| \bar{v}^{t+1} - \overline{\nabla F}^{t+1} \right\|^2 \right] \leq (1 - \beta)^2 \mathbb{E}\left[ \left\| \bar{v}^t - \overline{\nabla F}^t \right\|^2 \right] + 2\beta^2 \frac{\sigma^2}{n}$$
$$+ 8(1 - \beta)^2 \frac{L^2}{n^2} \eta^2 (1 - \rho\gamma)^2 \mathbb{E}\left[ \left\| G_o^t \right\|_F^2 \right]$$
$$+ 4n(1 - \beta)^2 \frac{L^2}{n^2} \eta^2 (1 - \rho\gamma)^2 \mathbb{E}\left[ \left\| \bar{g}^t \right\|^2 \right] \tag{39}$$
$$+ 8(1 - \beta)^2 \frac{L^2}{n^2} \bar{\omega}^2 \gamma^2 \mathbb{E}\left[ \left\| \Theta_o^t \right\|_F^2 \right]$$
$$+ 4(1 - \beta)^2 \frac{L^2}{n^2} \bar{\omega}^2 \gamma^2 (1 - \delta) \mathbb{E}\left[ \left\| \Theta^t - \eta G^t - \hat{\Theta}^t \right\|_F^2 \right]$$

This concludes our proof for the stated lemma.

**Bound on the Matrix Form**  Observe that

$$V^{t+1} - \nabla F^{t+1} = \nabla \widehat{F}^{t+1} + (1 - \beta)(V^t - \nabla \widetilde{F}^t) - \nabla F^{t+1} \tag{40}$$
$$= (1 - \beta)(V^t - \nabla F^t) - \beta(\nabla F^{t+1} - \nabla \widehat{F}^{t+1})$$
$$+ (1 - \beta)(\nabla F^t - \nabla \widetilde{F}^t - (\nabla F^{t+1} - \nabla \widehat{F}^{t+1}))$$

Taking the full expectation yields

$$\mathbb{E}\left[ \left\| V^{t+1} - \nabla F^{t+1} \right\|_F^2 \right] \leq (1 - \beta)^2 \mathbb{E}\left[ \left\| V^t - \nabla F^t \right\|_F^2 \right] + 2\beta^2 n \sigma^2 + 2(1 - \beta)^2 L^2 \mathbb{E}\left[ \left\| \Theta^{t+1} - \Theta^t \right\|_F^2 \right]$$

Again, applying Lemma A.1 leads to

$$
\begin{aligned}
\mathbb{E}\left[\left\|V^{t+1} - \nabla F^{t+1}\right\|_F^2\right] \leq {} & (1-\beta)^2 \mathbb{E}\left[\left\|V^t - \nabla F^t\right\|_F^2\right] + 2\beta^2 n\sigma^2 \\
& + 8(1-\beta)^2 L^2 \eta^2 (1-\rho\gamma)^2 \mathbb{E}\left[\left\|G_o^t\right\|_F^2\right] \\
& + 4n(1-\beta)^2 L^2 \eta^2 (1-\rho\gamma)^2 \mathbb{E}\left[\left\|\bar{g}^t\right\|^2\right] \\
& + 8(1-\beta)^2 L^2 \bar{\omega}^2 \gamma^2 \mathbb{E}\left[\left\|\Theta_o^t\right\|_F^2\right] \\
& + 4(1-\beta)^2 L^2 \bar{\omega}^2 \gamma^2 (1-\delta) \mathbb{E}\left[\left\|\Theta^t - \eta G^t - \hat{\Theta}^t\right\|_F^2\right]
\end{aligned}
\tag{41}
$$

This concludes our proof.

## A.4   Proof of Lemma 3.7

We begin by observing the update for $G_o^{t+1}$ as:

$$
\begin{aligned}
G_o^{t+1} &= \mathbf{U}^\top[G^t + V^{t+1} - V^t + \gamma(\mathbf{W} - \mathbf{I})\hat{G}^{t+1}] \\
&= \mathbf{U}^\top\left[G^t + V^{t+1} - V^t + \gamma(\mathbf{W} - \mathbf{I})\left(\hat{G}^t + \mathcal{Q}(G^t + V^{t+1} - V^t - \hat{G}^t)\right)\right] \\
&= \mathbf{U}^\top\left[(I + \gamma(\mathbf{W} - \mathbf{I}))(G^t + V^{t+1} - V^t)\right] \\
&\quad + \gamma\mathbf{U}^\top(\mathbf{W} - \mathbf{I})\left[\mathcal{Q}(G^t + V^{t+1} - V^t - \hat{G}^t) - (G^t + V^{t+1} - V^t - \hat{G}^t)\right]
\end{aligned}
$$

The above implies that

$$
\begin{aligned}
\mathbb{E}_t\left[\left\|G_o^{t+1}\right\|_F^2\right] \leq {} & (1+\alpha_0)(1-\rho\gamma)^2 \mathbb{E}_t\left[\left\|G_o^t + \mathbf{U}^\top(V^{t+1} - V^t)\right\|_F^2\right] \\
& + (1+\alpha_0^{-1})\gamma^2\bar{\omega}^2(1-\delta)\mathbb{E}_t\left[\left\|G^t + V^{t+1} - V^t - \hat{G}^t\right\|_F^2\right] \\
\leq {} & (1+\alpha_0)(1-\rho\gamma)^2 \mathbb{E}_t\left[(1+\alpha_1)\left\|G_o^t\right\|_F^2 + (1+\alpha_1^{-1})\left\|V^{t+1} - V^t\right\|_F^2\right] \\
& + 2(1+\alpha_0^{-1})\gamma^2\bar{\omega}^2(1-\delta)\mathbb{E}_t\left[\left\|G^t - \hat{G}^t\right\|_F^2 + \left\|V^{t+1} - V^t\right\|_F^2\right]
\end{aligned}
$$

Taking $\alpha_0 = \frac{\rho\gamma}{1-\rho\gamma}$, $\alpha_1 = \frac{\rho\gamma}{2}$ gives

$$
\begin{aligned}
\mathbb{E}_t\left[\left\|G_o^{t+1}\right\|_F^2\right] \leq {} & \left(1 - \frac{\rho\gamma}{2}\right)\left\|G_o^t\right\|_F^2 + \gamma\frac{2\bar{\omega}^2}{\rho}\left\|G^t - \hat{G}^t\right\|_F^2 \\
& + \frac{2}{\rho\gamma}\left(1 + \gamma^2\bar{\omega}^2\right)\mathbb{E}_t\left[\left\|V^{t+1} - V^t\right\|_F^2\right]
\end{aligned}
$$

Taking the full expectation and applying Lemma A.2 give

$$
\begin{aligned}
\mathbb{E}\left[\left\|G_o^{t+1}\right\|_F^2\right] \leq {} & \left(1 - \frac{\rho\gamma}{2}\right)\mathbb{E}\left[\left\|G_o^t\right\|_F^2\right] + \gamma\frac{2\bar{\omega}^2}{\rho}\mathbb{E}\left[\left\|G^t - \hat{G}^t\right\|_F^2\right] \\
& + \frac{2}{\rho\gamma}\left(1 + \gamma^2\bar{\omega}^2\right)\left(3L^2\mathbb{E}\left[\left\|\Theta^{t+1} - \Theta^t\right\|_F^2\right] + 3\beta^2\mathbb{E}\left[\left\|V^t - \nabla F^t\right\|_F^2\right] + 3n\beta^2\sigma^2\right)
\end{aligned}
$$

Furthermore, applying Lemma A.1 yields

$$
\begin{aligned}
\mathbb{E}\left[\left\|G_o^{t+1}\right\|_F^2\right] \leq {} & \left(1 - \frac{\rho\gamma}{2} + \eta^2\frac{24L^2(1+\gamma^2\bar{\omega}^2)(1-\rho\gamma)^2}{\rho\gamma}\right)\mathbb{E}\left[\left\|G_o^t\right\|_F^2\right] + \gamma\frac{2\bar{\omega}^2}{\rho}\mathbb{E}\left[\left\|G^t - \hat{G}^t\right\|_F^2\right] \\
& + \frac{24L^2(1+\gamma^2\bar{\omega}^2)}{\rho\gamma}\bar{\omega}^2\gamma^2\mathbb{E}\left[\left\|\Theta_o^t\right\|_F^2\right]
\end{aligned}
$$

$$+ \frac{12L^2(1+\gamma^2\bar{\omega}^2)}{\rho\gamma}\bar{\omega}^2\gamma^2(1-\delta)\mathbb{E}\left[\left\|\Theta^t - \eta G^t - \hat{\Theta}^t\right\|_F^2\right]$$

$$+ \frac{12L^2(1+\gamma^2\bar{\omega}^2)}{\rho\gamma}(1-\rho\gamma)^2 n\,\eta^2\,\mathbb{E}\left[\left\|\bar{g}^t\right\|^2\right]$$

$$+ \frac{6(1+\gamma^2\bar{\omega}^2)}{\rho\gamma}\beta^2\mathbb{E}\left[\left\|V^t - \nabla F^t\right\|_F^2\right]$$

$$+ \frac{6}{\rho\gamma}\left(1+\gamma^2\bar{\omega}^2\right)\beta^2 n\sigma^2$$

The step size condition

$$\eta \leq \frac{\rho\gamma}{10L(1-\rho\gamma)\sqrt{1+\gamma^2\bar{\omega}^2}}, \quad \gamma \leq \frac{1}{8\bar{\omega}} \tag{42}$$

implies that

$$\mathbb{E}\left[\left\|G_o^{t+1}\right\|_F^2\right] \leq \left(1 - \frac{\rho\gamma}{4}\right)\mathbb{E}\left[\left\|G_o^t\right\|_F^2\right] + \gamma\frac{2\bar{\omega}^2}{\rho}\mathbb{E}\left[\left\|G^t - \hat{G}^t\right\|_F^2\right] + \gamma\frac{25L^2\bar{\omega}^2}{\rho}\mathbb{E}\left[\left\|\Theta_o^t\right\|_F^2\right]$$

$$+ \gamma\frac{13L^2}{\rho}\bar{\omega}^2(1-\delta)\mathbb{E}\left[\left\|\Theta^t - \eta G^t - \hat{\Theta}^t\right\|_F^2\right] + \gamma\frac{\rho n}{5}\mathbb{E}\left[\left\|\bar{g}^t\right\|^2\right]$$

$$+ \frac{7}{\rho\gamma}\beta^2\mathbb{E}\left[\left\|V^t - \nabla F^t\right\|_F^2\right] + \frac{7n}{\rho\gamma}\beta^2\sigma^2.$$

This concludes our proof.

### A.5 Proof of Lemma 3.8

Observe that

$$\mathbb{E}_t\left[\left\|\Theta^{t+1} - \eta G^{t+1} - \hat{\Theta}^{t+1}\right\|_F^2\right] = \mathbb{E}_t\left[\left\|\Theta^{t+1} - \eta G^{t+1} - (\hat{\Theta}^t + \mathcal{Q}(\Theta^t - \eta G^t - \hat{\Theta}^t))\right\|_F^2\right]$$

$$= \mathbb{E}_t\left[\left\|\Theta^{t+1} - \eta G^{t+1} - (\Theta^t - \eta G^t) + (\Theta^t - \eta G^t - \hat{\Theta}^t) - \mathcal{Q}(\Theta^t - \eta G^t - \hat{\Theta}^t)\right\|_F^2\right]$$

$$\leq (1 + \frac{2}{\delta})\mathbb{E}_t\left[\left\|\Theta^{t+1} - \Theta^t - \eta(G^{t+1} - G^t)\right\|_F^2\right] + (1 + \frac{\delta}{2})(1-\delta)\left\|\Theta^t - \eta G^t - \hat{\Theta}^t\right\|_F^2$$

$$\leq 2(1 + \frac{2}{\delta})\mathbb{E}_t\left[\left\|\Theta^{t+1} - \Theta^t\right\|_F^2\right] + 2\eta^2(1 + \frac{2}{\delta})\mathbb{E}_t\left[\left\|G^{t+1} - G^t\right\|_F^2\right] + (1 - \frac{\delta}{2})\left\|\Theta^t - \eta G^t - \hat{\Theta}^t\right\|_F^2 \tag{43}$$

Note that as

$$G^{t+1} - G^t = G^{t+1} - \mathbf{1}(\bar{g}^{t+1})^\top + \mathbf{1}(\bar{g}^{t+1})^\top - \left[G^t - \mathbf{1}(\bar{g}^t)^\top + \mathbf{1}(\bar{g}^t)^\top\right]$$

$$= \mathbf{U}G_o^{t+1} - \mathbf{U}G_o^t + \mathbf{1}(\bar{g}^{t+1} - \bar{g}^t)^\top,$$

we obtain the bound

$$\mathbb{E}\left[\left\|G^{t+1} - G^t\right\|_F^2\right] \leq \frac{1}{n}\mathbb{E}\left[\left\|\mathbf{1}^\top(V^{t+1} - V^t)\right\|^2\right] + 2\mathbb{E}\left[\left\|G_o^{t+1}\right\|_F^2\right] + 2\mathbb{E}\left[\left\|G_o^t\right\|_F^2\right]$$

With Lemma A.2, we substitute back into (43) and obtain

$$\mathbb{E}\left[\left\|\Theta^{t+1} - \eta G^{t+1} - \hat{\Theta}^{t+1}\right\|_F^2\right] \leq (1 - \frac{\delta}{2})\mathbb{E}\left[\left\|\Theta^t - \eta G^t - \hat{\Theta}^t\right\|_F^2\right] + 4\eta^2(1 + \frac{2}{\delta})\mathbb{E}\left[\left\|G_o^{t+1}\right\|_F^2 + \left\|G_o^t\right\|_F^2\right]$$

$$+ 2(1 + \frac{2}{\delta})(3\eta^2 L^2 + 1)\mathbb{E}\left[\left\|\Theta^{t+1} - \Theta^t\right\|_F^2\right] +$$

$$6\beta^2\eta^2(1 + \frac{2}{\delta})\mathbb{E}\left[\left\|V^t - \nabla F^t\right\|^2\right] + 6\beta^2\eta^2(1 + \frac{2}{\delta})n\sigma^2 \tag{44}$$

We further apply Lemma A.1 to obtain

$$
\begin{aligned}
\mathbb{E}\left[\left\|\Theta^{t+1} - \eta G^{t+1} - \hat{\Theta}^{t+1}\right\|_F^2\right] &\leq (1 - \frac{\delta}{2})\mathbb{E}\left[\left\|\Theta^t - \eta G^t - \hat{\Theta}^t\right\|_F^2\right] + 4\eta^2(1 + \frac{2}{\delta})\mathbb{E}\left[\left\|G_o^{t+1}\right\|_F^2 + \left\|G_o^t\right\|_F^2\right] \\
&+ 6\beta^2\eta^2(1 + \frac{2}{\delta})\mathbb{E}\left[\left\|V^t - \nabla F^t\right\|^2\right] + 6\beta^2\eta^2(1 + \frac{2}{\delta})n\sigma^2 + 8(1 + \frac{2}{\delta})(3\eta^2 L^2 + 1)\eta^2(1 - \rho\gamma)^2\mathbb{E}\left[\left\|G_o^t\right\|_F^2\right] \\
&+ 8(1 + \frac{2}{\delta})(3\eta^2 L^2 + 1)\bar{\omega}^2\gamma^2\mathbb{E}\left[\left\|\Theta_o^t\right\|_F^2\right] + 4(1 + \frac{2}{\delta})(3\eta^2 L^2 + 1)n\eta^2(1 - \rho\gamma)^2\mathbb{E}\left[\left\|\bar{g}^t\right\|^2\right] \\
&+ 4(1 + \frac{2}{\delta})(3\eta^2 L^2 + 1)\bar{\omega}^2\gamma^2(1 - \delta)\mathbb{E}\left[\left\|\Theta^t - \eta G^t - \hat{\Theta}^t\right\|_F^2\right]
\end{aligned}
\tag{45}
$$

Using the step size condition:
$$
\gamma^2 \leq \frac{\delta}{16\bar{\omega}^2(1 - \delta)(1 + 3\eta^2 L^2)(1 + 2/\delta)}
$$
and we recall that $\eta \leq 1/(4L)$, the upper bound in (45) can be simplified as

$$
\begin{aligned}
\mathbb{E}\left[\left\|\Theta^{t+1} - \eta G^{t+1} - \hat{\Theta}^{t+1}\right\|_F^2\right] &\leq (1 - \frac{\delta}{4})\mathbb{E}\left[\left\|\Theta^t - \eta G^t - \hat{\Theta}^t\right\|_F^2\right] + \frac{12}{\delta}\eta^2\mathbb{E}\left[\left\|G_o^{t+1}\right\|_F^2\right] + \frac{18}{\delta}\beta^2\eta^2\mathbb{E}\left[\left\|V^t - \nabla F^t\right\|^2\right] \\
&+ \frac{18}{\delta}\beta^2\eta^2 n\sigma^2 + \frac{41}{\delta}\eta^2\mathbb{E}\left[\left\|G_o^t\right\|_F^2\right] + \frac{29}{\delta}\bar{\omega}^2\gamma^2\mathbb{E}\left[\left\|\Theta_o^t\right\|_F^2\right] + \frac{15}{\delta}\eta^2 n\,\mathbb{E}\left[\left\|\bar{g}^t\right\|^2\right]
\end{aligned}
$$

The above bound can be combined with Lemma 3.7 and $\gamma\rho \leq 1$ to give

$$
\begin{aligned}
\mathbb{E}\left[\left\|\Theta^{t+1} - \eta G^{t+1} - \hat{\Theta}^{t+1}\right\|_F^2\right] &\leq \left(1 - \frac{\delta}{4} + \eta^2\gamma\frac{156\bar{\omega}^2 L^2(1 - \delta)}{\rho\delta}\right)\mathbb{E}\left[\left\|\Theta^t - \eta G^t - \hat{\Theta}^t\right\|_F^2\right] \\
&+ \eta^2\frac{50}{\delta}\mathbb{E}\left[\left\|G_o^t\right\|_F^2\right] + \eta^2\gamma\frac{24\bar{\omega}^2}{\delta\rho}\mathbb{E}\left[\left\|G^t - \hat{G}^t\right\|_F^2\right] \\
&+ \left[\frac{29}{\delta}\bar{\omega}^2\gamma^2 + \frac{300 L^2\bar{\omega}^2\eta^2\gamma}{\rho\delta}\right]\mathbb{E}\left[\left\|\Theta_o^t\right\|_F^2\right] + \frac{18\eta^2}{\delta}n\,\mathbb{E}\left[\left\|\bar{g}^t\right\|^2\right] \\
&+ \left(18 + \frac{84}{\rho\gamma}\right)\frac{\beta^2\eta^2}{\delta}\mathbb{E}\left[\left\|V^t - \nabla F^t\right\|_F^2\right] + \left(18 + \frac{84}{\rho\gamma}\right)\frac{\beta^2\eta^2 n\sigma^2}{\delta}
\end{aligned}
$$

Taking $\eta^2\gamma \leq \frac{\delta^2\rho}{1248(1 - \delta)\bar{\omega}^2 L^2}$ and $\gamma \leq \frac{\delta}{8\bar{\omega}}$ simplifies the bound into

$$
\begin{aligned}
\mathbb{E}\left[\left\|\Theta^{t+1} - \eta G^{t+1} - \hat{\Theta}^{t+1}\right\|_F^2\right] &\leq \left(1 - \frac{\delta}{8}\right)\mathbb{E}\left[\left\|\Theta^t - \eta G^t - \hat{\Theta}^t\right\|_F^2\right] \\
&+ \eta^2\frac{50}{\delta}\mathbb{E}\left[\left\|G_o^t\right\|_F^2\right] + \eta^2\frac{3\bar{\omega}}{\rho}\mathbb{E}\left[\left\|G^t - \hat{G}^t\right\|_F^2\right] \\
&+ \left[\frac{29}{\delta}\bar{\omega}^2\gamma^2 + \frac{38 L^2\bar{\omega}\eta^2}{\rho}\right]\mathbb{E}\left[\left\|\Theta_o^t\right\|_F^2\right] + \frac{18\eta^2}{\delta}n\,\mathbb{E}\left[\left\|\bar{g}^t\right\|^2\right] \\
&+ \left(18 + \frac{84}{\rho\gamma}\right)\frac{\beta^2\eta^2}{\delta}\mathbb{E}\left[\left\|V^t - \nabla F^t\right\|_F^2\right] + \left(18 + \frac{84}{\rho\gamma}\right)\frac{\beta^2\eta^2 n\sigma^2}{\delta}
\end{aligned}
$$

This concludes the proof.

## A.6 Proof of Lemma 3.9

We begin by observing the following recursion for $G^t - \hat{G}^t$:

$$
\begin{aligned}
G^{t+1} - \hat{G}^{t+1} &= G^t + V^{t+1} - V^t + (\gamma(\mathbf{W} - \mathbf{I}) - I)\hat{G}^{t+1} \\
&= \gamma(\mathbf{W} - \mathbf{I})(G^t + V^{t+1} - V^t) \\
&+ (\gamma(\mathbf{W} - \mathbf{I}) - I)\left[\mathcal{Q}(G^t + V^{t+1} - V^t - \hat{G}^t) - (G^t + V^{t+1} - V^t - \hat{G}^t)\right].
\end{aligned}
$$

This implies

$$\mathbb{E}\left[\left\|G^{t+1} - \hat{G}^{t+1}\right\|_F^2\right] \le (1+\alpha_0)(1+\gamma\bar{\omega})^2(1-\delta)\mathbb{E}\left[(1+\alpha_1)\left\|G^t - \hat{G}^t\right\|_F^2 + (1+\alpha_1^{-1})\left\|V^{t+1} - V^t\right\|_F^2\right]$$
$$+ 2(1+\alpha_0^{-1})\gamma^2\bar{\omega}^2\mathbb{E}\left[\left\|G_o^t\right\|_F^2 + \left\|V^{t+1} - V^t\right\|_F^2\right]$$

Taking $\alpha_0 = \frac{\delta}{4}$, $\alpha_1 = \frac{\delta}{8}$ and the step size condition

$$\gamma \le \frac{\delta}{8\bar{\omega}} \le \frac{\delta}{6\bar{\omega}}$$

give

$$\mathbb{E}\left[\left\|G^{t+1} - \hat{G}^{t+1}\right\|_F^2\right] \le \left(1 - \frac{\delta}{8}\right)\mathbb{E}\left[\left\|G^t - \hat{G}^t\right\|_F^2\right] + \frac{10\gamma^2\bar{\omega}^2}{\delta}\mathbb{E}\left[\left\|G_o^t\right\|_F^2\right]$$
$$+ \left((1 - \frac{\delta}{4})(1 + \frac{8}{\delta}) + 2\gamma^2\bar{\omega}^2(1 + \frac{4}{\delta})\right)\mathbb{E}\left[\left\|V^{t+1} - V^t\right\|_F^2\right]$$
$$\le \left(1 - \frac{\delta}{8}\right)\mathbb{E}\left[\left\|G^t - \hat{G}^t\right\|_F^2\right] + \frac{10\gamma^2\bar{\omega}^2}{\delta}\mathbb{E}\left[\left\|G_o^t\right\|_F^2\right] + \frac{10(1 + \gamma^2\bar{\omega}^2)}{\delta}\mathbb{E}\left[\left\|V^{t+1} - V^t\right\|_F^2\right]$$

Applying Lemma A.2 and Lemma A.1 gives

$$\mathbb{E}\left[\left\|G^{t+1} - \hat{G}^{t+1}\right\|_F^2\right] \le \left(1 - \frac{\delta}{8}\right)\mathbb{E}\left[\left\|G^t - \hat{G}^t\right\|_F^2\right] + \frac{10\gamma^2\bar{\omega}^2}{\delta}\mathbb{E}\left[\left\|G_o^t\right\|_F^2\right] + \frac{31}{\delta}\beta^2\left(\mathbb{E}\left[\left\|V^t - \nabla F^t\right\|_F^2\right] + n\sigma^2\right)$$
$$+ \frac{30(1 + \gamma^2\bar{\omega}^2)L^2}{\delta}\mathbb{E}\left[\left\|\Theta^{t+1} - \Theta^t\right\|_F^2\right]$$
$$\le \left(1 - \frac{\delta}{8}\right)\mathbb{E}\left[\left\|G^t - \hat{G}^t\right\|_F^2\right] + \frac{10}{\delta}\left(\gamma^2\bar{\omega}^2 + 12\eta^2L^2(1 + \gamma^2\bar{\omega}^2)(1 - \rho\gamma)^2\right)\mathbb{E}\left[\left\|G_o^t\right\|_F^2\right]$$
$$+ \frac{120(1 + \gamma^2\bar{\omega}^2)L^2}{\delta}\bar{\omega}^2\gamma^2\mathbb{E}\left[\left\|\Theta_o^t\right\|_F^2\right]$$
$$+ \frac{60(1 + \gamma^2\bar{\omega}^2)L^2}{\delta}\bar{\omega}^2\gamma^2(1 - \delta)\mathbb{E}\left[\left\|\Theta^t - \eta G^t - \hat{\Theta}^t\right\|_F^2\right]$$
$$+ \frac{60(1 + \gamma^2\bar{\omega}^2)L^2}{\delta}n\eta^2(1 - \rho\gamma)^2\mathbb{E}\left[\left\|\bar{g}^t\right\|^2\right]$$
$$+ \frac{31}{\delta}\beta^2\left(\mathbb{E}\left[\left\|V^t - \nabla F^t\right\|_F^2\right] + n\sigma^2\right)$$

Using the step size condition from (42), i.e., $\eta^2L^2(1 - \rho\gamma)^2(1 + \gamma^2\bar{\omega}^2) \le \frac{\rho^2\gamma^2}{100}$, and $\gamma \le \frac{\delta}{8\bar{\omega}}$ simplifies the above to

$$\mathbb{E}\left[\left\|G^{t+1} - \hat{G}^{t+1}\right\|_F^2\right] \le \left(1 - \frac{\delta}{8}\right)\mathbb{E}\left[\left\|G^t - \hat{G}^t\right\|_F^2\right] + \frac{10}{\delta}\gamma^2\left(\bar{\omega}^2 + \frac{\rho^2}{8}\right)\mathbb{E}\left[\left\|G_o^t\right\|_F^2\right]$$
$$+ \gamma^2\frac{122L^2\bar{\omega}^2}{\delta}\mathbb{E}\left[\left\|\Theta_o^t\right\|_F^2\right] + \gamma^2\frac{60L^2\bar{\omega}^2}{\delta}\mathbb{E}\left[\left\|\Theta^t - \eta G^t - \hat{\Theta}^t\right\|_F^2\right]$$
$$+ \gamma^2\frac{3\rho^2}{5\delta}n\mathbb{E}\left[\left\|\bar{g}^t\right\|^2\right] + \frac{31}{\delta}\beta^2\mathbb{E}\left[\left\|V^t - \nabla F^t\right\|_F^2\right] + \frac{31}{\delta}\beta^2n\sigma^2.$$

This concludes our proof.

## A.7 Proof of Lemma 3.10

Below, we illustrate how to find a set of tight conditions for the free parameters $a, b, c > 0$. Combining Lemma 3.5, 3.6, 3.7, 3.9, 3.8 and (41) yields

$$\mathbb{E}\left[L^2\left\|\Theta_o^{t+1}\right\|_F^2 + n\left\|\bar{v}^{t+1} - \overline{\nabla F}^{t+1}\right\|^2 + \tfrac{1}{n}\left\|V^{t+1} - \nabla F^{t+1}\right\|_F^2 + a\left\|G_o^{t+1}\right\|_F^2 + b\left\|G^{t+1} - \hat{G}^{t+1}\right\|_F^2 + c\left\|\Theta^{t+1} - \eta G^{t+1} - \hat{\Theta}^{t+1}\right\|_F^2\right]$$

$$\leq \left(1 - \frac{\rho\gamma}{2} + \frac{16}{n}(1-\beta)^2\bar{\omega}^2\gamma^2 + a\gamma\frac{25\bar{\omega}^2}{\rho} + b\gamma^2\frac{122\bar{\omega}^2}{\delta} + c(\gamma^2\frac{29\bar{\omega}^2}{\delta L^2} + \eta^2\frac{38\bar{\omega}}{\rho})\right)\mathbb{E}\left[L^2\left\|\Theta_o^t\right\|_F^2\right]$$

$$+ (1-\beta)^2 n\mathbb{E}\left[\left\|\bar{v}^t - \overline{\nabla F}^t\right\|^2\right] + \left((1-\beta)^2 + a\beta^2\frac{7n}{\rho\gamma} + b\beta^2\frac{31n}{\delta} + c\beta^2\eta^2(18 + \frac{84}{\rho\gamma})\frac{n}{\delta}\right)\frac{1}{n}\mathbb{E}\left[\left\|V^t - \nabla F^t\right\|_F^2\right]$$

$$+ a\left(1 - \frac{\rho\gamma}{4} + \frac{1}{a}\eta^2\frac{2L^2}{\rho\gamma} + \frac{1}{a}\frac{16}{n}\eta^2(1-\beta)^2(1-\rho\gamma)^2L^2 + \frac{b}{a}\gamma^2\frac{10}{\delta}(\bar{\omega}^2 + \frac{\rho^2}{8}) + \frac{c}{a}\eta^2\frac{50}{\delta}\right)\mathbb{E}\left[\left\|G_o^t\right\|_F^2\right]$$

$$+ b\left(1 - \frac{\delta}{8} + \frac{a}{b}\gamma\frac{2\bar{\omega}^2}{\rho} + \frac{c}{b}\eta^2\frac{3\bar{\omega}}{\rho}\right)\mathbb{E}\left[\left\|G^t - \hat{G}^t\right\|_F^2\right]$$

$$+ c\left(1 - \frac{\delta}{8} + \frac{1}{c}\gamma\frac{L^2\bar{\omega}^2}{\rho} + \frac{1}{c}\frac{8}{n}\gamma^2(1-\beta)^2L^2\bar{\omega}^2(1-\delta) + \frac{a}{c}\gamma\frac{13L^2}{\rho}\bar{\omega}^2(1-\delta) + \frac{b}{c}\gamma^2\frac{60L^2\bar{\omega}^2}{\delta}\right)\mathbb{E}\left[\left\|\Theta^t - \eta G^t - \hat{\Theta}^t\right\|_F^2\right]$$

$$+ \left(\frac{4}{n}\beta^2 + a\beta^2\frac{7}{\rho\gamma} + b\beta^2\frac{31}{\delta} + c\beta^2\eta^2(18 + \frac{84}{\rho\gamma})\frac{1}{\delta}\right)n\sigma^2$$

$$+ \left(\frac{8}{n}(1-\beta)^2L^2\eta^2(1-\rho\gamma)^2 + a\gamma\frac{\rho}{5} + b\gamma^2\frac{3\rho^2}{5\delta} + c\eta^2\frac{18}{\delta}\right)n\,\mathbb{E}\left[\left\|\bar{g}^t\right\|^2\right]$$

Our goal is to find conditions on step sizes and the choices of $a, b, c$ such that

$$1 - \frac{\rho\gamma}{2} + \frac{16}{n}(1-\beta)^2\bar{\omega}^2\gamma^2 + a\gamma\frac{25\bar{\omega}^2}{\rho} + b\gamma^2\frac{122\bar{\omega}^2}{\delta} + c(\gamma^2\frac{29\bar{\omega}^2}{\delta L^2} + \eta^2\frac{38\bar{\omega}}{\rho}) \leq 1 - \frac{\rho\gamma}{8}$$

$$(1-\beta)^2 + a\beta^2\frac{7n}{\rho\gamma} + b\beta^2\frac{31n}{\delta} + c\beta^2\eta^2(18 + \frac{84}{\rho\gamma})\frac{n}{\delta} \leq (1-\beta)$$

$$1 - \frac{\rho\gamma}{4} + \frac{1}{a}\eta^2\frac{2L^2}{\rho\gamma} + \frac{1}{a}\frac{16}{n}\eta^2(1-\beta)^2(1-\rho\gamma)^2L^2 + \frac{b}{a}\gamma^2\frac{10}{\delta}(\bar{\omega}^2 + \frac{\rho^2}{8}) + \frac{c}{a}\eta^2\frac{50}{\delta} \leq 1 - \frac{\rho\gamma}{8}$$

$$1 - \frac{\delta}{8} + \frac{a}{b}\gamma\frac{2\bar{\omega}^2}{\rho} + \frac{c}{b}\eta^2\frac{3\bar{\omega}}{\rho} \leq 1 - \frac{\delta\gamma}{8}$$

$$1 - \frac{\delta}{8} + \frac{1}{c}\gamma\frac{L^2\bar{\omega}^2}{\rho} + \frac{1}{c}\frac{8}{n}\gamma^2(1-\beta)^2L^2\bar{\omega}^2(1-\delta) + \frac{a}{c}\gamma\frac{13L^2}{\rho}\bar{\omega}^2(1-\delta) + \frac{b}{c}\gamma^2\frac{60L^2\bar{\omega}^2}{\delta} \leq 1 - \frac{\delta\gamma}{8}.$$

To this end, with the step size condition

$$\gamma \leq \min\left\{\frac{1}{4\rho}, \frac{\rho n}{64(1-\beta)^2\bar{\omega}^2}, \frac{n}{8(1-\beta)^2(1-\delta)\rho}\right\}, \tag{S1}$$

the above set of inequalities can be guaranteed if $a, b, c$ satisfy

$$\frac{96L^2}{\rho^2\gamma^2}\eta^2 \leq a \leq \min\left\{\frac{(1-\beta)\gamma\rho}{21\beta n}, \frac{2}{13(1-\delta)}, \frac{\rho^2}{600\bar{\omega}^2}\right\} \tag{46}$$

$$\max\left\{a\frac{\gamma}{1-\gamma}\frac{32\bar{\omega}^2}{\rho\delta}, c\frac{\eta^2}{1-\gamma}\frac{48\bar{\omega}^2}{\rho\delta}\right\} \leq b \leq \min\left\{\frac{\eta^2}{\gamma^3}\frac{2\delta L^2}{5\rho(\bar{\omega}^2 + \frac{\rho^2}{8})}, \frac{\delta}{30\rho\gamma}, \frac{(1-\beta)\delta}{93\beta n}, \frac{\delta\rho}{2928\gamma\bar{\omega}^2}\right\} \tag{47}$$

$$\frac{\gamma}{1-\gamma}\frac{48L^2\bar{\omega}^2}{\delta\rho} \leq c \leq \min\left\{\frac{\delta\rho L^2}{1392\gamma\bar{\omega}^2}, \frac{2\delta L^2}{25\rho\gamma}, \frac{1-\beta}{\beta\eta^2}\frac{\delta}{3n(18 + \frac{84}{\rho\gamma})}, \frac{\gamma}{\eta^2}\frac{\rho^2}{1824\bar{\omega}}\right\} \tag{48}$$

Notice that the step size conditions:

$$\eta^2 \leq \min\left\{\frac{(1-\beta)\gamma^3}{2016\beta n}\frac{\rho^3}{L^2}, \frac{\gamma^2\rho^2}{624(1-\delta)L^2}, \frac{\gamma^2\rho^4}{57600\bar{\omega}^2 L^2}\right\} \tag{S2}$$

guarantees the existence of $a$ which satisfies (46). In particular, we take $a = \frac{96L^2}{\rho^2\gamma^2}\eta^2$.

At the same time, with the step size conditions:

$$\eta^2 \leq \min\left\{\frac{(1-\beta)\gamma}{\beta n}\frac{464\bar{\omega}^2}{(18 + \frac{84}{\rho\gamma})\rho L^2}, \gamma^2\frac{29\rho\bar{\omega}}{38\delta L^2}\right\}, \quad \frac{\gamma^2}{1-\gamma} \leq \frac{\delta^2\rho^2}{66816\bar{\omega}^4}, \tag{S3}$$

we guarantee the existence of $c$ which satisfies (48). In particular, we take $c = \frac{\gamma}{1-\gamma}\frac{48L^2\bar{\omega}^2}{\delta\rho}$. This simplifies (47) into

$$\max\left\{\frac{\eta^2}{\gamma(1-\gamma)}\frac{3072\bar{\omega}^2 L^2}{\delta\rho^3}, \frac{\eta^2\gamma}{(1-\gamma)^2}\frac{2304\bar{\omega}^4 L^2}{\delta^2\rho^2}\right\} \le b \le \min\left\{\frac{\eta^2}{\gamma^3}\frac{2\delta L^2}{5\rho(\bar{\omega}^2 + \frac{\rho^2}{8})}, \frac{\delta}{30\rho\gamma}, \frac{(1-\beta)\delta}{93\beta n}, \frac{\delta\rho}{2928\gamma\bar{\omega}^2}\right\} \tag{49}$$

Combining with the step size conditions:

$$\eta^2 \le \min\left\{\frac{\gamma^2(\bar{\omega}^2 + \frac{\rho^2}{8})}{12L^2}, \frac{(1-\beta)\gamma^3}{\beta n}\frac{5\rho(\bar{\omega}^2 + \frac{\rho^2}{8})}{186L^2}, \gamma^2\frac{5\rho^2(\bar{\omega}^2 + \frac{\rho^2}{8})}{5856\bar{\omega}^2 L^2}\right\},$$

$$\frac{\gamma^2}{1-\gamma} \le \min\left\{\frac{\delta^2\rho^2}{7680\bar{\omega}^2(\bar{\omega}^2 + \frac{\rho^2}{8})}, \frac{4\delta}{3\bar{\omega}^2\rho}\right\} \tag{S4}$$

guarantees the existence of $b$ which satisfies (49). Finally, we take $b = \frac{\eta^2}{\gamma(1-\gamma)}\frac{3072\bar{\omega}^2 L^2}{\delta\rho^3}$.

Using the upper bound on $\gamma^2/(1-\gamma)$ from (S4) and the above choices of $a, b, c$ yield:

$$\mathbf{v}^{t+1} \le \left(1 - \min\left\{\frac{\rho\gamma}{8}, \frac{\delta\gamma}{8}, \beta\right\}\right)\mathbf{v}^t + \beta^2\left[4 + \frac{\eta^2}{\gamma^3}\frac{672L^2 n}{\rho^3} + \frac{\eta^2}{\gamma}\frac{6L^2 n\rho^4\delta}{25\bar{\omega}^2} + \frac{\eta^2}{\gamma^2}\frac{4L^2 n}{\bar{\omega}^2}\right]\sigma^2$$

$$+ \eta^2\left[8(1-\beta)^2 L^2(1-\rho\gamma)^2 + \frac{L^2 n}{\rho\gamma}\left(96 + \frac{141}{400}\frac{\rho^2}{\bar{\omega}^2}\right)\right]\mathbb{E}\left[\left\|\bar{g}^t\right\|^2\right]$$

Furthermore, we observe that the above steps require step size conditions (S1), (S2), (S3), (S4). Together with the requirements in Lemma 3.5, 3.6, 3.7, 3.9, 3.8, we need

$$\eta^2 \le \min\left\{\frac{(1-\beta)\gamma^3}{2016\beta n}\frac{\rho^3}{L^2}, \frac{(1-\beta)\gamma}{\beta n}\frac{464\bar{\omega}^2}{(18 + \frac{84}{\rho\gamma})\rho L^2}, \gamma^2\frac{29\rho\bar{\omega}}{38\delta L^2}, \frac{\gamma^2(\bar{\omega}^2 + \frac{\rho^2}{8})}{12L^2}, \frac{(1-\beta)\gamma^3}{\beta n}\frac{5\rho(\bar{\omega}^2 + \frac{\rho^2}{8})}{186L^2}, \right.$$

$$\left. \frac{5\gamma^2\rho^2(\bar{\omega}^2 + \frac{\rho^2}{8})}{5856\bar{\omega}^2 L^2}, \frac{\gamma^2\rho^2}{624(1-\delta)L^2}, \frac{\gamma^2\rho^4}{57600\bar{\omega}^2 L^2}, \frac{\rho^2\gamma^2}{100L^2(1-\rho\gamma)^2(1 + \gamma^2\bar{\omega}^2)}, \frac{\delta^2\rho}{1248\bar{\omega}^2 L^2\gamma}\right\}$$

$$\gamma \le \min\left\{\frac{1}{4\rho}, \frac{\rho n}{64(1-\beta)^2\bar{\omega}^2}, \frac{n}{8(1-\beta)^2(1-\delta)\rho}, \frac{\delta}{8\bar{\omega}}, \frac{\sqrt{\delta}}{4\bar{\omega}\sqrt{(1-\delta)(1 + 3\eta^2 L^2)(1 + 2/\delta)}}\right\},$$

$$\frac{\gamma^2}{1-\gamma} \le \min\left\{\frac{\delta^2\rho^2}{66816\bar{\omega}^4}\frac{\delta^2\rho^2}{7680\bar{\omega}^2(\bar{\omega}^2 + \frac{\rho^2}{8})}, \frac{4\delta}{3\bar{\omega}^2\rho}\right\}$$

Taking the restriction that $\bar{\omega} \in [1, 2]$, the above can be simplified and implied by

$$\gamma \le \min\left\{\frac{1}{4\rho}, \frac{\rho n}{64\bar{\omega}^2}, \frac{\delta}{10\bar{\omega}}, \frac{\delta\rho\sqrt{1 - \delta/(8\bar{\omega}^2)}}{259\bar{\omega}^2}\right\} =: \gamma_\infty,$$

$$\eta \le \frac{\gamma}{L}\min\left\{\sqrt{\frac{1-\beta}{\beta n}}\frac{\sqrt{\gamma\rho^3}}{45}, \frac{\rho^2}{240\bar{\omega}}\right\} =: \eta_\infty.$$

where we have used the upper bound $\gamma \le \frac{\delta}{8\bar{\omega}}$ to remove the self-dependence on $1 - \gamma$ for the constraints on $\gamma$. This concludes the proof.

## A.8 Auxilliary Lemmas

**Lemma A.1.** *Under Assumption 2.2, 2.4. For any $t \ge 0$, it holds*

$$\mathbb{E}\left[\left\|\Theta^{t+1} - \Theta^t\right\|_F^2\right] \le 4\eta^2(1-\rho\gamma)^2\mathbb{E}\left[\left\|G_o^t\right\|_F^2\right] + 2n\eta^2(1-\rho\gamma)^2\mathbb{E}\left[\left\|\bar{g}^t\right\|^2\right] \tag{50}$$

$$+ 4\bar{\omega}^2\gamma^2\mathbb{E}\left[\left\|\Theta_o^t\right\|_F^2\right] + 2\bar{\omega}^2\gamma^2(1-\delta)\mathbb{E}\left[\left\|\Theta^t - \eta G^t - \hat{\Theta}^t\right\|_F^2\right].$$

*Proof.* We observe that:

$$\left\|\Theta^{t+1} - \Theta^t\right\|_F^2 = \left\|\eta G^t - \gamma(\mathbf{W} - \mathbf{I})\hat{\Theta}^{t+1}\right\|_F^2$$

$$= \left\|(I + \gamma(\mathbf{W} - \mathbf{I}))(-\eta G^t) + \gamma(\mathbf{W} - \mathbf{I})\Theta^t + \gamma(\mathbf{W} - \mathbf{I})\left[\mathcal{Q}(\Theta^t - \eta G^t - \hat{\Theta}^t) - (\Theta^t - \eta G^t - \hat{\Theta}^t)\right]\right\|_F^2$$

$$\leq 2\left\|(I + \gamma(\mathbf{W} - \mathbf{I}))(-\eta G^t) + \gamma(\mathbf{W} - \mathbf{I})\Theta^t\right\|_F^2$$

$$\quad + 2\left\|\gamma(\mathbf{W} - \mathbf{I})\left[\mathcal{Q}(\Theta^t - \eta G^t - \hat{\Theta}^t) - (\Theta^t - \eta G^t - \hat{\Theta}^t)\right]\right\|_F^2$$

$$\leq 2\left[\eta^2\left\|(I + \gamma(\mathbf{W} - \mathbf{I}))G^t\right\|_F^2 + \gamma^2\left\|(\mathbf{W} - \mathbf{I})\Theta^t\right\|_F^2 + 2\left\langle(I + \gamma(\mathbf{W} - \mathbf{I}))(-\eta G^t) \mid \gamma(\mathbf{W} - \mathbf{I})\Theta^t\right\rangle\right]$$

$$\quad + 2\bar{\omega}^2\gamma^2\left\|\mathcal{Q}(\Theta^t - \eta G^t - \hat{\Theta}^t) - (\Theta^t - \eta G^t - \hat{\Theta}^t)\right\|_F^2 \tag{51}$$

Observe that $\langle(\mathbf{I} + \gamma(\mathbf{W} - \mathbf{I}))(-\eta G^t) \mid \gamma(\mathbf{W} - \mathbf{I})\Theta^t\rangle = \langle(\mathbf{I} + \gamma(\mathbf{W} - \mathbf{I}))(-\eta \mathbf{U}G_o^t) \mid \gamma(\mathbf{W} - \mathbf{I})\Theta^t\rangle$, the above leads to

$$\mathbb{E}_t\left[\left\|\Theta^{t+1} - \Theta^t\right\|_F^2\right] \leq 2\eta^2(1 - \rho\gamma)^2\left\|G^t\right\|_F^2 + 2\eta^2(1 - \rho\gamma)^2\left\|G_o^t\right\|_F^2 + 4\gamma^2\left\|(\mathbf{W} - \mathbf{I})\Theta^t\right\|_F^2$$

$$\quad + 2\bar{\omega}^2\gamma^2(1 - \delta)\left\|\Theta^t - \eta G^t - \hat{\Theta}^t\right\|_F^2 \tag{52}$$

Notice that

$$\mathbb{E}\left[\left\|G^t\right\|_F^2\right] = \mathbb{E}\left[\left\|((1/n)\mathbf{1}\mathbf{1}^\top + \mathbf{U}\mathbf{U}^\top)G^t\right\|_F^2\right] = \mathbb{E}\left[\left\|\mathbf{U}\mathbf{U}^\top G^t\right\|_F^2 + \left\|(1/n)\mathbf{1}\mathbf{1}^\top G^t\right\|_F^2\right]$$

$$\leq \mathbb{E}\left[\left\|G_o^t\right\|_F^2 + n\left\|\bar{g}^t\right\|^2\right] \tag{53}$$

By combining (52), (53), and the fact $\|(\mathbf{W} - \mathbf{I})\Theta^t\|_F^2 \leq \bar{\omega}^2\|\Theta_o^t\|_F^2$, we have

$$\mathbb{E}\left[\left\|\Theta^{t+1} - \Theta^t\right\|_F^2\right] \leq 4\eta^2(1 - \rho\gamma)^2\mathbb{E}\left[\left\|G_o^t\right\|_F^2\right] + 2n\eta^2(1 - \rho\gamma)^2\mathbb{E}\left[\left\|\bar{g}^t\right\|^2\right]$$

$$\quad + 4\bar{\omega}^2\gamma^2\mathbb{E}\left[\left\|\Theta_o^t\right\|_F^2\right] + 2\bar{\omega}^2\gamma^2(1 - \delta)\mathbb{E}\left[\left\|\Theta^t - \eta G^t - \hat{\Theta}^t\right\|_F^2\right]$$

This concludes the proof. □

**Lemma A.2.** *Under Assumption 2.1, 2.3. For any $t \geq 0$, it holds*

$$\mathbb{E}\left[\left\|V^{t+1} - V^t\right\|_F^2\right] \leq 3L^2\mathbb{E}\left[\left\|\Theta^{t+1} - \Theta^t\right\|_F^2\right] + 3\beta^2\mathbb{E}\left[\left\|V^t - \nabla F^t\right\|_F^2\right] + 3n\beta^2\sigma^2$$

*Proof.* Observe that

$$V^{t+1} - V^t = \nabla\widehat{F}^{t+1} + (1 - \beta)(V^t - \nabla\widetilde{F}^t) - V^t$$

$$= \nabla\widehat{F}^{t+1} - \nabla\widetilde{F}^t - \beta(V^t - \nabla F^t) + \beta(\nabla\widetilde{F}^t - \nabla F^t)$$

It holds that

$$\mathbb{E}\left[\left\|V^{t+1} - V^t\right\|_F^2\right] \leq 3L^2\mathbb{E}\left[\left\|\Theta^{t+1} - \Theta^t\right\|_F^2\right] + 3\beta^2\mathbb{E}\left[\left\|V^t - \nabla F^t\right\|_F^2\right] + 3\beta^2\mathbb{E}\left[\left\|\nabla\widetilde{F}^t - \nabla F^t\right\|_F^2\right]$$

$$\leq 3L^2\mathbb{E}\left[\left\|\Theta^{t+1} - \Theta^t\right\|_F^2\right] + 3\beta^2\mathbb{E}\left[\left\|V^t - \nabla F^t\right\|_F^2\right] + 3n\beta^2\sigma^2$$

This concludes the proof. □

### A.9 Transient Time of `DoCoM`

We follow a similar argument as in (12). Particularly, consider setting the step sizes and parameters as $\beta = \Theta(\frac{1}{T^{2/3}}), \eta = \Theta(\frac{1}{LT^{1/3}}), \gamma = \gamma_\infty, b_0 = \Omega(T^{1/3})$. Then for sufficiently large $T$, we obtain

$$\frac{1}{n} \sum_{i=1}^{n} \mathbb{E}\left[\left\|\nabla f(\theta_i^\mathsf{T})\right\|^2\right] = \mathcal{O}\left(\frac{L(f(\bar{\theta}^0) - f^\star)}{T^{2/3}} + \frac{\sigma^2}{nT^{2/3}} + \frac{\overline{G}_0}{\delta^2 \rho^4 T} + \frac{\sigma^2}{\delta^3 \rho^6 T^{4/3}}\right). \tag{54}$$

The transient time can be calculated by bounding $T$ such that the second term dominates over the last two terms. We get

$$T_{\text{trans}} := \Omega\left(\max\left\{\frac{n^3 \overline{G}_0^3}{\sigma^6 \delta^6 \rho^{12}}, \frac{n^{1.5}}{\delta^{1.5} \rho^6}\right\}\right) \tag{55}$$

Taking $\sigma \leq 1$ guarantees that $T_{\text{trans}} = \Omega(\frac{n^3 \overline{G}_0^3}{\sigma^6 \delta^6 \rho^{12}})$.

### A.10 Sublinear bound of $\mathcal{O}(\log T/T)$ under PL Condition

We consider setting $\beta = \eta = \log T/(\mu T)$, $\gamma = \gamma_\infty$, $b_0 = 1$. Notice that for a sufficiently large $T$, these step sizes will satisfy (10). Furthermore, setting $t = T$, the upper bound in (14) is given by:

$$\left(1 - \frac{\log T}{T}\right)^T \left(\Delta^0 + \frac{2}{n}\left(2\sigma^2 + \frac{118 L^2 n}{\rho^2 \gamma^2 (1-\gamma)} \overline{G}_0 \frac{(\log T)^2}{T^2}\right)\right) + \frac{2\mathbb{C}_\sigma \sigma^2}{n} \frac{\log T}{\mu T} \tag{56}$$

We observe that

$$\left(1 - \frac{\log T}{T}\right)^T \leq e^{-\log T} = \frac{1}{T}.$$

Thus the expression in (56) can be further upper bounded by $\mathcal{O}(\log T/T)$. This implies (16).

## B    Convergence Analysis of `DoCoM` with $\beta = 1$

This section provides the convergence analysis of (9) for the special case when the momentum parameter is $\beta = 1$, i.e., there is no momentum applied. Observe that the `DoCoM` algorithm can be simplified as

$$
\begin{aligned}
\Theta^{t+1} &= \Theta^t - \eta G^t + \gamma(\mathbf{W} - \mathbf{I})\widehat{\Theta}^{t+1} \\
\widehat{\Theta}^{t+1} &= \widehat{\Theta}^t + \mathcal{Q}(\Theta^t - \eta G^t - \widehat{\Theta}^t) \\
G^{t+1} &= G^t + \nabla\widehat{F}^{t+1} - \nabla\widehat{F}^t + \gamma(\mathbf{W} - \mathbf{I})\widehat{G}^{t+1} \\
\widehat{G}^{t+1} &= \widehat{G}^t + \mathcal{Q}(G^t + \nabla\widehat{F}^{t+1} - \nabla\widehat{F}^t - \widehat{G}^t)
\end{aligned}
$$

where we have eliminated the use of $V^t$ since $V^t = \nabla\widehat{F}^t$ for any $t \geq 0$. Additionally, we define $\bar{\Theta}^t = (1/n)\mathbf{1}\mathbf{1}^\top \Theta^t$.

In such setting, the convergence analysis has to follow a different path from the case of $\beta < 1$. We begin by analyzing the 1-iteration progress with

**Lemma B.1.** *Under Assumption 2.1, 2.2, 2.3, and the step size satisfies $\eta \leq 1/(4L)$. Then, for any $t \geq 0$, it holds*

$$\mathbb{E}_t[f(\bar{\theta}^{t+1})] \leq f(\bar{\theta}^t) - \frac{\eta}{4}\left\|\nabla f(\bar{\theta}^t)\right\|^2 + \frac{3L^2\eta}{4n}\left\|\Theta_o^t\right\|_F^2 + \frac{L\eta^2\sigma^2}{2n}. \tag{57}$$

The proof is relegated to Appendix B.1. Notice that the above lemma departs from Lemma 3.4 as it results in a bound that depends only on the consensus error. Our next endeavor is to bound $\|\Theta_o^t\|_F^2$, which can be conveniently controlled by Lemma 3.5 as quoted below:

$$\mathbb{E}[\left\|\Theta_o^{t+1}\right\|_F^2] \leq (1 - \frac{\rho\gamma}{2})\mathbb{E}[\left\|\Theta_o^t\right\|_F^2] + \frac{2}{\rho}\frac{\eta^2}{\gamma}\mathbb{E}[\left\|G_o^t\right\|_F^2] + \frac{\bar{\omega}^2}{\rho}\gamma\,\mathbb{E}\left[\left\|\Theta^t - \eta G^t - \widehat{\Theta}^t\right\|_F^2\right]. \tag{58}$$

Moreover,

**Lemma B.2.** *Under Assumption 2.1, 2.2, 2.3, 2.4. Suppose the step size satisfies $\eta \leq \frac{\rho\gamma}{8L(1-\rho\gamma)\sqrt{1+\gamma^2\bar{\omega}^2}}$, $\gamma \leq \frac{1}{8\bar{\omega}}$. Then, for any $t \geq 0$, the consensus error of $G^{t+1}$ is bounded as follows:*

$$\mathbb{E}\left[\left\|G_o^{t+1}\right\|_F^2\right] \leq \left(1 - \frac{\rho\gamma}{4}\right)\mathbb{E}\left[\left\|G_o^t\right\|_F^2\right] + \gamma\frac{2\bar{\omega}^2}{\rho}\mathbb{E}\left[\left\|G^t - \widehat{G}^t\right\|_F^2\right] + \gamma\left(\frac{\rho}{4} + \frac{18L^2\bar{\omega}^2}{\rho}\right)\mathbb{E}\left[\left\|\Theta_o^t\right\|_F^2\right]$$

$$+ \gamma\frac{9L^2}{\rho}\bar{\omega}^2\mathbb{E}\left[\left\|\Theta^t - \eta G^t - \widehat{\Theta}^t\right\|_F^2\right] + \gamma\frac{\rho n}{4}\mathbb{E}\left[\left\|\nabla f(\bar{\theta}^t)\right\|^2\right] + \left(\frac{7n}{\rho\gamma} + \frac{\rho\gamma}{8}\right)\sigma^2.$$

The proof is relegated to Appendix B.2. We also bound the subsequent terms by

**Lemma B.3.** *Under Assumption 2.1, 2.2, 2.3, 2.4. Suppose the step size satisfies $\eta \leq \frac{\rho\gamma}{8L(1-\rho\gamma)\sqrt{1+\gamma^2\bar{\omega}^2}}$, $\gamma \leq \frac{\delta}{8\bar{\omega}}$. Then, for any $t \geq 0$, it holds:*

$$\mathbb{E}\left[\left\|G^{t+1} - \hat{G}^{t+1}\right\|_F^2\right] \leq \left(1 - \frac{\delta}{8}\right)\mathbb{E}\left[\left\|G^t - \hat{G}^t\right\|_F^2\right] + \frac{10}{\delta}\gamma^2\left(\bar{\omega}^2 + \frac{\rho^2}{8}\right)\mathbb{E}\left[\left\|G_o^t\right\|_F^2\right]$$

$$+ \gamma^2\frac{5}{4\delta}\left(\rho^2 + 72L^2\bar{\omega}^2\right)\mathbb{E}\left[\left\|\Theta_o^t\right\|_F^2\right] + \gamma^2\frac{40L^2\bar{\omega}^2}{\delta}\mathbb{E}\left[\left\|\Theta^t - \eta G^t - \hat{G}^t\right\|_F^2\right]$$

$$+ \gamma^2\frac{5\rho^2}{4\delta}n\mathbb{E}\left[\left\|\nabla f(\bar{\theta}^t)\right\|^2\right] + \frac{5}{\delta}\left(\frac{\rho^2\gamma^2}{8} + 7n\right)\sigma^2.$$

**Lemma B.4.** *Under Assumption 2.1, 2.2, 2.3, 2.4. Suppose the step size satisfies $\gamma \leq \frac{\sqrt{\delta}}{4\bar{\omega}\sqrt{(1-\delta)(1+\eta^2L^2)(1+2/\delta)}}$. Then, for any $t \geq 0$, it holds*

$$\mathbb{E}\left[\left\|\Theta^{t+1} - \eta G^{t+1} - \hat{\Theta}^{t+1}\right\|_F^2\right] \leq (1 - \frac{\delta}{4})\mathbb{E}\left[\left\|\Theta^t - \eta G^t - \hat{\Theta}^t\right\|_F^2\right] + \eta^2\frac{12}{\delta}\mathbb{E}\left[\left\|G_o^{t+1}\right\|_F^2\right]$$

$$+ \frac{24}{\delta}\eta^2\sigma^2 + \frac{38}{\delta}\eta^2\mathbb{E}\left[\left\|G_o^t\right\|_F^2\right] + \frac{26}{\delta}\mathbb{E}\left[(\eta^2L^2 + \bar{\omega}^2\gamma^2)\left\|\Theta_o^t\right\|_F^2 + \eta^2n\left\|\nabla f(\bar{\theta}^t)\right\|^2\right]$$

*Furthermore, if the step size satisfies $\eta^2\gamma \leq \frac{\delta^2\rho}{864\bar{\omega}^2L^2}$, then*

$$\mathbb{E}\left[\left\|\Theta^{t+1} - \eta G^{t+1} - \hat{\Theta}^{t+1}\right\|_F^2\right] \leq \left(1 - \frac{\delta}{8}\right)\mathbb{E}\left[\left\|\Theta^t - \eta G^t - \hat{\Theta}^t\right\|_F^2\right]$$

$$+ \eta^2\frac{50}{\delta}\mathbb{E}\left[\left\|G_o^t\right\|_F^2\right] + \eta^2\frac{3\bar{\omega}}{\rho}\mathbb{E}\left[\left\|G^t - \hat{G}^t\right\|_F^2\right]$$

$$+ \left[\eta^2\frac{29L^4}{\delta}\left(1 + \frac{\bar{\omega}}{\rho}\right) + \frac{26}{\delta}\bar{\omega}^2\gamma^2\right]\mathbb{E}\left[\left\|\Theta_o^t\right\|_F^2\right]$$

$$+ \eta^2\frac{29n}{\delta}\mathbb{E}\left[\left\|\nabla f(\bar{\theta}^t)\right\|^2\right] + \eta^2\frac{24\sigma^2}{\delta}\left(1 + \frac{4n}{\rho\gamma}\right)$$

The proofs are relegated to Appendix B.3, B.4, respectively. Combining the above lemmas and optimizing the bounds for step sizes lead to

**Lemma B.5.** *Under Assumption 2.1, 2.2, 2.3, 2.4 and the step size conditions*

$$\eta \leq \frac{1}{L}\min\left\{\frac{1}{4}, \frac{\rho\gamma}{8(1-\rho\gamma)\sqrt{1+\gamma^2\bar{\omega}^2}}, \frac{\rho^2\gamma}{105\bar{\omega}}, \frac{\rho^2\delta\sqrt{1-\gamma}}{1303\bar{\omega}^2}, \frac{\rho^{3/2}\delta\sqrt{1-\gamma}}{130L\bar{\omega}^{3/2}}\right\}$$

$$\gamma \leq \min\left\{\frac{1}{\rho}, \frac{\delta}{8\bar{\omega}}, \frac{\sqrt{\delta}}{4\bar{\omega}\sqrt{(1-\delta)(1+\eta^2L^2)(1+2/\delta)}}, \frac{\rho\delta^{3/2}\sqrt{1-\delta/(8\bar{\omega})}}{88\bar{\omega}\sqrt{\delta\bar{\omega}^2+\rho^2}}, \frac{\rho\delta\sqrt{1-\delta/(8\bar{\omega})}}{123\bar{\omega}^2}\right\}, \quad \eta^2\gamma \leq \frac{\delta^2\rho}{864\bar{\omega}^2L^2}.$$

*Define the constants:*

$$\mathbb{C}_\sigma^{\mathsf{NM}} = \frac{192}{1-\gamma}\left[\frac{2(1-\gamma)}{\rho^3\gamma^3} + \frac{320\bar{\omega}^2}{\rho^3\delta^2\gamma} + \frac{15\bar{\omega}^2}{\rho^2\delta^2\gamma}\right], \quad \mathbb{C}_{\nabla f}^{\mathsf{NM}} = \frac{12}{1-\gamma}\left[\frac{1-\gamma}{\rho\gamma} + \frac{256\bar{\omega}^2}{\rho\delta^2}\gamma + \frac{58\bar{\omega}^2}{\rho\delta^2}\right]$$

*Then, for any $t \geq 0$, it holds*

$$\mathbb{E}\left[\left\|\Theta_o^{t+1}\right\|_F^2 + a\left\|G_o^{t+1}\right\|_F^2 + b\left\|G^{t+1} - \hat{G}^{t+1}\right\|_F^2 + c\left\|\Theta^{t+1} - \eta G^{t+1} - \hat{\Theta}^{t+1}\right\|_F^2\right]$$

$$\leq \left(1 - \frac{\min\{\rho, \delta\}}{8}\gamma\right)\mathbb{E}\left[\left\|\Theta_o^t\right\|_F^2 + a\left\|G_o^t\right\|_F^2 + b\left\|G^t - \hat{G}^t\right\|_F^2 + c\left\|\Theta^t - \eta G^t - \hat{\Theta}^t\right\|_F^2\right]$$

$$+ \eta^2 \mathbb{C}_\sigma n\sigma^2 + \eta^2 \mathbb{C}_{\nabla f}^{\mathsf{NM}} n \mathbb{E}\left[\left\|\nabla f(\bar{\theta}^t)\right\|^2\right]$$

*where $a = \frac{48\eta^2}{\rho^2\gamma^2}$, $b = \frac{1536\bar{\omega}^2\eta^2}{\rho^3\gamma\delta(1-\gamma)}$, $c = \frac{24\bar{\omega}^2}{\rho\delta(1-\gamma)}$.*

The proof is relegated to Appendix B.5.

To simplify notations, we let $\bar{\rho}^{\mathsf{NM}} := \min\{\rho, \delta\}/8$. Using $\mathbb{E}[\|G_o^0\|_F^2] = n\sigma^2$ and Lemma B.5 imply that

$$\mathbb{E}\left[\left\|\Theta_o^t\right\|_F^2\right] \leq a\left(1 - \bar{\rho}^{\mathsf{NM}}\gamma\right)^t n\sigma^2 + \eta^2 \sum_{s=0}^{t-1}(1 - \bar{\rho}^{\mathsf{NM}}\gamma)^{t-s-1}\{\mathbb{C}_\sigma^{\mathsf{NM}} n\sigma^2 + \mathbb{C}_{\nabla f}^{\mathsf{NM}} n \mathbb{E}\left[\left\|\nabla f(\bar{\theta}^s)\right\|^2\right]\}$$

$$\leq \mathbb{C}_\sigma^{\mathsf{NM}}\frac{\eta^2 n\sigma^2}{\bar{\rho}^{\mathsf{NM}}\gamma} + \eta^2 n\mathbb{C}_{\nabla f}^{\mathsf{NM}}\sum_{s=0}^{t-1}(1 - \bar{\rho}^{\mathsf{NM}}\gamma)^{t-s-1}\mathbb{E}\left[\left\|\nabla f(\bar{\theta}^s)\right\|^2\right], \tag{59}$$

where the inequality is due to the fact that $\mathbb{C}_\sigma^{\mathsf{NM}}\eta^2 \geq a$. We now observe that (57) implies

$$\frac{\eta}{4T}\sum_{t=0}^{T-1}\mathbb{E}\left[\left\|\nabla f(\bar{\theta}^t)\right\|^2\right] \leq \mathbb{E}\left[\frac{f(\bar{\theta}^0) - f(\bar{\theta}^T)}{T}\right] + \eta^2\frac{\sigma^2 L}{2n}\left(1 + \mathbb{C}_\sigma^{\mathsf{NM}}\frac{3Ln}{2\bar{\rho}^{\mathsf{NM}}\gamma}\eta\right)$$

$$+ \eta^3\mathbb{C}_{\nabla f}^{\mathsf{NM}}\frac{3L^2}{4}\frac{1}{T}\sum_{t=0}^{T-1}\sum_{s=0}^{t-1}(1 - \bar{\rho}^{\mathsf{NM}}\gamma)^{t-s-1}\mathbb{E}\left[\left\|\nabla f(\bar{\theta}^s)\right\|^2\right]$$

$$\leq \mathbb{E}\left[\frac{f(\bar{\theta}^0) - f(\bar{\theta}^T)}{T}\right] + \eta^2\frac{\sigma^2 L}{2n}\left(1 + \mathbb{C}_\sigma^{\mathsf{NM}}\frac{3Ln}{2\bar{\rho}^{\mathsf{NM}}\gamma}\eta\right)$$

$$+ \eta^3\mathbb{C}_{\nabla f}^{\mathsf{NM}}\frac{3L^2}{4\bar{\rho}^{\mathsf{NM}}\gamma}\frac{1}{T}\sum_{s=0}^{T-2}\mathbb{E}\left[\left\|\nabla f(\bar{\theta}^s)\right\|^2\right]$$

Therefore, under the additional step size condition $\eta \leq \sqrt{\frac{\bar{\rho}^{\mathsf{NM}}\gamma}{6\mathbb{C}_{\nabla f}^{\mathsf{NM}}L^2}}$, it holds

$$\frac{1}{T}\sum_{t=0}^{T-1}\mathbb{E}\left[\left\|\nabla f(\bar{\theta}^t)\right\|^2\right] \leq \frac{8}{\eta}\mathbb{E}\left[\frac{f(\bar{\theta}^0) - f(\bar{\theta}^T)}{T}\right] + \eta\frac{4\sigma^2 L}{n}\left(1 + \mathbb{C}_\sigma^{\mathsf{NM}}\frac{3Ln}{2\bar{\rho}^{\mathsf{NM}}\gamma}\eta\right) \tag{60}$$

Setting $\eta = \mathcal{O}(1/\sqrt{T})$ shows the expected convergence rate of $\frac{1}{T}\sum_{t=0}^{T-1}\mathbb{E}\left[\left\|\nabla f(\bar{\theta}^t)\right\|^2\right] = \mathcal{O}(1/\sqrt{T})$. We remark that similar result to Corollary 3.3 can be established under the PL condition Assumption 3.2.

## B.1 Proof of Lemma B.1

Using the $L$-smoothness of $f$, we obtain:

$$f(\bar{\theta}^{t+1}) \leq f(\bar{\theta}^t) + \left\langle\nabla f(\bar{\theta}^t) \mid \bar{\theta}^{t+1} - \bar{\theta}^t\right\rangle + \frac{L}{2}\left\|\bar{\theta}^{t+1} - \bar{\theta}^t\right\|^2$$

$$= f(\bar{\theta}^t) - \eta\left\|\nabla f(\bar{\theta}^t)\right\|^2 - \eta\left\langle\nabla f(\bar{\theta}^t) \mid \bar{g}^t - \nabla f(\bar{\theta}^t)\right\rangle + \frac{L\eta^2}{2}\left\|\bar{g}^t\right\|^2 \tag{61}$$

Taking the conditional expectation on the inner product:

$$-\eta\mathbb{E}_t\left\langle\nabla f(\bar{\theta}^t) \mid \bar{g}^t - \nabla f(\bar{\theta}^t)\right\rangle$$

$$= -\eta \left\langle \nabla f(\bar{\theta}^t) \, \middle| \, \frac{1}{n} \sum_{i=1}^n \nabla f_i(\theta_i^t) - \nabla f(\bar{\theta}^t) \right\rangle$$

$$\leq \frac{\eta}{2} \left\| \nabla f(\bar{\theta}^t) \right\|^2 + \frac{\eta}{2} \left\| \frac{1}{n} \sum_{i=1}^n \left\{ \nabla f_i(\theta_i^t) - \nabla f_i(\bar{\theta}^t) \right\} \right\|^2$$

$$\leq \frac{\eta}{2} \left\| \nabla f(\bar{\theta}^t) \right\|^2 + \frac{\eta}{2n} \sum_{i=1}^n \left\| \nabla f_i(\theta_i^t) - \nabla f_i(\bar{\theta}^t) \right\|^2$$

$$\leq \frac{\eta}{2} \left\| \nabla f(\bar{\theta}^t) \right\|^2 + \frac{\eta}{2n} \sum_{i=1}^n L^2 \left\| \theta_i^t - \bar{\theta}^t \right\|^2$$

$$= \frac{\eta}{2} \left\| \nabla f(\bar{\theta}^t) \right\|^2 + \frac{\eta L^2}{2n} \left\| \Theta^t - \bar{\Theta}^t \right\|_F^2$$

where we denote $\bar{\Theta}^t = \mathbf{1}(\bar{\theta}^t)^\top = \frac{1}{n}\mathbf{1}\mathbf{1}^\top \Theta^t$.

Putting back into (61) yields

$$\mathbb{E}_t[f(\bar{\theta}^{t+1})] \leq f(\bar{\theta}^t) - \frac{\eta}{2} \left\| \nabla f(\bar{\theta}^t) \right\|^2 + \frac{L\eta^2}{2} \mathbb{E}_t \left\| \bar{g}^t \right\|^2 + \frac{\eta L^2}{2n} \left\| \Theta^t - \bar{\Theta}^t \right\|_F^2 \tag{62}$$

Observe that as $\mathbb{E}_t \left\| \bar{g}^t \right\|^2 = \frac{1}{n^2} \mathbb{E}_t \left\| \mathbf{1}^\top G^t \right\|^2$, we have

$$\mathbb{E}_t \left\| \mathbf{1}^\top G^t \right\|_F^2 = \mathbb{E}_t \left\| \sum_{i=1}^n \nabla f_i(\theta_i^t; \zeta_i^{t+1}) \right\|^2 = \mathbb{E}_t \left\| \sum_{i=1}^n \nabla f_i(\theta_i^t; \zeta_i^{t+1}) - \sum_{i=1}^n \nabla f_i(\theta_i^t) \right\|^2 + \left\| \sum_{i=1}^n \nabla f_i(\theta_i^t) \right\|^2$$

$$\leq n\sigma^2 + 2 \left\| \sum_{i=1}^n \left\{ \nabla f_i(\theta_i^t) - \nabla f_i(\bar{\theta}^t) \right\} \right\|^2 + 2 \left\| \sum_{i=1}^n \nabla f_i(\bar{\theta}^t) \right\|^2$$

$$\leq n\sigma^2 + 2nL^2 \left\| \Theta^t - \bar{\Theta}^t \right\|_F^2 + 2n^2 \left\| \nabla f(\bar{\theta}^t) \right\|^2. \tag{63}$$

Putting back into (62), and assuming that $-(\frac{1}{2} - L\eta) \leq -\frac{1}{4} \Leftrightarrow \eta \leq \frac{1}{4L}$,

$$\mathbb{E}_t[f(\bar{\theta}^{t+1})] \leq f(\bar{\theta}^t) - \frac{\eta}{2} \left\| \nabla f(\bar{\theta}^t) \right\|^2 + \frac{L\eta^2}{2n^2} \left( n\sigma^2 + 2nL^2 \left\| \Theta^t - \bar{\Theta}^t \right\|_F^2 + 2n^2 \left\| \nabla f(\bar{\theta}^t) \right\|^2 \right) + \frac{\eta L^2}{2n} \left\| \Theta^t - \bar{\Theta}^t \right\|_F^2$$

$$= f(\bar{\theta}^t) - \eta(\frac{1}{2} - L\eta) \left\| \nabla f(\bar{\theta}^t) \right\|^2 + \left( \frac{L^3\eta^2}{n} + \frac{L^2\eta}{2n} \right) \left\| \Theta^t - \bar{\Theta}^t \right\|_F^2 + \frac{L\eta^2\sigma^2}{2n}$$

$$\overset{(\eta \leq \frac{1}{4L})}{\leq} f(\bar{\theta}^t) - \frac{\eta}{4} \left\| \nabla f(\bar{\theta}^t) \right\|^2 + \frac{3L^2\eta}{4n} \left\| \Theta^t - \bar{\Theta}^t \right\|_F^2 + \frac{L\eta^2\sigma^2}{2n} \tag{64}$$

The proof is completed.

## B.2 Proof of Lemma B.2

We preface the proof by stating two lemmas that will be instrumental to the proof of Lemma B.2. Their proofs can be found in the later part of this subsection.

**Lemma B.6.** *Under Assumption 2.3. For any $t \geq 0$, it holds:*

$$\mathbb{E}\left[ \left\| \nabla \widehat{F}^{t+1} - \nabla \widehat{F}^t \right\|_F^2 \right] \leq 2L^2 \mathbb{E}\left[ \left\| \Theta^{t+1} - \Theta^t \right\|_F^2 \right] + 3n\sigma^2. \tag{65}$$

**Lemma B.7.** *Under Assumption 2.1, 2.2, 2.3, 2.4. For any $t \geq 0$, it holds*

$$\mathbb{E}_t\left[ \left\| \Theta^{t+1} - \Theta^t \right\|_F^2 \right] \leq 4\eta^2(1 - \rho\gamma)^2 \left\| G_o^t \right\|_F^2 + 4\left( \eta^2(1 - \rho\gamma)^2 L^2 + \bar{\omega}^2\gamma^2 \right) \left\| \Theta_o^t \right\|_F^2 + 2\bar{\omega}^2\gamma^2(1 - \delta) \left\| \Theta^t - \eta G^t - \hat{\Theta}^t \right\|_F^2$$

$$+ 2\eta^2(1 - \rho\gamma)^2\sigma^2 + 4n\eta^2(1 - \rho\gamma)^2 \left\| \nabla f(\bar{\theta}^t) \right\|^2. \tag{66}$$

*Proof of Lemma B.2.* We begin by observing the update for $G_o^{t+1}$ as:

$$
\begin{aligned}
G_o^{t+1} &= \mathbf{U}^\top [G^t + \nabla \widehat{F}^{t+1} - \nabla \widehat{F}^t + \gamma(\mathbf{W} - \mathbf{I})\hat{G}^{t+1}] \\
&= \mathbf{U}^\top \left[ G^t + \nabla \widehat{F}^{t+1} - \nabla \widehat{F}^t + \gamma(\mathbf{W} - \mathbf{I})(\hat{G}^t + \mathcal{Q}(G^t + \nabla \widehat{F}^{t+1} - \nabla \widehat{F}^t - \hat{G}^t)) \right] \\
&= \mathbf{U}^\top \left[ (I + \gamma(\mathbf{W} - \mathbf{I}))(G^t + \nabla \widehat{F}^{t+1} - \nabla \widehat{F}^t) \right] \\
&\quad + \gamma \mathbf{U}^\top(\mathbf{W} - \mathbf{I}) \left[ \mathcal{Q}(G^t + \nabla \widehat{F}^{t+1} - \nabla \widehat{F}^t - \hat{G}^t) - (G^t + \nabla \widehat{F}^{t+1} - \nabla \widehat{F}^t - \hat{G}^t) \right]
\end{aligned}
$$

The above implies that

$$
\begin{aligned}
\mathbb{E}_t \left[ \left\| G_o^{t+1} \right\|_F^2 \right] &\le (1 + \alpha_0)(1 - \rho\gamma)^2 \mathbb{E}_t \left[ \left\| G_o^t + \mathbf{U}^\top(\nabla \widehat{F}^{t+1} - \nabla \widehat{F}^t) \right\|_F^2 \right] \\
&\quad + (1 + \alpha_0^{-1})\gamma^2 \bar{\omega}^2 (1 - \delta) \mathbb{E}_t \left[ \left\| G^t + \nabla \widehat{F}^{t+1} - \nabla \widehat{F}^t - \hat{G}^t \right\|_F^2 \right] \\
&\le (1 + \alpha_0)(1 - \rho\gamma)^2 \mathbb{E}_t \left[ (1 + \alpha_1) \left\| G_o^t \right\|_F^2 + (1 + \alpha_1^{-1}) \left\| \nabla \widehat{F}^{t+1} - \nabla \widehat{F}^t \right\|_F^2 \right] \\
&\quad + 2(1 + \alpha_0^{-1})\gamma^2 \bar{\omega}^2 (1 - \delta) \mathbb{E}_t \left[ \left\| G^t - \hat{G}^t \right\|_F^2 + \left\| \nabla \widehat{F}^{t+1} - \nabla \widehat{F}^t \right\|_F^2 \right]
\end{aligned}
$$

Taking $\alpha_0 = \frac{\rho\gamma}{1 - \rho\gamma}$, $\alpha_1 = \frac{\rho\gamma}{2}$ gives

$$
\begin{aligned}
\mathbb{E}_t \left[ \left\| G_o^{t+1} \right\|_F^2 \right] &\le \left( 1 - \frac{\rho\gamma}{2} \right) \left\| G_o^t \right\|_F^2 + \gamma \frac{2\bar{\omega}^2}{\rho} \left\| G^t - \hat{G}^t \right\|_F^2 \\
&\quad + \frac{2}{\rho\gamma} \left( 1 + \gamma^2 \bar{\omega}^2 \right) \mathbb{E}_t \left[ \left\| \nabla \widehat{F}^{t+1} - \nabla \widehat{F}^t \right\|_F^2 \right]
\end{aligned}
$$

Taking the full expectation and applying Lemma B.6 give

$$
\begin{aligned}
\mathbb{E} \left[ \left\| G_o^{t+1} \right\|_F^2 \right] &\le \left( 1 - \frac{\rho\gamma}{2} \right) \mathbb{E} \left[ \left\| G_o^t \right\|_F^2 \right] + \gamma \frac{2\bar{\omega}^2}{\rho} \mathbb{E} \left[ \left\| G^t - \hat{G}^t \right\|_F^2 \right] \\
&\quad + \frac{2}{\rho\gamma} \left( 1 + \gamma^2 \bar{\omega}^2 \right) \left( 3n\sigma^2 + 2L^2 \mathbb{E} \left[ \left\| \Theta^{t+1} - \Theta^t \right\|_F^2 \right] \right)
\end{aligned}
$$

Furthermore, applying Lemma B.7 yields

$$
\begin{aligned}
\mathbb{E} \left[ \left\| G_o^{t+1} \right\|_F^2 \right] &\le \left( 1 - \frac{\rho\gamma}{2} + \eta^2 \frac{16L^2(1 + \gamma^2 \bar{\omega}^2)(1 - \rho\gamma)^2}{\rho\gamma} \right) \mathbb{E} \left[ \left\| G_o^t \right\|_F^2 \right] + \gamma \frac{2\bar{\omega}^2}{\rho} \mathbb{E} \left[ \left\| G^t - \hat{G}^t \right\|_F^2 \right] \\
&\quad + \frac{16L^2(1 + \gamma^2 \bar{\omega}^2)}{\rho\gamma}(\eta^2 L^2 (1 - \rho\gamma)^2 + \bar{\omega}^2 \gamma^2) \mathbb{E} \left[ \left\| \Theta_o^t \right\|_F^2 \right] \\
&\quad + \frac{8L^2(1 + \gamma^2 \bar{\omega}^2)}{\rho\gamma} \bar{\omega}^2 \gamma^2 (1 - \delta) \mathbb{E} \left[ \left\| \Theta^t - \eta G^t - \hat{\Theta}^t \right\|_F^2 \right] \\
&\quad + \frac{16L^2(1 + \gamma^2 \bar{\omega}^2)}{\rho\gamma}(1 - \rho\gamma)^2 n \, \eta^2 \, \mathbb{E} \left[ \left\| \nabla f(\bar{\theta}^t) \right\|^2 \right] \\
&\quad + \left( \frac{6}{\rho\gamma} \left( 1 + \gamma^2 \bar{\omega}^2 \right) n + \frac{8}{\rho\gamma} \eta^2 (1 + \gamma^2 \bar{\omega}^2) L^2 (1 - \rho\gamma)^2 \right) \sigma^2
\end{aligned}
$$

Using the step size condition

$$
\eta \le \frac{\rho\gamma}{8L(1 - \rho\gamma)\sqrt{1 + \gamma^2 \bar{\omega}^2}}, \quad \gamma \le \frac{1}{8\bar{\omega}} \tag{67}
$$

implies that

$$
\mathbb{E} \left[ \left\| G_o^{t+1} \right\|_F^2 \right] \le \left( 1 - \frac{\rho\gamma}{4} \right) \mathbb{E} \left[ \left\| G_o^t \right\|_F^2 \right] + \gamma \frac{2\bar{\omega}^2}{\rho} \mathbb{E} \left[ \left\| G^t - \hat{G}^t \right\|_F^2 \right] + \gamma \left( \frac{\rho}{4} + \frac{18L^2 \bar{\omega}^2}{\rho} \right) \mathbb{E} \left[ \left\| \Theta_o^t \right\|_F^2 \right]
$$

$$+ \gamma \frac{9L^2}{\rho} \bar{\omega}^2 (1-\delta) \mathbb{E}\left[\left\|\Theta^t - \eta G^t - \hat{\Theta}^t\right\|_F^2\right] + \gamma \frac{\rho n}{4} \mathbb{E}\left[\left\|\nabla f(\bar{\theta}^t)\right\|^2\right] + \left(\frac{7n}{\rho\gamma} + \frac{\rho\gamma}{8}\right)\sigma^2.$$

This concludes our proof. $\qquad\square$

### B.2.1 Proofs for the Auxilliary Lemmas

**Proof of Lemma B.6**  Observe that:

$$\mathbb{E}\left[\left\|\nabla\widehat{F}^{t+1} - \nabla\widehat{F}^t\right\|_F^2\right] = \mathbb{E}\left[\left\|(\nabla\widehat{F}^{t+1} - \nabla F^{t+1}) - (\nabla\widehat{F}^t - \nabla F^t)\right\|_F^2 + \left\|\nabla F^{t+1} - \nabla F^t\right\|_F^2\right]$$
$$+ 2\mathbb{E}\left[\left\langle (\nabla\widehat{F}^{t+1} - \nabla F^{t+1}) - (\nabla\widehat{F}^t - \nabla F^t) \,\middle|\, \nabla F^{t+1} - \nabla F^t\right\rangle\right]$$

Notice that

$$\mathbb{E}\left[\left\langle (\nabla\widehat{F}^{t+1} - \nabla F^{t+1}) - (\nabla\widehat{F}^t - \nabla F^t) \,\middle|\, \nabla F^{t+1} - \nabla F^t\right\rangle\right] = -\mathbb{E}\left[\left\langle \nabla\widehat{F}^t - \nabla F^t \,\middle|\, \nabla F^{t+1} - \nabla F^t\right\rangle\right]$$

As such,

$$\mathbb{E}\left[\left\|\nabla\widehat{F}^{t+1} - \nabla\widehat{F}^t\right\|_F^2\right] \leq 3n\sigma^2 + 2L^2\mathbb{E}\left[\left\|\Theta^{t+1} - \Theta^t\right\|_F^2\right].$$

This concludes the proof. $\qquad\square$

**Proof of Lemma B.7**  Observe that:

$$\left\|\Theta^{t+1} - \Theta^t\right\|_F^2 = \left\|\eta G^t - \gamma(\mathbf{W}-\mathbf{I})\hat{\Theta}^{t+1}\right\|_F^2$$
$$= \left\|(I + \gamma(\mathbf{W}-\mathbf{I}))(-\eta G^t) + \gamma(\mathbf{W}-\mathbf{I})\Theta^t + \gamma(\mathbf{W}-\mathbf{I})\left[\mathcal{Q}(\Theta^t - \eta G^t - \hat{\Theta}^t) - (\Theta^t - \eta G^t - \hat{\Theta}^t)\right]\right\|_F^2$$
$$\leq 2\left\|(I + \gamma(\mathbf{W}-\mathbf{I}))(-\eta G^t) + \gamma(\mathbf{W}-\mathbf{I})\Theta^t\right\|_F^2$$
$$+ 2\left\|\gamma(\mathbf{W}-\mathbf{I})\left[\mathcal{Q}(\Theta^t - \eta G^t - \hat{\Theta}^t) - (\Theta^t - \eta G^t - \hat{\Theta}^t)\right]\right\|_F^2$$
$$\leq 2\left[\eta^2\left\|(I + \gamma(\mathbf{W}-\mathbf{I}))G^t\right\|_F^2 + \gamma^2\left\|(\mathbf{W}-\mathbf{I})\Theta^t\right\|_F^2 + 2\left\langle(I + \gamma(\mathbf{W}-\mathbf{I}))(-\eta G^t) \,\middle|\, \gamma(\mathbf{W}-\mathbf{I})\Theta^t\right\rangle\right]$$
$$+ 2\bar{\omega}^2\gamma^2\left\|\mathcal{Q}(\Theta^t - \eta G^t - \hat{\Theta}^t) - (\Theta^t - \eta G^t - \hat{\Theta}^t)\right\|_F^2 \tag{68}$$

Observe that $\langle(I + \gamma(\mathbf{W}-\mathbf{I}))(-\eta G^t) \mid \gamma(\mathbf{W}-\mathbf{I})\Theta^t\rangle = \langle(I + \gamma(\mathbf{W}-\mathbf{I}))(-\eta\mathbf{U}G_o^t) \mid \gamma(\mathbf{W}-\mathbf{I})\Theta^t\rangle$, the above leads to

$$\mathbb{E}_t\left[\left\|\Theta^{t+1} - \Theta^t\right\|_F^2\right] \leq 2\eta^2(1-\rho\gamma)^2\left\|G^t\right\|_F^2 + 2\eta^2(1-\rho\gamma)^2\left\|G_o^t\right\|_F^2 + 4\gamma^2\left\|(\mathbf{W}-\mathbf{I})\Theta^t\right\|_F^2$$
$$+ 2\bar{\omega}^2\gamma^2(1-\delta)\left\|\Theta^t - \eta G^t - \hat{\Theta}^t\right\|_F^2 \tag{69}$$

Notice that using (63), we obtain

$$\mathbb{E}_t\left[\left\|G^t\right\|_F^2\right] = \mathbb{E}_t\left[\left\|((1/n)\mathbf{1}\mathbf{1}^\top + \mathbf{U}\mathbf{U}^\top)G^t\right\|_F^2\right] = \mathbb{E}_t\left[\left\|\mathbf{U}\mathbf{U}^\top G^t\right\|_F^2 + \left\|(1/n)\mathbf{1}\mathbf{1}^\top G^t\right\|_F^2\right]$$
$$\leq \left\|G_o^t\right\|_F^2 + 2L^2\left\|\Theta_o^t\right\|_F^2 + 2n\left\|\nabla f(\bar{\theta}^t)\right\|^2 + \sigma^2 \tag{70}$$

By combining (69), (70), and the fact $\|(\mathbf{W}-\mathbf{I})\Theta^t\|_F^2 \leq \bar{\omega}^2\|\Theta_o^t\|_F^2$, we have

$$\mathbb{E}_t\left[\left\|\Theta^{t+1} - \Theta^t\right\|_F^2\right] \leq 4\eta^2(1-\rho\gamma)^2\left\|G_o^t\right\|_F^2 + 4\eta^2(1-\rho\gamma)^2 L^2\left\|\Theta_o^t\right\|_F^2 + 4n\eta^2(1-\rho\gamma)^2\left\|\nabla f(\bar{\theta}^t)\right\|^2$$
$$+ 2\eta^2(1-\rho\gamma)^2\sigma^2 + 4\bar{\omega}^2\gamma^2\left\|\Theta_o^t\right\|_F^2 + 2\bar{\omega}^2\gamma^2(1-\delta)\left\|\Theta^t - \eta G^t - \hat{\Theta}^t\right\|_F^2$$

This concludes the proof. $\qquad\square$

## B.3   Proof of Lemma B.3

We begin by observing the following recursion for $G^t - \hat{G}^t$:

$$
\begin{aligned}
G^{t+1} - \hat{G}^{t+1} &= G^t + \nabla\widehat{F}^{t+1} - \nabla\widehat{F}^t + (\gamma(\mathbf{W} - \mathbf{I}) - \mathbf{I})\hat{G}^{t+1} \\
&= \gamma(\mathbf{W} - \mathbf{I})(G^t + \nabla\widehat{F}^{t+1} - \nabla\widehat{F}^t) \\
&\quad + (\gamma(\mathbf{W} - \mathbf{I}) - \mathbf{I})\left[ \mathcal{Q}(G^t + \nabla\widehat{F}^{t+1} - \nabla\widehat{F}^t - \hat{G}^t) - (G^t + \nabla\widehat{F}^{t+1} - \nabla\widehat{F}^t - \hat{G}^t) \right].
\end{aligned}
$$

This implies

$$
\begin{aligned}
\mathbb{E}\left[ \left\| G^{t+1} - \hat{G}^{t+1} \right\|_F^2 \right] &\leq (1 + \alpha_0)(1 + \gamma\bar{\omega})^2(1 - \delta)\mathbb{E}\left[ (1 + \alpha_1)\left\| G^t - \hat{G}^t \right\|_F^2 + (1 + \alpha_1^{-1})\left\| \nabla\widehat{F}^{t+1} - \nabla\widehat{F}^t \right\|_F^2 \right] \\
&\quad + 2(1 + \alpha_0^{-1})\gamma^2\bar{\omega}^2\mathbb{E}\left[ \left\| G_o^t \right\|_F^2 + \left\| \nabla\widehat{F}^{t+1} - \nabla\widehat{F}^t \right\|_F^2 \right]
\end{aligned}
$$

Taking $\alpha_0 = \frac{\delta}{4}$, $\alpha_1 = \frac{\delta}{8}$ and the step size condition

$$
\gamma \leq \frac{\delta}{8\bar{\omega}} \leq \frac{\delta}{6\bar{\omega}}
$$

give

$$
\begin{aligned}
\mathbb{E}\left[ \left\| G^{t+1} - \hat{G}^{t+1} \right\|_F^2 \right] &\leq \left(1 - \frac{\delta}{8}\right)\mathbb{E}\left[ \left\| G^t - \hat{G}^t \right\|_F^2 \right] + \frac{10\gamma^2\bar{\omega}^2}{\delta}\mathbb{E}\left[ \left\| G_o^t \right\|_F^2 \right] \\
&\quad + \left( (1 - \frac{\delta}{4})(1 + \frac{8}{\delta}) + 2\gamma^2\bar{\omega}^2(1 + \frac{4}{\delta}) \right)\mathbb{E}\left[ \left\| \nabla\widehat{F}^{t+1} - \nabla\widehat{F}^t \right\|_F^2 \right] \\
&\leq \left(1 - \frac{\delta}{8}\right)\mathbb{E}\left[ \left\| G^t - \hat{G}^t \right\|_F^2 \right] + \frac{10\gamma^2\bar{\omega}^2}{\delta}\mathbb{E}\left[ \left\| G_o^t \right\|_F^2 \right] + \frac{10(1 + \gamma^2\bar{\omega}^2)}{\delta}\mathbb{E}\left[ \left\| \nabla\widehat{F}^{t+1} - \nabla\widehat{F}^t \right\|_F^2 \right]
\end{aligned}
$$

Applying Lemma B.6 and Lemma B.7 gives

$$
\begin{aligned}
\mathbb{E}\left[ \left\| G^{t+1} - \hat{G}^{t+1} \right\|_F^2 \right] &\leq \left(1 - \frac{\delta}{8}\right)\mathbb{E}\left[ \left\| G^t - \hat{G}^t \right\|_F^2 \right] + \frac{10\gamma^2\bar{\omega}^2}{\delta}\mathbb{E}\left[ \left\| G_o^t \right\|_F^2 \right] + \frac{30(1 + \gamma^2\bar{\omega}^2)}{\delta}n\sigma^2 \\
&\quad + \frac{20(1 + \gamma^2\bar{\omega}^2)L^2}{\delta}\mathbb{E}\left[ \left\| \Theta^{t+1} - \Theta^t \right\|_F^2 \right] \\
&\leq \left(1 - \frac{\delta}{8}\right)\mathbb{E}\left[ \left\| G^t - \hat{G}^t \right\|_F^2 \right] + \frac{10}{\delta}\left(\gamma^2\bar{\omega}^2 + 8\eta^2 L^2(1 + \gamma^2\bar{\omega}^2)(1 - \rho\gamma)^2\right)\mathbb{E}\left[ \left\| G_o^t \right\|_F^2 \right] \\
&\quad + \frac{80(1 + \gamma^2\bar{\omega}^2)L^2}{\delta}(\eta^2 L^2(1 - \rho\gamma)^2 + \bar{\omega}^2\gamma^2)\mathbb{E}\left[ \left\| \Theta_o^t \right\|_F^2 \right] \\
&\quad + \frac{40(1 + \gamma^2\bar{\omega}^2)L^2}{\delta}\bar{\omega}^2\gamma^2(1 - \delta)\mathbb{E}\left[ \left\| \Theta^t - \eta G^t - \hat{G}^t \right\|_F^2 \right] \\
&\quad + \frac{80(1 + \gamma^2\bar{\omega}^2)L^2}{\delta}n\eta^2(1 - \rho\gamma)^2\mathbb{E}\left[ \left\| \nabla f(\bar{\theta}^t) \right\|^2 \right] \\
&\quad + \frac{10(1 + \gamma^2\bar{\omega}^2)}{\delta}\left(4L^2\eta^2(1 - \rho\gamma)^2\sigma^2 + 3n\sigma^2\right)
\end{aligned}
$$

Using the step size condition from (67), i.e., $\eta^2 L^2(1 - \rho\gamma)^2(1 + \gamma^2\bar{\omega}^2) \leq \frac{\rho^2\gamma^2}{64}$, simplifies the above to

$$
\begin{aligned}
\mathbb{E}\left[ \left\| G^{t+1} - \hat{G}^{t+1} \right\|_F^2 \right] &\leq \left(1 - \frac{\delta}{8}\right)\mathbb{E}\left[ \left\| G^t - \hat{G}^t \right\|_F^2 \right] + \frac{10}{\delta}\gamma^2\left(\bar{\omega}^2 + \frac{\rho^2}{8}\right)\mathbb{E}\left[ \left\| G_o^t \right\|_F^2 \right] \\
&\quad + \gamma^2\frac{5}{4\delta}\left(\rho^2 + 72L^2\bar{\omega}^2\right)\mathbb{E}\left[ \left\| \Theta_o^t \right\|_F^2 \right] + \gamma^2\frac{40L^2\bar{\omega}^2}{\delta}\mathbb{E}\left[ \left\| \Theta^t - \eta G^t - \hat{G}^t \right\|_F^2 \right] \\
&\quad + \gamma^2\frac{5\rho^2}{4\delta}n\mathbb{E}\left[ \left\| \nabla f(\bar{\theta}^t) \right\|^2 \right] + \frac{5}{\delta}\left(\frac{\rho^2\gamma^2}{8} + 7n\right)\sigma^2
\end{aligned}
$$

This concludes our proof.

## B.4 Proof of Lemma B.4

Observe that

$$
\mathbb{E}_t\left[\left\|\Theta^{t+1} - \eta G^{t+1} - \hat{\Theta}^{t+1}\right\|_F^2\right] = \mathbb{E}_t\left[\left\|\Theta^{t+1} - \eta G^{t+1} - (\hat{\Theta}^t + \mathcal{Q}(\Theta^t - \eta G^t - \hat{\Theta}^t))\right\|_F^2\right]
$$

$$
= \mathbb{E}_t\left[\left\|\Theta^{t+1} - \eta G^{t+1} - (\Theta^t - \eta G^t) + (\Theta^t - \eta G^t - \hat{\Theta}^t) - \mathcal{Q}(\Theta^t - \eta G^t - \hat{\Theta}^t)\right\|_F^2\right]
$$

$$
\leq (1 + \frac{2}{\delta})\mathbb{E}_t\left[\left\|\Theta^{t+1} - \Theta^t - \eta(G^{t+1} - G^t)\right\|_F^2\right] + (1 + \frac{\delta}{2})(1 - \delta)\left\|\Theta^t - \eta G^t - \hat{\Theta}^t\right\|_F^2
$$

$$
\leq 2(1 + \frac{2}{\delta})\mathbb{E}_t\left[\left\|\Theta^{t+1} - \Theta^t\right\|_F^2\right] + 2\eta^2(1 + \frac{2}{\delta})\mathbb{E}_t\left[\left\|G^{t+1} - G^t\right\|_F^2\right] + (1 - \frac{\delta}{2})\left\|\Theta^t - \eta G^t - \hat{\Theta}^t\right\|_F^2 \tag{71}
$$

We can bound the second term as

$$
\mathbb{E}\left[\left\|G^{t+1} - G^t\right\|_F^2\right] = \frac{1}{n}\mathbb{E}\left[\left\|\mathbf{1}^\top(\nabla\widehat{F}^{t+1} - \nabla\widehat{F}^t)\right\|^2\right] + 2\mathbb{E}\left[\left\|G_o^{t+1}\right\|_F^2\right] + 2\mathbb{E}\left[\left\|G_o^t\right\|_F^2\right]
$$

Observe that

$$
\mathbb{E}\left[\left\|\mathbf{1}^\top(\nabla\widehat{F}^{t+1} - \nabla\widehat{F}^t)\right\|^2\right] \leq \mathbb{E}\left[\left\|\mathbf{1}^\top(\nabla\widehat{F}^{t+1} - \nabla F^{t+1})\right\|^2\right] + 2\mathbb{E}\left[\left\|\mathbf{1}^\top(\nabla\widehat{F}^t - \nabla F^t)\right\|^2\right]
$$

$$
+ 2\mathbb{E}\left[\left\|\mathbf{1}^\top(\nabla F^{t+1} - \nabla F^t)\right\|^2\right]
$$

$$
\leq 3n\sigma^2 + 2nL^2\mathbb{E}\left[\left\|\Theta^{t+1} - \Theta^t\right\|_F^2\right]
$$

Substituting back into (71) yields

$$
\mathbb{E}\left[\left\|\Theta^{t+1} - \eta G^{t+1} - \hat{\Theta}^{t+1}\right\|_F^2\right] \leq (1 - \frac{\delta}{2})\mathbb{E}\left[\left\|\Theta^t - \eta G^t - \hat{\Theta}^t\right\|_F^2\right] + 4\eta^2(1 + \frac{2}{\delta})\mathbb{E}\left[\left\|G_o^{t+1}\right\|_F^2 + \left\|G_o^t\right\|_F^2\right]
$$

$$
+ 2(1 + \frac{2}{\delta})(\eta^2 L^2 + 1)\mathbb{E}\left[\left\|\Theta^{t+1} - \Theta^t\right\|_F^2\right] + 6\sigma^2(1 + \frac{2}{\delta})\eta^2 \tag{72}
$$

We further apply Lemma B.7 to obtain

$$
\mathbb{E}\left[\left\|\Theta^{t+1} - \eta G^{t+1} - \hat{\Theta}^{t+1}\right\|_F^2\right] \leq (1 - \frac{\delta}{2})\mathbb{E}\left[\left\|\Theta^t - \eta G^t - \hat{\Theta}^t\right\|_F^2\right] + 4\eta^2(1 + \frac{2}{\delta})\mathbb{E}\left[\left\|G_o^{t+1}\right\|_F^2 + \left\|G_o^t\right\|_F^2\right]
$$

$$
+ 6\sigma^2(1 + \frac{2}{\delta})\eta^2 + 4\bar{\omega}^2(1 + \eta^2 L^2)(1 + \frac{2}{\delta})\gamma^2(1 - \delta)\mathbb{E}\left[\left\|\Theta^t - \eta G^t - \hat{\Theta}^t\right\|_F^2\right]
$$

$$
+ 8(1 + \eta^2 L^2)(1 + \frac{2}{\delta})\eta^2(1 - \rho\gamma)^2\mathbb{E}\left[\left\|G_o^t\right\|_F^2 + L^2\left\|\Theta_o^t\right\|_F^2 + n\left\|\nabla f(\bar{\theta}^t)\right\|^2 + \sigma^2/2\right]
$$

$$
+ 8(1 + \eta^2 L^2)(1 + \frac{2}{\delta})\bar{\omega}^2\gamma^2\mathbb{E}\left[\left\|\Theta_o^t\right\|_F^2\right] \tag{73}
$$

Using the step size condition:

$$
\gamma^2 \leq \frac{\delta}{16\bar{\omega}^2(1 - \delta)(1 + \eta^2 L^2)(1 + 2/\delta)}
$$

and we recall that $\eta \leq 1/(4L)$, the upper bound in (73) can be simplified as

$$
\mathbb{E}\left[\left\|\Theta^{t+1} - \eta G^{t+1} - \hat{\Theta}^{t+1}\right\|_F^2\right] \leq (1 - \frac{\delta}{4})\mathbb{E}\left[\left\|\Theta^t - \eta G^t - \hat{\Theta}^t\right\|_F^2\right] + \eta^2\frac{12}{\delta}\mathbb{E}\left[\left\|G_o^{t+1}\right\|_F^2\right]
$$

$$
+ \frac{24}{\delta}\eta^2\sigma^2 + \frac{38}{\delta}\eta^2\mathbb{E}\left[\left\|G_o^t\right\|_F^2\right] + \frac{26}{\delta}\mathbb{E}\left[(\eta^2 L^2 + \bar{\omega}^2\gamma^2)\left\|\Theta_o^t\right\|_F^2 + \eta^2 n\left\|\nabla f(\bar{\theta}^t)\right\|^2\right]
$$

This concludes the proof of the first part.

The above bound can be combined with Lemma B.2 and $\gamma\rho \leq 1$ to give

$$
\mathbb{E}\left[\left\|\Theta^{t+1} - \eta G^{t+1} - \hat{\Theta}^{t+1}\right\|_F^2\right] \leq \left(1 - \frac{\delta}{4} + \eta^2\gamma\frac{108\bar{\omega}^2 L^2}{\rho\delta}\right)\mathbb{E}\left[\left\|\Theta^t - \eta G^t - \hat{\Theta}^t\right\|_F^2\right]
$$
$$
+ \eta^2\frac{50}{\delta}\mathbb{E}\left[\left\|G_o^t\right\|_F^2\right] + \eta^2\gamma\frac{24\bar{\omega}^2}{\rho}\mathbb{E}\left[\left\|G^t - \hat{G}^t\right\|_F^2\right]
$$
$$
+ \left[\frac{\eta^2 L^2}{\delta}\left(3 + 26L^2 + \frac{216\gamma L^2\bar{\omega}^2}{\rho}\right) + \frac{26}{\delta}\bar{\omega}^2\gamma^2\right]\mathbb{E}\left[\left\|\Theta_o^t\right\|_F^2\right]
$$
$$
+ \frac{29\eta^2}{\delta}n\,\mathbb{E}\left[\left\|\nabla f(\bar{\theta}^t)\right\|^2\right] + \frac{24\eta^2\sigma^2}{\delta}\left(1 + \frac{4n}{\rho\gamma}\right)
$$

Taking $\eta^2\gamma \leq \frac{\delta^2\rho}{864\bar{\omega}^2 L^2}$ simplifies the bound into

$$
\mathbb{E}\left[\left\|\Theta^{t+1} - \eta G^{t+1} - \hat{\Theta}^{t+1}\right\|_F^2\right] \leq \left(1 - \frac{\delta}{8}\right)\mathbb{E}\left[\left\|\Theta^t - \eta G^t - \hat{\Theta}^t\right\|_F^2\right]
$$
$$
+ \eta^2\frac{50}{\delta}\mathbb{E}\left[\left\|G_o^t\right\|_F^2\right] + \eta^2\frac{3\bar{\omega}}{\rho}\mathbb{E}\left[\left\|G^t - \hat{G}^t\right\|_F^2\right]
$$
$$
+ \left[\eta^2\frac{29L^4}{\delta}\left(1 + \frac{\bar{\omega}}{\rho}\right) + \frac{26}{\delta}\bar{\omega}^2\gamma^2\right]\mathbb{E}\left[\left\|\Theta_o^t\right\|_F^2\right]
$$
$$
+ \eta^2\frac{29n}{\delta}\mathbb{E}\left[\left\|\nabla f(\bar{\theta}^t)\right\|^2\right] + \eta^2\frac{24\sigma^2}{\delta}\left(1 + \frac{4n}{\rho\gamma}\right)
$$

This concludes the proof.

## B.5 Proof of Lemma B.5

Let $a, b, c > 0$ be some constants to be determined later, combining Lemma 3.5, B.2, B.3, B.4 yields

$$
\mathbb{E}\left[\left\|\Theta_o^{t+1}\right\|_F^2 + a\left\|G_o^{t+1}\right\|_F^2 + b\left\|G^{t+1} - \hat{G}^{t+1}\right\|_F^2 + c\left\|\Theta^{t+1} - \eta G^{t+1} - \hat{\Theta}^{t+1}\right\|_F^2\right]
$$
$$
\leq \left(1 - \frac{\rho\gamma}{2} + a\gamma(\frac{\rho}{4} + \frac{18L^2\bar{\omega}^2}{\rho}) + b\gamma^2\frac{2}{\delta}(\rho^2 + 45L^2\bar{\omega}^2) + \frac{c}{\delta}\left[29L^4(1 + \frac{\bar{\omega}}{\rho})\eta^2 + 26\bar{\omega}^2\gamma^2\right]\right)\mathbb{E}\left[\left\|\Theta_o^t\right\|_F^2\right]
$$
$$
+ a\left(1 - \frac{\rho\gamma}{4} + \frac{1}{a}\eta^2\frac{2}{\rho\gamma} + \frac{b}{a}\gamma^2\frac{10}{\delta^2}(\delta\bar{\omega}^2 + \rho^2) + \frac{c}{a}\eta^2\frac{50}{\delta}\right)\mathbb{E}\left[\left\|G_o^t\right\|_F^2\right]
$$
$$
+ b\left(1 - \frac{\delta}{8} + \frac{a}{b}\gamma\frac{2\bar{\omega}^2}{\rho} + \frac{c}{b}\eta^2\frac{3\bar{\omega}^2}{\rho}\right)\mathbb{E}\left[\left\|G^t - \hat{G}^t\right\|_F^2\right]
$$
$$
+ c\left(1 - \frac{\delta}{8} + \frac{1}{c}\gamma\frac{\bar{\omega}^2}{\rho} + \frac{a}{c}\gamma\frac{9L^2\bar{\omega}^2}{\rho} + \frac{b}{c}\gamma^2\frac{40L^2\bar{\omega}^2}{\delta}\right)\mathbb{E}\left[\left\|\Theta^t - \eta G^t - \hat{\Theta}^t\right\|_F^2\right]
$$
$$
+ 8n\sigma^2\left[a\frac{1}{\rho\gamma} + b\frac{5}{\delta} + c\eta^2\frac{15}{\rho\delta\gamma}\right] + \left[a\gamma\frac{\rho}{4} + b\gamma^2\frac{2\rho^2}{\delta} + c\eta^2\frac{29}{\delta}\right]n\mathbb{E}\left[\left\|\nabla f(\bar{\theta}^t)\right\|^2\right]
$$

We wish to find conditions on the step sizes and $a, b, c$ such that

$$
1 - \frac{\rho\gamma}{2} + a\gamma\frac{19L^2\bar{\omega}^2}{\rho} + b\gamma^2\frac{92L^2\bar{\omega}^2}{\delta} + \frac{c}{\delta}\left[\frac{58L^4\bar{\omega}}{\rho}\eta^2 + 26\bar{\omega}^2\gamma^2\right] \leq 1 - \frac{\rho\gamma}{4},
$$
$$
1 - \frac{\rho\gamma}{4} + \frac{1}{a}\eta^2\frac{2}{\rho\gamma} + \frac{b}{a}\gamma^2\frac{10}{\delta^2}(\delta\bar{\omega}^2 + \rho^2) + \frac{c}{a}\eta^2\frac{50}{\delta} \leq 1 - \frac{\rho\gamma}{8},
$$
$$
1 - \frac{\delta}{8} + \frac{a}{b}\gamma\frac{2\bar{\omega}^2}{\rho} + \frac{c}{b}\eta^2\frac{3\bar{\omega}^2}{\rho} \leq 1 - \frac{\delta\gamma}{8},
$$
$$
1 - \frac{\delta}{8} + \frac{1}{c}\gamma\frac{\bar{\omega}^2}{\rho} + \frac{a}{c}\gamma\frac{9L^2\bar{\omega}^2}{\rho} + \frac{b}{c}\gamma^2\frac{40L^2\bar{\omega}^2}{\delta} \leq 1 - \frac{\delta\gamma}{8}.
$$

The above set of inequalities can be guaranteed if $a, b, c$ satisfy

$$\frac{48}{\rho^2\gamma^2}\eta^2 \le a \le \frac{\rho^2}{228L^2\bar{\omega}^2} \tag{74}$$

$$\max\left\{a\frac{\gamma}{1-\gamma}\frac{32\bar{\omega}^2}{\rho\delta}, c\frac{\eta^2}{1-\gamma}\frac{48\bar{\omega}^2}{\rho\delta}\right\} \le b \le \min\left\{\frac{1}{\gamma}\frac{\rho\delta}{1104L^2\bar{\omega}^2}, \frac{\eta^2}{\gamma^3}\frac{\delta^2}{5\rho(\delta\bar{\omega}^2+\rho^2)}\right\} \tag{75}$$

$$\max\left\{\frac{\gamma}{1-\gamma}\frac{24\bar{\omega}^2}{\rho\delta}, a\frac{\gamma}{1-\gamma}\frac{216L^2\bar{\omega}^2}{\rho\delta}, b\frac{\gamma^2}{1-\gamma}\frac{960L^2\bar{\omega}^2}{\rho\delta}\right\} \le c \le \min\left\{\frac{\rho\delta\gamma}{12(\frac{58L^4\bar{\omega}}{\rho}\eta^2+26\bar{\omega}^2\gamma^2)}, \frac{\delta}{25\rho\gamma}\right\} \tag{76}$$

Notice that the step size condition:

$$\eta^2 \le \frac{\rho^4\gamma^2}{10944L^2\bar{\omega}^2}$$

guarantees the existence of $a$ which satisfies (74). In particular, we take $a = \frac{48}{\rho^2\gamma^2}\eta^2$. This simplifies (75), (76) into

$$\frac{48\bar{\omega}^2}{\rho\delta(1-\gamma)}\eta^2\max\left\{\frac{32}{\rho^2\gamma}, c\right\} \le b \le \min\left\{\frac{1}{\gamma}\frac{\rho\delta}{1104L^2\bar{\omega}^2}, \frac{\eta^2}{\gamma^3}\frac{\delta^2}{5\rho(\delta\bar{\omega}^2+\rho^2)}\right\}$$

$$\frac{24\bar{\omega}^2}{\rho\delta(1-\gamma)}\max\left\{\gamma, \frac{432}{\rho^2\gamma}L^2\eta^2, 40L^2\gamma^2 b\right\} \le c \le \min\left\{\frac{\rho\delta\gamma}{12(\frac{58L^4\bar{\omega}}{\rho}\eta^2+26\bar{\omega}^2\gamma^2)}, \frac{\delta}{25\rho\gamma}\right\}$$

Observing that as $\eta^2 \le \frac{\rho^4\gamma^2}{10944L^2\bar{\omega}^2} \le \frac{\rho^2\gamma^2}{432L^2}$ and we impose the extra condition $c \le \frac{32}{\rho^2\gamma}$. We obtain the simplification:

$$\frac{1536\bar{\omega}^2}{\rho^3\delta\gamma(1-\gamma)}\eta^2 \le b \le \min\left\{\frac{1}{\gamma}\frac{\rho\delta}{1104L^2\bar{\omega}^2}, \frac{\eta^2}{\gamma^3}\frac{\delta^2}{5\rho(\delta\bar{\omega}^2+\rho^2)}\right\}$$

$$\frac{24\bar{\omega}^2}{\rho\delta(1-\gamma)}\max\left\{\gamma, 40L^2\gamma^2 b\right\} \le c \le \min\left\{\frac{\rho\delta\gamma}{12(\frac{58L^4\bar{\omega}}{\rho}\eta^2+26\bar{\omega}^2\gamma^2)}, \frac{\delta}{25\rho\gamma}, \frac{32}{\rho^2\gamma}\right\}$$

Again, the condition on $b$ is feasible if

$$\eta^2 \le \frac{\rho^4\delta^2(1-\gamma)}{1536\times 1104\times L^2\bar{\omega}^4}, \quad \frac{\gamma^2}{1-\gamma} \le \frac{\rho^2\delta^3}{7680\bar{\omega}^2(\delta\bar{\omega}^2+\rho^2)}$$

and we take $b = \frac{1536\bar{\omega}^2}{\rho^3\gamma\delta(1-\gamma)}\eta^2$. Note that as $\gamma \le \delta/8\bar{\omega}$, the bound on $\gamma$ can be implied by:

$$\gamma^2 \le \frac{\rho^2\delta^3(1-\delta/8\bar{\omega})}{7680\bar{\omega}^2(\delta\bar{\omega}^2+\rho^2)}$$

Observe that with this choice of $b$ and the step size condition, we have $40L^2\gamma^2 b \le \gamma$. Finally, the condition on $c$ is simplified to

$$\frac{24\bar{\omega}^2}{\rho\delta(1-\gamma)}\gamma \le c \le \min\left\{\frac{\rho\delta\gamma}{12(\frac{58L^4\bar{\omega}}{\rho}\eta^2+26\bar{\omega}^2\gamma^2)}, \frac{\delta}{25\rho\gamma}, \frac{32}{\rho^2\gamma}\right\}$$

The above condition is feasible if

$$\frac{\gamma^2}{1-\gamma} \le \min\left\{\frac{4\delta}{3\rho\bar{\omega}^2}, \frac{\delta^2}{600\bar{\omega}^2}, \frac{\rho^2\delta^2}{14976\bar{\omega}^4}\right\} = \frac{\rho^2\delta^2}{14976\bar{\omega}^4}, \quad \eta^2 \le \frac{\rho^3\delta^2(1-\gamma)}{16704L^4\bar{\omega}^3}$$

and we take $c = \frac{24\bar{\omega}^2}{\rho\delta(1-\gamma)}$.

The above choice of $a, b, c$ ensures that

$$\mathbb{E}\left[\left\|\Theta_o^{t+1}\right\|_F^2 + a\left\|G_o^{t+1}\right\|_F^2 + b\left\|G^{t+1}-\hat{G}^{t+1}\right\|_F^2 + c\left\|\Theta^{t+1}-\eta G^{t+1}-\hat{\Theta}^{t+1}\right\|_F^2\right]$$

$$\leq \left(1 - \min\left\{\frac{\rho}{8}, \frac{\delta}{8}\right\}\gamma\right)\mathbb{E}\left[\left\|\Theta_o^t\right\|_F^2 + a\left\|G_o^t\right\|_F^2 + b\left\|G^t - \hat{G}^t\right\|_F^2 + c\left\|\Theta^t - \eta G^t - \hat{\Theta}^t\right\|_F^2\right]$$

$$+ \frac{192n}{1-\gamma}\left[\frac{2(1-\gamma)}{\rho^3\gamma^3} + \frac{320\bar{\omega}^2}{\rho^3\delta^2\gamma} + \frac{15\bar{\omega}^2}{\rho^2\delta^2\gamma}\right]\sigma^2\eta^2 + \frac{12n}{1-\gamma}\left[\frac{1-\gamma}{\rho\gamma} + \frac{256\bar{\omega}^2}{\rho\delta^2}\gamma + \frac{58\bar{\omega}^2}{\rho\delta^2}\right]\eta^2\mathbb{E}\left[\left\|\nabla f(\bar{\theta}^t)\right\|^2\right]$$

The proof is completed.

# C    Additional Numerical Results

In the first two sections we provide additional plots and the tuned hyper-parameters of our simulation. In the last section we describe the implementation details of our simulation.

## C.1    Synthetic Dataset

This dataset is generated from the benchmark framework `leaf` (Caldas et al., 2019). The number of data points possessed by each agent is different. In particular, we have the distribution $\{m_i\}_{i=1}^{25}$ which follows [470, 403, 91, 84, 79, 51, 51, 38, 31, 25, 24, 19, 14, 10, 9, 6, 6, 5, 5, 4, 4, 4, 4, 3, 3]. Table 2 provides the tuned parameters used for the experiment in Figure 1.

Table 2: Tuned hyper-parameters for linear model on synthetic dataset.

| Algorithms | Learning rate $\eta$ | Consensus step size $\gamma$ | Momentum param. $\beta$ | Batch size |
|---|---|---|---|---|
| GNSD | 0.005 | - | - | 4 |
| DeTAG ($R = 1$) | 0.005 | 0.1 | - | 4 |
| GT-HSGD | 0.01 | - | 0.01 | 2 |
| CHOCO-SGD (Top-k 10%) | 0.01 | 0.5 | - | 4 |
| CHOCO-SGD (Random Quant. 8bits) | 0.01 | 0.9 | - | 4 |
| BEER (Top-k 5%) | 0.01 | 0.16 | - | 100 |
| BEER (Ramdon Quant. 4bits) | 0.01 | 0.5 | - | 100 |
| DoCoM (Top-k 5%) | 0.01 | 0.2 | 0.01 | 2 |
| DoCoM (Random Quant. 4bits) | 0.01 | 0.6 | 0.01 | 2 |

In Figure 4, we provide additional numerical results on the trajectories of gradient norm, training/testing accuracy of the algorithms. Similar comparisons between `DoCoM` and existing algorithms are observed. Notably, we see that `DoCoM` achieves the best gradient stationary solution in limited communication budget and recovers the same level of gradient stationary solution as the uncompressed `GT-HSGD` at the last iteration. Also, we observe that `DoCoM` is the first to achieve the best accuracy with the least network cost.

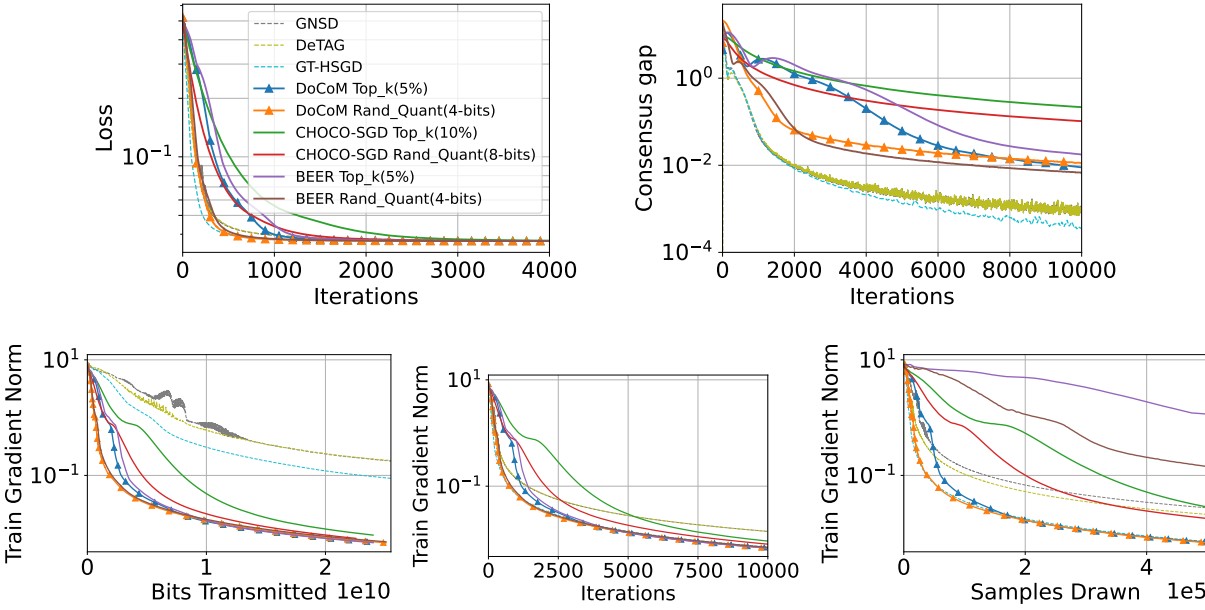

Figure 4: **Additional Results on Synthetic Data and Linear Model.** Loss and consensus gap against iterations, and worst-agent's gradient norm, training/testing accuracy.

## C.2 1 Layer Feed-forward Network on MNIST Dataset

Table 3 summarizes the tuned hyper parameters used by the experiment in Figure 2. We provide additional plots against wall clock time to demonstrate the practical improvement one can achieve using DoCoM.

Table 3: Tuned hyper-parameters for 1 layer feed-forward network on MNIST.

| Algorithms | Learning rate $\eta$ | Consensus step size $\gamma$ | Momentum param. $\beta$ |
|---|---|---|---|
| CHOCO-SGD (Top-k 10%) | 0.01 | 0.3 | - |
| BEER (Top-k 5%) | 0.01 | 0.2 | - |
| DoCoM (Top-k 5%) | 0.01 | 0.2 | 0.01 |

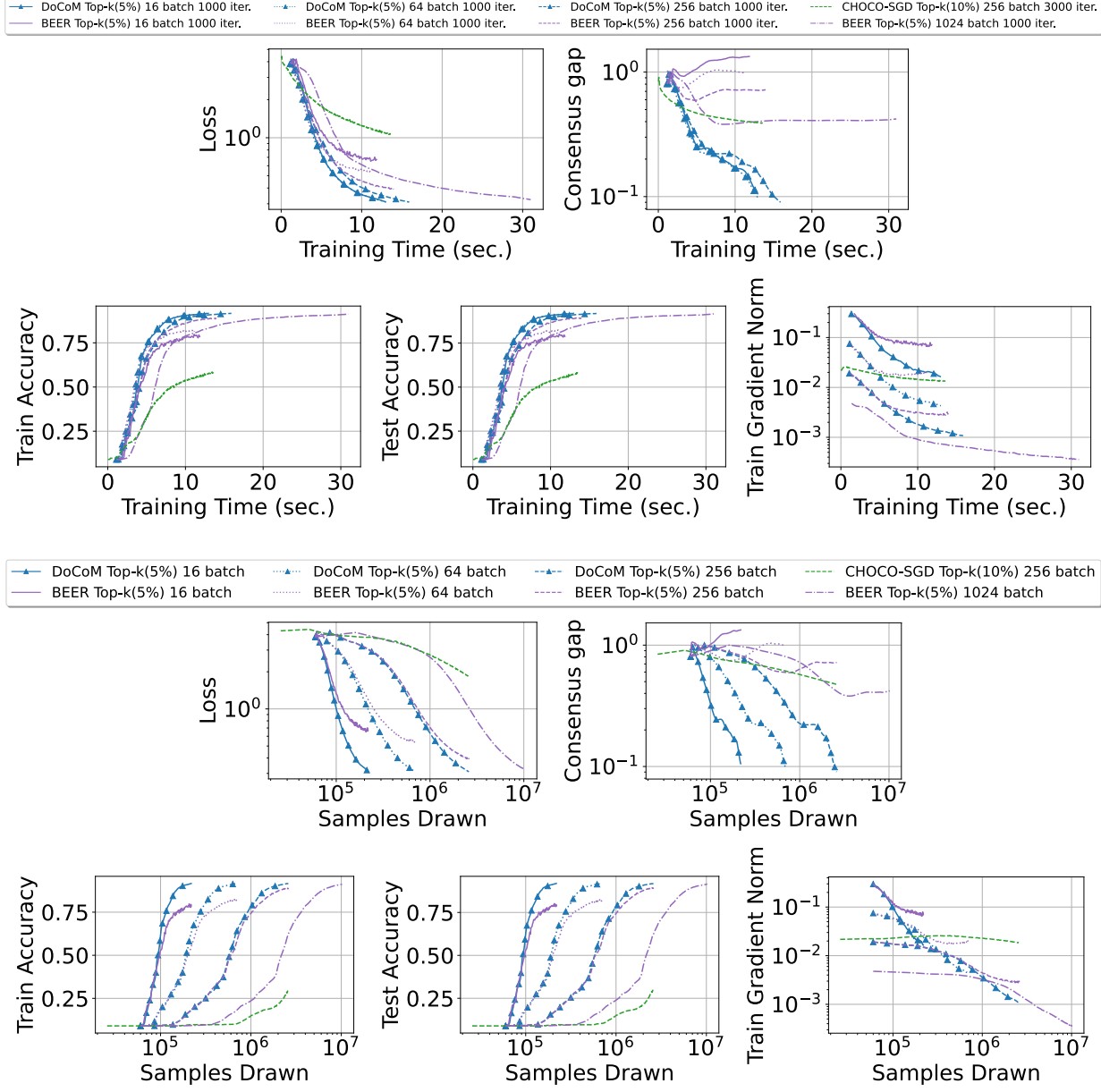

Figure 5: **Additional Results on MNIST Data with Feed-forward Network.** Worst-agent's loss, consensus gap, training/testing accuracy and gradient norm against wall clock time in seconds and the number of samples drawn. In the legend of wall clock time plots we denoted the number of iterations used for training.

### C.3 LeNet-5 on FEMNIST Dataset

We conduct another practical experiment and consider training a LeNet-5 neural network which has $d = 60850$ parameters. The dataset contains $m = 805263$ samples of $28 \times 28$ hand-written character images, each belongs to one of the 62 classes. The samples are partitioned into $n = 36$ agents (of ring topology) according to the groups specified in (Caldas et al., 2019). We scheduled the learning rate at its initial value for the initial stage, then decay by a factor of $10^{-1}$ at certain iterations. This allows us to perform large step size training at the initial stage and arrive at a consensual solution eventually.

Additionally, we observe that when training LeNet-5 using momentum-based variance reduced algorithms (including `GT-HSGD` and `DoCoM`), we found that the algorithms can be unstable when a small momentum parameter (e.g., $\beta = 0.1$) is adopted, unlike the experiments on synthetic data / MNIST. We suspect that this is due to the stronger requirements on Lipschitz continuity of the gradient of objective function in Assumption 2.1, i.e., we require [1]

$$\mathbb{E}_\zeta \left[ \|\nabla f_i(\theta; \zeta) - \nabla f_i(\theta'; \zeta)\|^2 \right] \le L^2 \|\theta - \theta'\|^2, \ \forall \ \theta, \theta' \in \mathbb{R}^d.$$

This is stronger than the typical Lipschitz continuity on the *expected gradient* which only demands $\|\mathbb{E}[\nabla f_i(\theta; \zeta)] - \mathbb{E}[\nabla f_i(\theta'; \zeta)]\| \le L\|\theta - \theta'\|$. Particularly, the convergence of these momentum-based algorithms depend on the less smooth loss landscape from the neural network model and data distribution.

Table 4: Tuned hyper-parameters for LeNet-5 on FEMNIST. Numbers on the $\eta, \beta$ scaling schedule columns indicate the iteration number (in thousands) when $\eta, \beta$ are scaled by 0.1 correspondingly.

| Algorithms | Learning rate $\eta$ | Consensus step size $\gamma$ | Momentum param. $\beta$ | Batch size | $\eta$ scaling schedule | $\beta$ scaling schedule |
|---|---|---|---|---|---|---|
| GNSD | 0.02 | - | - | 32 | 4, 20, 35 | - |
| GT-HSGD | 0.02 | - | 0.3 | 16 | 4, 20, 35 | 10, 20, 35 |
| CHOCO-SGD (Top-k 10%) | 0.01 | 0.25 | - | 32 | 4, 40, 80 | - |
| BEER (Top-k 5%) | 0.01 | 0.15 | - | 32 | 4, 40, 80 | - |
| DoCoM (Top-k 5%) | 0.01 | 0.2 | 0.3 | 16 | 4, 40, 80 | 10, 40, 80 |

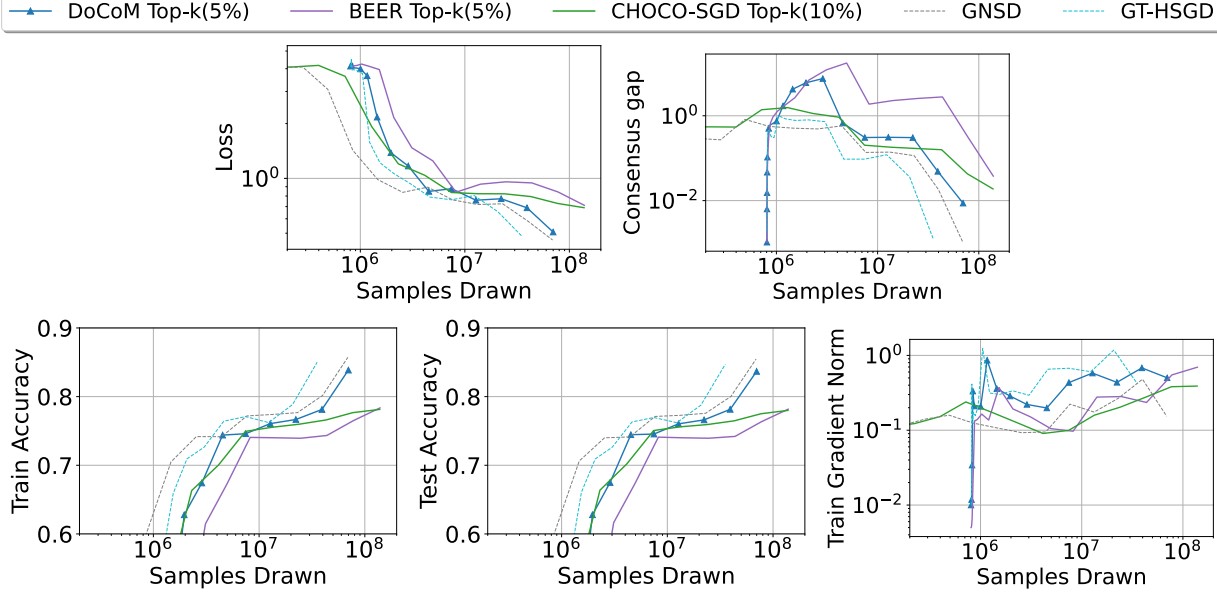

Figure 6: **Additional Results on FEMNIST Data with LeNet-5.** Worst-agent's loss, consensus gap, training/testing accuracy and gradient norm against the number of samples drawn.

[1] This assumption is known as the mean squared smoothness condition (Arjevani et al., 2022). Specifically, this assumption is only used in the proof of Lemma 3.6. The analysis for $\beta = 1$ can be further relaxed to the typical smoothness assumption $\|\mathbb{E}[\nabla f_i(\theta; \zeta)] - \mathbb{E}[\nabla f_i(\theta'; \zeta)]\|^2 \le L^2 \|\theta - \theta'\|^2$, applied in the proof of Lemma B.6.

### C.4 Implementation Details

**Compression Operators.** We adopt either a greedy biased compressor Top-k (sparsification) or a re-scaled random compressor (quantization) (Alistarh et al., 2017) in our experiment for algorithms with compressed communication. Specifically, a re-scaled $b$-bits random quantization (Koloskova et al., 2019) on $x \in \mathbb{R}^d$ that satisfies Assumption 2.4 can be described as

$$\mathcal{Q}(x_i; \xi_i) = \frac{\text{sign}(x_i) \|x\|_2 \xi_i}{\tau}, \quad \xi_i = \begin{cases} \ell_i/2^b & \text{w.p. } 1 - p(|x_i|/\|x\|_2, 2^b); \\ (\ell_i + 1)/2^b & \text{otherwise.} \end{cases} \tag{77}$$

with level $\ell_i$ of $x_i$ satisfying $|x_i|/\|x\|_2 \in [\ell_i/2^b, (\ell_i + 1)/2^b]$, $\tau = 1 + \min\{d/(2^b)^2, \sqrt{d}/2^b\}$, $p(a, s) = as - \ell$. With the unbiasedness of random quantization (Alistarh et al., 2017) that gives $\mathbb{E}_\xi[\mathcal{Q}(x; \xi)] = \frac{1}{\tau}x$, and with the proof in Appendix A.1 of (Alistarh et al., 2017) that gives $\mathbb{E}_\xi[\|\mathcal{Q}(x; \xi)\|^2] \leq \frac{1}{\tau}\|x\|^2$,

$$\mathbb{E}_\xi[\|\mathcal{Q}(x) - x\|^2] = \mathbb{E}_\xi[\|\mathcal{Q}(x; \xi)\|^2] + \|x\|^2 - 2\mathbb{E}_\xi[\mathcal{Q}(x; \xi)]^\top x \leq (\frac{1}{\tau} + 1 - \frac{2}{\tau})\|x\|^2 = (1 - \frac{1}{\tau})\|x\|^2. \tag{78}$$

Therefore, a re-scaled random quantizer satisfies Assumption 2.4 with $\delta = 1/\tau$. Moreover, as pointed out in (Koloskova et al., 2019, Section 3.5), the above re-scaling trick can be applied on any unbiased compressor $\mathcal{Q}(\cdot)$ satisfying $\mathbb{E}_\xi[\|\mathcal{Q}(x; \xi)\|^2] \leq \tau\|x\|^2$ such that the rescaled compressor $(1/\tau)\mathcal{Q}(\cdot)$ satisfies Assumption 2.4 with $\delta = 1/\tau$.

For Top-$k$ compressor, the communication cost per iteration is $k \cdot b_{\text{pre}}$ bits, where $b_{\text{pre}}$ is the number of bits for representing a full-precision scalar, added to the cost for sending $k$ indices. For random quantization with $b$ bits, the communication cost per iteration is $(b + 1)d$ bits added to the cost of sending the $\ell_2$-norm of the vector as a full-precision floating point scalar.

Table 5: Comparison of decentralized optimization algorithms on computational complexity per iteration and node. $b$ denotes the batch size used at every iteration. $b_0$ denotes the initial batch size. $R$ denotes the potentially multiple rounds of gossip for DeTAG. $d$ denotes the model dimension. †BEER uses large batch size of $\mathcal{O}(1/\epsilon^2)$.

| Algorithms | Init. Stoch. Grad. | Stoch. Grad. | Gossip Comm. Round | Memory Usage |
|---|---|---|---|---|
| GNSD | $b$ | $b$ | 2 | $3d$ |
| DeTAG | $b$ | $b$ | $R$ | $3d$ |
| GT-HSGD | $b_0$ | $2b$ | 2 | $5d$ |
| CHOCO-SGD | $b$ | $b$ | 1 (+ Compress) | $3d$ |
| BEER | $b^\dagger$ | $b^\dagger$ | 2 (+ Compress) | $7d$ |
| DoCoM | $b_0$ | $2b$ | 2 (+ Compress) | $9d$ |

**Memory Efficient Implementation.** From line 1 and 1 of Algorithm 1, we observe that DoCoM relies on the sum $\sum_j W_{ij}\widehat{\theta}_{i,j}^t$ and $\sum_j W_{ij}\widehat{g}_{i,j}^t$. Similar to the steps described in Appendix E of (Koloskova et al., 2019) for the CHOCO-SGD algorithm, the DoCoM algorithm can be implemented with a per-node memory complexity of $\mathcal{O}(d)$. See Table 5 for details.

## D Connection between Assumption 2.2 and Spectral Gap

Conditions (i), (ii) of Assumption 2.2 are standard in the literature of decentralized optimization, while at the same time (iii) is equivalent to the spectral gap condition. To see this, we suppose $\mathbf{W}$ is the weighted adjacency matrix of a connected graph and note from the Perron-Frobenius theorem that $\mathbf{1}$ is the eigenvector corresponding to the leading eigenvalue of $\mathbf{W}$ which has multiplicity of 1. It follows from orthogonality of $\mathbf{U}$ and $\mathbf{U}\mathbf{U}^\top = \mathbf{I}_n - (1/n)\mathbf{1}\mathbf{1}^\top$ that

$$\|\mathbf{U}^\top \mathbf{W}\mathbf{U}\|_2 = \|\mathbf{U}\mathbf{U}^\top \mathbf{W}\mathbf{U}\mathbf{U}^\top\|_2 = \max\{\lambda_2, |\lambda_n|\} \tag{79}$$

Therefore, condition (iii) that asserts the existence of $\rho \in (0, 1]$, $\max\{\lambda_2, |\lambda_n|\} \leq 1 - \rho$ is equivalent to $\max\{\lambda_2, |\lambda_n|\} < 1$. Lastly, it is obvious that $\|\mathbf{W} - \mathbf{I}\|_2 \leq \|\mathbf{W}\|_2 + \|\mathbf{I}\|_2 \leq 2$ and we used $\bar{\omega} \in (0, 2]$ in (iv) of Assumption 2.2 to simplify notations.

