# OpenReview forum: "DoCoM: Compressed Decentralized Optimization with Near-Optimal Sample Complexity"
_TMLR — Accepted by TMLR_

### Review · Reviewer_insd · 2023-06-02

**Summary Of Contributions:**

This work proposes the first compression-enabled decentralized optimization algorithm called DOCOM that achieves near-optimal sample complexity, and the complexity result does not require any assumption on the data heterogeneity level nor the boundedness of gradient. To achieve this goal, DOCOM uses compression with gradient tracking and momentum-based variance reduction. Experimental results on both synthetic and real-world data demonstrate the sample and communication efficiency of the proposed DOCOM.

**Audience:**

Yes

**Broader Impact Concerns:**

I did not see any broader impact concerns for this theoretical paper.

**Claims And Evidence:**

Yes

**Requested Changes:**

**Requested changes (critical for acceptance):**

(1) In the italic research question of the introduction, I think "iteration/sampling complexity" could be changed to "sample complexity", since the iteration complexity could further be reduced using minibatch.

(2) In the caption of Table 1, change "dominate" to "dominant".

(3) In the experiment section, you said ''where each agent only get''. You could change ''get'' to ''gets''.

(4) Fix “In Figure ??” in Section C.3.

(5) In the experiment section, what's $f_i(\theta)$ for FEMNIST Data with LeNet-5? You could write that answer in the paper.

**Requested changes (not critical for acceptance):**

(6) In the second paragraph of the introduction, "also see (Gorbunov et al., 2019)" looks vague. Why not move "(Gorbunov et al., 2019)" to before "studied algorithms with" or before "focused on non-convex problems"?

(7) You could add citations to Table 1 if convenient.

(8) At the end of the third paragraph of the introduction, do you mean "See the recent work (Richtárik et al., 2021) for more details on compression strategies."?

(9) Assumption 2.2 (iv) will certainly hold with $\overline{\omega}=2$, since Assumption 2.2 (ii) and (iii) imply that the eigenvalues of $W$ are real numbers in $[-1,1]$. Isn't it simpler to remove Assumption 2.2 (iv) and let $\overline{\omega}=2$ throughout the paper?

(10) Between Assumptions 2.2 and 2.3, add parenthesis in ''see (Boyd et al. (2004))''.

(11) You could mention the name of the variance reduction technique. Like STORM or Hyrbrid SGD?

(12) $=$ in eq. (12) is better to be $\le$.

(13) In the experiment section, there are figures with the x-axis "samples used" for synthetic data, but there are no such figures for MNIST and FEMNIST datasets. Why? Could you add the latter? For example, Figures 2 and 3 could each have 8 subfigures of which 4 have x-axis "samples used".

**Strengths And Weaknesses:**

**Strengths:**
The selected topic of decentralized optimization is popular and important. The paper is well written. The presentation is very clear. The contribution in near-optimal sample complexity with compression is non-trivial. I did not check the proof, but based on my knowledge of STORM variance reduction and decentralized optimization, the algorithm design and theoretical results look valid. There are also sufficient experimental results on both synthetic and real-world data which demonstrate the sample and communication efficiency of the proposed DOCOM.

**Weaknesses:**
A few omissions in detail are listed in "Requested Changes" below.

---

> ### Author Response · Authors · 2023-06-30
> **Response to Reviewer insd**
>
> Thank you for all the suggestions. We have corrected all the typos mentioned by the reviewer. The updated parts have been highlighted in green. Below are the responses to some specific questions:
>
> >(9) Assumption 2.2 (iv) will certainly hold with $\overline{\omega}=2$, since Assumption 2.2 (ii) and (iii) imply that the eigenvalues $W$ of are real numbers in $[-1,1]$. Isn't it simpler to remove Assumption 2.2 (iv) and let $\overline{\omega}=2$ throughout the paper?
>
> Thanks for the suggestion. While setting $\overline{\omega} = 2$ will not affect the main results in the paper, in most cases $|| {\bf W} - {\bf I} ||$ can be significantly smaller than $2$. As such, we decided to keep the current presentation to allow for a potentially tighter bound.
>
> >(11) You could mention the name of the variance reduction technique. Like STORM or Hyrbrid SGD?
>
> The variance reduction technique comes with a few existing names, such as Momentum SARAH [[Liu et al, 2020]](https://arxiv.org/pdf/2008.09055.pdf) and Hybrid SGD [[Tran-DinhTran-Dinh et al., 2020]](https://arxiv.org/pdf/1907.03793.pdf). At the same time, this technique is closely related to STORM [[Cutkosky and Orabona, 2020]](https://arxiv.org/pdf/1905.10018.pdf) except that our proof does not require adaptive step size. We decided to use "Momentum-based variance reduction" to highlight on the use of momentum in the variance reduction technique.
>
> >(13) In the experiment section, there are figures with the x-axis "samples used" for synthetic data, but there are no such figures for MNIST and FEMNIST datasets. Why? Could you add the latter? For example, Figures 2 and 3 could each have 8 subfigures of which 4 have x-axis "samples used".
>
> To keep our discussion in Section 4 focused and easy to understand, we have put detailed figures such as performance against "samples drawn" in the revised Appendix C.2-C.3.

---

> > ### Comment · Reviewer_insd · 2023-07-01
> > **Reviewer insd's questions are well addressed.**
> >
> > Reviewer insd's questions are well addressed.
> > Thanks for the authors' efforts.
> >
> > Reviewer insd

---

### Review · Reviewer_9Bad · 2023-06-08

**Summary Of Contributions:**

The authors propose a training algorithm in decentralized settings that uses compression with gradient tracking in addition to momentum-based variance reduction. The convergence of Doubly Compressed Momentum-assisted Stochastic Gradient Tracking does not require bounded similarity assumption between data distributions. They establish a rate of ${\cal O}(1/T^{2/3})$ for smooth optimization and the averaged iterates. They rate is imrpoved to ${\cal O}(\log(T)/T)$ under PL condition and stocahstic gradients.


**Audience:**

Yes

**Claims And Evidence:**

No

**Requested Changes:**

Please address the comments listed in the weaknesses above.

**Strengths And Weaknesses:**

Strengths:

The main strength of the paper is that the convergence bounds do not require assumptions on the data heterogeneity and the boundedness of gradient as previous work, which seems to be due to gradient tracking.

-----------------

Weaknesses:

- I am mainly concerned about the rates in Theorem 3.1. When $n$ is large, $\eta_{\infty}$ and $\eta$ will be very close to zero (23). I am not sure how the authors could set $\eta$ as shown in the paragraph after Theorem 3.1 given that they should satsify (23). Satisfying the conditions together will result in a bound on $n$ in terms of $T$, which should be discussed clearly. Such a bound will change the rate in terms of $T$.

A similar comment applies to Corollary 3.1.

- Discussion of lower bound: "Such sampling complexity matches the complexity
lower bound for stochastic first order algorithms (Arjevani et al., 2022)"

For smooth optimization and without additional assumptions, Theorem 3 (Arjevani et al., 2022) establishes a lower bound in $\Omega(\epsilon^{-4})$. They also establish another lower bound in $\Omega(\epsilon^{-3})$ but under different sets of assumptions compared to this paper. The authors should clarify whether the improved upper bound in this paper is due to additional assumptions? Please clarify the connections with the lower bounds in Theorem 3 (Arjevani et al., 2022)  considering that the papers have different sets of assumptions.

- The condition in 2 is a bit strong. What if the upper bound does not hold for some points in the support with some negligible probability measure. Imposing this condition will result in a relatively large $L$.

- Assumption 2.2. (iii) and (iv) are not very standard in my view. The authors state that such assumptions are satisfied with a particular weight matrix. However, it is not clear whether they hold for all ${\bf W}$'s that satisfy Assumption 2.2. (i) and (ii).

- Compressor: "Other compressors such as random quantization can also satisfy (5)"
This is not necessarily correct. For unbiased quantization schemes e.g., QSGD, TernGrad, and NUQSGD, the coefficient in the upper bound of (5) is  greater than one although they usually outperform the biased compression schemes.

- "we note that DoCoM shares a similar communication and computation cost per iteration as CHOCO-SGD"
This is not accurate.  Considering the updates in DoCoM, the additional computation (backpropagation) compared to the original CHOCO-SGD or decentralized method with unbiased compression schemes is at least three fold since the stochastic gradients are obtained for different points in the space of parameters of different baches. It is unclear whether those baselines without variance reduction can outperform DoCoM using equivalent batches for example.

- Checking the tuned hyperparameters in Appendix C, it seems the momentum is set to 0.01 in most experiments, which essentially means no momentum is applied. This does not validate the theory where the authors state that adding momentum provably accelerates convergence, e..g,  "the momentum term is crucial in accelerating DoCoM" before the discussion of PL condition.

- On the MNIST experiments, different baselines have different batch sizes. The authors wrote in Section 4 that "We choose the batch sizes such that all algorithms spend the same amount of computation on stochastic gradient per iteration" However, it is not clear why for DoCoM and BEER multiple batch sizes are shown in the same plot? It is challenging to compare performance of different baselines with such bacth sizes.

---

> ### Author Response · Authors · 2023-06-30
> **Response to Reviewer 9Bad (1/3)**
>
> Thank you for your comments and careful review of our paper. The revised parts have been highlighted in green.
>
> >I am mainly concerned about the rates in Theorem 3.1. When $n$ is large, $\eta_{\infty}$ and $\eta$ will be very close to zero (23). I am not sure how the authors could set $\eta$ as shown in the paragraph after Theorem 3.1 given that they should satsify (23). Satisfying the conditions together will result in a bound on $n$ in terms of $T$, which should be discussed clearly. Such a bound will change the rate in terms of $T$.
>
> As detailed in the paragraph on "Near-optimal Iteration/Sample Complexity", we can satisfy the condition (23) (now (24) in the revised verision) by setting $\eta = {\cal O}(1/T^{1/3})$. Notice that $T$ refers to the total number of *iterations (i.e., samples used)* in the algorithm and is typically quite large ($10^5$ or above) for the stochastic algorithms of interest. Now, the condition (23) which relies on a constant $\eta_{\infty}$ *independent* of $T$ can be satisfied provided that $T$ is large enough. It is shown in (12) that the convergence rate of DoCoM will be ${\cal O}(1/T^{2/3})$ as desired. Similar argument applies to Corollary 3.3 with $\eta = \log T / T$.
>
> On the other hand, the condition (23) does impose a *lower bound* on $T$ in terms of $n$, i.e., it requires $T = \Omega(n^2)$. Such a lower bound on $T$, which is related to the *transient time* of DoCoM, is actually common in the literature of decentralized optimization. E.g., the convergence bound in the CHOCO-SGD algorithm requires $T = \Omega(n)$ [[Corollary A.4., Koloskova et al., 2020]](https://openreview.net/pdf?id=SkgGCkrKvH) to hold. In the revision, we show that this *lower bound* on $T$ can be improved to $T = \Omega(n)$ if one selects a slightly different set of step sizes parameters, though it has other trade-offs.
>
> Lastly, we observe that in practice, decentralized algorithms for training ML models are seldom used in setting with very large $n$ due to the availability of distributed computing resources. E.g., as seen in the numerical experiments conducted in [[Koloskova et al., 2020]](https://openreview.net/pdf?id=SkgGCkrKvH), [[Lian et al., 2017]](https://arxiv.org/pdf/1705.09056.pdf), the decentralized algorithms proposed therein are tested on cluster of machines with at most $n \approx 100$ nodes.
>
> >Discussion of lower bound: "Such sampling complexity matches the complexity lower bound for stochastic first order algorithms (Arjevani et al., 2022)". For smooth optimization and without additional assumptions ...
> >
> >The condition in 2 is a bit strong. What if the upper bound does not hold for some points in the support with some negligible probability measure. Imposing this condition will result in a relatively large $L$.
>
> We apologize for having used a more conservative assumption in the original submission. We have double checked the proofs of DoCoM and can confirm that the same mean square smoothness condition as in [[Eq. (4), Arjevani et al., 2022]](https://arxiv.org/pdf/1912.02365.pdf) can be used directly for the analysis of DoCoM without incurring any changes to the proof. The crucial parts to check are the proofs of Lemma 3.6 and Lemma A.2. Here, we note that for all $i \in [n]$, the random samples $\zeta^{t+1}_i$ are drawn *after* $\theta_i^t$ and $\theta_i^{t+1}$, as such the bounds on $\mathbb{E}_t [ || \nabla\widetilde{F}^{t+1} - \nabla\widetilde{F}^{t} ||_F^2 ]$ can be handled directly using the mean square smoothness condition. (Note that such extension has been briefly mentioned in a footnote on p.37 of the original submission.)
>
> The revised paper has incorporated the above mean square smoothness condition into the updated Assumption 2.1. It shows that the query (i.e., sample) complexity upper bound of DoCoM matches the lower bound $\mathcal{\Omega}(\epsilon^{-3})$ [[Theorem 3, Arjevani et al., 2022]](https://arxiv.org/pdf/1912.02365.pdf) under the same mean square smoothness assumption.

---

> ### Author Response · Authors · 2023-06-30
> **Response to Reviewer 9Bad (2/3)**
>
> >Assumption 2.2. (iii) and (iv) are not very standard in my view. The authors state that such assumptions are satisfied with a particular weight matrix. However, it is not clear whether they hold for all ${\bf W}$'s that satisfy Assumption 2.2. (i) and (ii).
>
> It is obvious that not all ${\bf W}$ satisfying Assumption 2.2 (i) and (ii) will satisfy (iii) and (iv), for example, when ${\bf W} = {\bf I}$. That said, the conditions (iii) and (iv) are in fact equivalent to some of the standard assumptions used in the decentralized optimization literature.
>
> For example, when ${\bf W}$ is a non-negative matrix corresponding to the weighted adjacency matrix of a connected, undirected graph, the said conditions are actually equivalent to [[Assm. 1, Koloskova, et al., 2020]](https://arxiv.org/pdf/1907.09356.pdf) or [[Assm. 1.2, Lian et al., 2017]](https://arxiv.org/pdf/1705.09056.pdf). To see this, we note from the Perron Frobenius theorem that ${\bf 1}$ is the eigenvector corresponding to the leading eigenvalue of ${\bf W}$ which has multiplicity of 1. It follows from orthogonality of $\mathbf{U}$ and ${\bf UU^\top} = {\bf I} - \frac{1}{n} {\bf 11^\top}$ that
>
> $$||{\bf U^\top W U} ||_2 = ||{\bf UU^\top WUU^\top} ||_2 = \max\\{\lambda_2, |\lambda_n| \\} $$
>
> This trick is well-studied in the literature such as [[Eq. (4), Boyd et al., 2004]](https://web.stanford.edu/~boyd/papers/pdf/fmmc.pdf). Therefore, condition (iii) that asserts the existence of $\rho \in (0,1]$, $\max\\{ \lambda_2, |\lambda_n| \\} \leq 1 - \rho$ is equivalent to $\max\\{ \lambda_2, |\lambda_n| \\} < 1$. This is the same condition used in [[Assm. 1, Koloskova, et al., 2020]](https://arxiv.org/pdf/1907.09356.pdf) or [[Assm. 1.2, Lian et al., 2017]](https://arxiv.org/pdf/1705.09056.pdf) (with slightly different definitions of $\rho$) as well as a number of literature in decentralized optimization.
>
> Lastly, it is obvious that $|| {\bf W} - {\bf I} ||_2 \leq || {\bf W} ||_2 + || {\bf I} ||_2 \leq 2$ and we used $\bar{\omega} \in (0,2]$ in (iv) of Assumption 2.2 to simplify notations.
>
> For completeness, we have included the above derivations in the revised Appendix D.
>
> >Compressor: "Other compressors such as random quantization can also satisfy (5)" This is not necessarily correct. For unbiased quantization schemes e.g., QSGD, TernGrad, and NUQSGD, the coefficient in the upper bound of (5) is greater than one although they usually outperform the biased compression schemes.
>
> Yes, the "plain" random quantization compressor does not satisfy (5) with $\delta \in (0,1]$. However, upon an appropriate re-scaling operation, the re-scaled compressor can satisfy the said condition (5). Our discussion about the class of compressors is inspired by [[Section 3.5, Koloskova et al., 2019b]](https://arxiv.org/pdf/1902.00340.pdf). As stated therein, any unbiased compressor satisfying $\mathbb{E}[|| {\cal Q}(x) ||^2] \leq \tau ||x||^2$ can be **re-scaled** as ${\cal Q}'(x) = (1/\tau) {\cal Q}(x)$ where ${\cal Q}'(x)$ will satisfy (5). One concrete example is the re-scaled random quantization with $s$ levels denoted by ${\cal Q}_{QSGD}(x) $: denote $\tau = 1 + \min \\{ d/s^2, \sqrt{d}/s \\}$, the following compressor
>
> $\mathcal{Q}(x) = \frac{1}{\tau} \mathcal{Q}_{QSGD}(x), ~\text{s.t. } \mathbb{E}[\mathcal{Q}(x)]  = \frac{1}{\tau}x$
>
> satisfies (5) because by the argument in [[Appen. A1, Alistarh et al., 2017]](https://arxiv.org/pdf/1610.02132.pdf) we have $\mathbb{E}[|| \mathcal{Q}(x) ||^2] \leq \frac{1}{\tau} ||x||^2$ and
>
> $$\mathbb{E}[|| \mathcal{Q}(x) - x ||^2] = \mathbb{E}[||\mathcal{Q}(x) ||^2] + ||x ||^2 - 2 \mathbb{E}[\mathcal{Q}(x)]^\top x \leq (\frac{1}{\tau} +1-\frac{2}{\tau}) ||x||^2 = (1-\frac{1}{\tau})||x||^2$$
> In addition to clarifying about the use of rescaled compressors to satisfy (5), we have included the above derivations in the revised Appendix C.4 for completeness.

---

> ### Author Response · Authors · 2023-06-30
> **Response to Reviewer 9Bad (3/3)**
>
> >"we note that DoCoM shares a similar communication and computation cost per iteration as CHOCO-SGD" This is not accurate. Considering the updates in DoCoM, the additional computation (backpropagation) compared to the original CHOCO-SGD or decentralized method with unbiased compression schemes is at least three fold since the stochastic gradients are obtained for different points in the space of parameters of different baches. It is unclear whether those baselines without variance reduction can outperform DoCoM using equivalent batches for example.
>
> Yes, for computation complexity, DoCoM is 2x that of CHOCO-SGD as we compute two instances of stochastic gradient $\nabla \widehat{f}_i^{t+1}, \nabla \widetilde{f}_i^t$ on the same mini-batch $\xi^{t+1}_i$ over $\theta_i^{t+1}$ and $\theta_i^{t}$. Note that they are of the same order w.r.t. the problem dimension $d$. We have clarified about that in the revised paper.
>
> For fair comparison, in the numerical experiments, we also compared DoCoM with other algorithms by halving the batch size of DoCoM (see Table 2 in the Appendix).
>
> >Checking the tuned hyperparameters in Appendix C, it seems the momentum is set to 0.01 in most experiments, which essentially means no momentum is applied. This does not validate the theory where the authors state that adding momentum provably accelerates convergence, e..g, "the momentum term is crucial in accelerating DoCoM" before the discussion of PL condition.
>
> Notice that from equation (9c), a smaller $\beta$ corresponds to including more momentum into the update. On the other hand, when $\beta=1$, no momentum is applied and the DoCoM algorithm is equivalent to a gradient tracking DSGD algorithm with compression.
>
> >On the MNIST experiments, different baselines have different batch sizes. The authors wrote in Section 4 that "We choose the batch sizes such that all algorithms spend the same amount of computation on stochastic gradient per iteration" However, it is not clear why for DoCoM and BEER multiple batch sizes are shown in the same plot? It is challenging to compare performance of different baselines with such bacth sizes.
>
> We explain in the revised Section 4 that the MNIST experiment aims to compare the effect of batch size among the compressed algorithms. We also added an extra run of BEER with batch size 16 and improved the linestyles. We see that DoCoM has a consistent performance across different batch size settings (16 to 256), whereas BEER has degraded performances under medium (64) to small (16) batch sizes. On the other hand, Figure 2 also compares the performance of all compressed baselines on the common batch sizes of 256, as well as comparing DoCoM to BEER on the common batch sizes of 16, 64, 256.

---

### Review · Reviewer_pmUt · 2023-06-22

**Summary Of Contributions:**

This paper proposed a new algorithm for decentralized optimization, which combines both gradient tracking, compression and momentum. Authors provided an improved sample complexities for both nonconvex smooth and PL condition cases, which nearly match known lower bounds. The proposed algorithm, compared to existing works, does not require some restrictive conditions, e.g., bounded gradient, bounded heterogeneity and large batch. Authors also provided numerical experiment results to validity the effectiveness of the proposed algorithm.

**Audience:**

Yes

**Broader Impact Concerns:**

Not applicable.

**Claims And Evidence:**

Yes

**Requested Changes:**

See above questions for more details.

**Strengths And Weaknesses:**

Strength:

1. New algorihm incorporating compression while achieving the near-optimal sample complexity for decentralized optimization.
2. Avoid some restrictive assumptions like bounded gradient/heterogeneity and large batch.
3. The flow of the paper is good and easy to understand.

Weakness:

1. Some comparison is a bit confusing. Authors mentioned the paper BEER (https://arxiv.org/abs/2201.13320), saying that "...... and results in a suboptimal sampling complexity......", but as far as I can see, their paper assumed only the population function $f$ to be Lipschitz smooth (Assumption 2.3), rather than your assumption that the estimator $f_i(\cdot;\zeta)$ is also Lipschtz smooth (Assumption 2.1), your assumption should be (much) stronger than their setting, which is a point that I think authors should clarify when comparing.

2. Also the "suboptimal" argument, due to their Assumption 2.3, I think their $O(\epsilon^{-4})$ complexity should have already matched with the lower bound in Arjevani's paper? If so, I think it is a bit unfair to say BEER is suboptimal? Could you please clarify this point, or do I miss any details?

3. In your algorithm Line 10, the computation of $\nabla \hat{f_i}^{t+1}$ and $\nabla\tilde{f_i}^{t+1}$ requires the same data $\zeta_i^{t+1}$, i.e., you need to reuse the data. My impression is that common gradient tracking applies things like
$$
v_{t+1}=v_t+\nabla f(\theta_{t+1};\zeta_{t+1})-\nabla f(\theta_t;\zeta_t).
$$
I think in some cases reusing may not be that plausible, e.g., online setting. Do you think the algorithm can be revised, or could you please elaborate more on the data reusing?

4. Regarding the time stamp of the TMLR submission, there appears some other closely related papers at least two months ago, authors may consider to add them into the text and have a comparison, e.g., Yan, Yonggui, et al. "Compressed decentralized proximal stochastic gradient method for nonconvex composite problems with heterogeneous data." arXiv preprint arXiv:2302.14252 (2023).

Generally I think the paper makes a good and interesting contribution to decentralized optimization, while there are still some points I am confused. I appreciate authors' efforts on the work, please definitely correct me if I misunderstand the details. Thank you very much.

---

> ### Author Response · Authors · 2023-06-30
> **Response to Reviewer pmUt**
>
>
> Thank you for the careful review and comments. We have revised the paper accordingly with the updated parts highlighted in green.
>
> >Some comparison is a bit confusing. Authors mentioned the paper BEER (https://arxiv.org/abs/2201.13320), saying that "...... and results in a suboptimal sampling complexity......", but as far as I can see, their paper assumed only the population function $f$
>  to be Lipschitz smooth (Assumption 2.3), rather than your assumption that the estimator $f_i(\cdot; \zeta)$ is also Lipschtz smooth (Assumption 2.1), your assumption should be (much) stronger than their setting, which is a point that I think authors should clarify when comparing.
>
> >Also the "suboptimal" argument, due to their Assumption 2.3, I think their $\mathcal{O}(1/\epsilon^4)$ complexity should have already matched with the lower bound in Arjevani's paper? If so, I think it is a bit unfair to say BEER is suboptimal? Could you please clarify this point, or do I miss any details?
>
> The reviewer is right about different assumption used in our paper and the BEER paper and we agree that our Assumption 2.1 is stronger than that used in the BEER paper. Also, when only Assumption 2.3 (of the BEER paper) is made, the $\mathcal{O}(1/\epsilon^4)$ complexity has already matched the lower bound in Arjevani's paper.
>
> In particular, we are not aware if BEER is able to take advantage of assumptions such as the mean square smoothness (our Assumption 2.1). In other words, even under the latter assumption, the analysis available for BEER seems to lead to a sampling complexity of $\mathcal{O}(1/\epsilon^4)$ only, which is suboptimal as pointed out in Arjevani's paper.
>
> We have clarified about these points in the revision with
>
> "... However, under the mean square smoothness assumption considered in this paper, the best known analysis for these algorithms only show a suboptimal sample complexity of $\mathcal{O}(\epsilon^{−4})$ as they do not incorporate a momentum-based variance reduction step as in DoCoM."
>
> > In your algorithm Line 10, the computation of $\nabla \hat{f_i}^{t+1}$ and $\nabla\tilde{f_i}^{t+1}$ requires the same data $\zeta_i^{t+1}$, i.e., you need to reuse the data. My impression is that common gradient tracking applies things like
> $v_{t+1}=v_t+\nabla f(\theta_{t+1};\zeta_{t+1})-\nabla f(\theta_t;\zeta_t)$
> I think in some cases reusing may not be that plausible, e.g., online setting. Do you think the algorithm can be revised, or could you please elaborate more on the data reusing?
>
> First, we would like to clarify that the gradient tracking step in DoCoM can be read from eq. (9d), where the difference of gradient estimator $v_i^{t+1} - v_i^t$ is added to gradient tracker.
>
> Back to the issue of data reusing. Under the setting with $\beta \neq 1$, at each iteration, the DoCoM algorithm requires the agent $i$ to (i) draw a data sample $\zeta$, and then (ii) evaluate two stochastic gradients on $\theta^{t+1}_i$ and $\theta^t_i$ using the *same* data sample $\zeta$. In other words, it requires re-using the same data sample.
>
> Unfortunately, we believe that this restriction is hard to relax as this design is crucial to proving the fast convergence rate of DoCoM. Particularly, it is critical to the proof of our Lemma 3.6 and Lemma A.2, where the latter lemma is instrumental to the proofs of our main results. Notice that the same limitation applies to existing works such as the Momentum SARAH estimator [[Liu et al, 2020]](https://arxiv.org/pdf/2008.09055.pdf) that combines SGD and SARAH from [[Nguyen et al., 2017]](https://arxiv.org/pdf/1703.00102.pdf).
>
> We acknowledge that this can be an algorithmic limitation of DoCoM and have added a sentence after (9) to clarify about the computation architecture needed. While it may not apply to all online settings, we believe that our algorithm is still applicable to scenarios where a gradient oracle $\nabla f_i (\cdot; \zeta)$ is accessible. Thanks for raising this point.
>
> >Regarding the time stamp of the TMLR submission, there appears some other closely related papers at least two months ago, authors may consider to add them into the text and have a comparison, e.g., Yan, Yonggui, et al. "Compressed decentralized proximal stochastic gradient method for nonconvex composite problems with heterogeneous data." arXiv preprint arXiv:2302.14252 (2023).
>
> Thank you for providing this recent work. We have added this paper to the list of related works.

---

### Decision · Action_Editors · 2023-07-24

**Recommendation:** Accept as is

**Comment:**

In this paper, the authors propose a communication-efficient decentralized optimization algorithm, named as Doubly Compressed Momentum-assisted stochastic gradient tracking (DoCoM). They demonstrate that DoCoM achieves improved sample complexities for both nonconvex smooth and PL condition cases, without relying on restrictive conditions such as bounded similarity between data distributions. Finally, the authors provide numerical results to evaluate the effectiveness of DoCoM.

All the reviewers are positive with this paper. The proposed DoCoM is strong in the sense that it is the only algorithm with compression and an $O(\epsilon^{-3})$ sample complexity. The contribution in near-optimal sample complexity with compression is non-trivial. The paper is well-written, and the theoretical results are supposed by empirical studies.


**Audience:**

Yes

**Claims And Evidence:**

Yes